# Dynamic de novo heterochromatin assembly and disassembly at replication forks ensures fork stability

Vincent Gaggioli[1,2,14], Calvin S. Y. Lo[1,14], Nazaret Reverón-Gómez[3,4], Zuzana Jasencakova[3,4], Heura Domenech[1], Hong Nguyen[1], Simone Sidoli[5,12], Andrey Tvardovskiy [5,13], Sidrit Uruci[1], Johan A. Slotman [6], Yi Chai[7], João G. S. C. Souto Gonçalves[8], Eleni Maria Manolika[1], Ole N. Jensen [5], David Wheeler [9], Sriram Sridharan[7], Sanjiban Chakrabarty[10], Jeroen Demmers [11], Roland Kanaar[1,2], Anja Groth [3,4] & Nitika Taneja [1]✉

Chromatin is dynamically reorganized when DNA replication forks are challenged. However, the process of epigenetic reorganization and its implication for fork stability is poorly understood. Here we discover a checkpoint-regulated cascade of chromatin signalling that activates the histone methyltransferase EHMT2/G9a to catalyse heterochromatin assembly at stressed replication forks. Using biochemical and single molecule chromatin fibre approaches, we show that G9a together with SUV39h1 induces chromatin compaction by accumulating the repressive modifications, H3K9me1/me2/me3, in the vicinity of stressed replication forks. This closed conformation is also favoured by the G9a-dependent exclusion of the H3K9-demethylase JMJD1A/KDM3A, which facilitates heterochromatin disassembly upon fork restart. Untimely heterochromatin disassembly from stressed forks by KDM3A enables PRIMPOL access, triggering single-stranded DNA gap formation and sensitizing cells towards chemotherapeutic drugs. These findings may help in explaining chemotherapy resistance and poor prognosis observed in patients with cancer displaying elevated levels of G9a/H3K9me3.

In eukaryotic cells, genetic information stored in DNA is packaged by histone proteins in nucleosomes. This fundamental unit of chromatin is composed of two copies of each core histone, H2A, H2B, H3 and H4, wrapped by about two turns of DNA consisting of 146 base pairs[1]. Histone post-translational modifications (PTMs) define chromatin environments that influence biological pathways such as gene expression and DNA replication and repair[2,3]. Repressive chromatin marks containing histone 3 lysine 9 methylation (H3K9me) and hypoacetylation promote a closed chromatin conformation that stabilizes nucleosomes and restricts the accessibility of the underlying DNA to, for example, maintain gene silencing[4–6]. H3K9me levels are balanced by the action of methyltransferases ('writers') and demethylases ('erasers')[7,8]. Misregulation of histone lysine methylation has been implicated in cancers and developmental disorders, and inhibitors of this process have shown promising results in pre-clinical studies[8–11]. Although links between chromatin conformation and gene regulation have been widely explored, and while recent studies highlight their role in DNA repair[12,13], the role of epigenome regulation in the response to replication stress is poorly understood.

The propagation of chromatin states through cell division relies on faithful restoration of chromatin on the new daughter strands during replication and requires a tight coordination of DNA replication with

histone dynamics[14]. During chromatin replication, exogenous and endogenous insults can impair fork progression, leading to fork stalling or collapse events that challenge genome stability[15,16]. Replication stress persists in cancers[17–19], including in early stages of cancer development[20–23]. How the chromatin landscape is modulated in response to replication stress remains largely unknown.

In this Article, we describe a checkpoint-regulated de novo heterochromatin assembly forming at replication forks in response to replication stress. We show that heterochromatin assembly is critical to maintain the chromatin landscape associated with fork protection while timely disassembly is critical to prevent access to non-canonical PRIMPOL-mediated repriming of forks that triggers genome instability. Such a process requires a fine regulation of the dynamics of 'writers' EHMT2/G9a and Suv39h1 and 'erasers' JMJD1A/KDM3A at replication forks, with potential clinical implications.

## Results

### H3K9me3 is enriched on chromatin under chronic replication stress

H3K9me3, a modification known to induce gene silencing, is enriched throughout many cancer genomes (Extended Data Fig. 1a)[24]. Yet, surprisingly, gene silencing is not systematically observed in these cancers[25,26], suggesting that the increased density of repressive epigenetic marks may be related to another biological process or hallmark of cancer, such as chronic endogenous replication stress[19,27]. To test this hypothesis, we investigated whether replicative stress results in the accumulation of H3K9me3 on chromatin in human foetal lung fibroblasts (TIG3 cells). The cells were treated with a low dose of hydroxyurea (HU) for several days to induce a progressive slowdown of the replication forks triggering DNA damage response (DDR) and the onset of senescence[28] (Extended Data Fig. 1b,c)[29,30]. We reasoned that conditions of persistent replication stress may result in the accumulation of epigenetic changes. To rule out the possibility that the changes detected upon prolonged HU treatment are a consequence of exit from the cell cycle[31], we used as a control cells rendered quiescent by contact inhibition[31] (Fig. 1a and Extended Data Fig. 1d). Upon treatment with low dose of HU, cells accumulated in S phase, and while DNA synthesis continued for several days, an increasing number of cells arrested in S phase, failing to incorporate bromodeoxyuridine (BrdU) (Extended Data Fig. 1b). We observed a decrease in mitotic cells 24 h after addition of HU and at day 2, as control cells became quiescent upon contact inhibition (Extended Data Fig. 1b,e). Cells challenged with HU over 1–6 days, but not quiescent cells, exhibited a progressive increase in H3K9me3 levels (Fig.1a,b) via immunoblotting on chromatin extracts, consistent with an increment in H3K9me3 levels upon oncogene-induced replicative stress conditions observed elsewhere[21].

To study the alteration of the epigenetic landscape upon replication stress, we performed a comprehensive analysis of histone PTMs by quantitative mass spectrometry on total histones from proliferating, quiescent and HU-treated cells. We confirmed a significant increase of H3K9me3 peptides under persistent replication stress compared with proliferating or quiescent cells (Fig. 1c). In addition, several modifications, including H3K36me2, H3K27me2/me3, H3K79me1/me2 and H4K20me2/me3 increased when cells were challenged with HU (Extended Data Fig. 1f and Supplementary Table 1). However, quiescent cells also exhibit elevated levels of H3K27me2/me3, H3K79me1/me2 and H4K20me2/me3, as previously reported[31]. Therefore, these changes cannot be attributed solely to replication stress but might reflect cell cycle arrest or withdrawal. Finally, H3K9me3 and H3K36me2 only showed replication-stress specific increase. Taken together, these results provide evidence that H3K9me3 accumulates at chromatin upon persistent replicative stress.

To determine if the genomic distribution of H3K9me3 is altered after prolonged exposure to HU, we performed chromatin immunoprecipitation followed by sequencing (ChIP–seq) for H3K9me3 on cells

subjected to persistent replication stress compared with proliferating or quiescent cells. H3K9me3 is preferentially detected at gene-poor repetitive regions and in a subset of unique gene loci[32,33]. Interestingly, upon replication stress induced by persistent treatment with HU, H3K9me3 showed a broader and more homogeneous distribution across the genome in contrast to distinct heterochromatin domains observed in proliferating cells (Fig. 1d). To compare in more detail the spatial distribution of the modification between the different conditions, we applied Hilbert curves, space-filling graphs that convert the data from its one-dimensional arrangement along the chromosome to a two-dimensional shape allowing for the visualization of the signal from a whole chromosome in a single plot preserving resolution and locality[34]. The Hilbert curves for H3K9me3 showed distinct patterns for proliferating and replication stress cells (Fig. 1e and Extended Data Fig. 2a). Large domains of H3K9me3 can be easily identified in the graphs for proliferating cells (darker dense areas), while cells experiencing prolonged replication stress exhibit a more homogeneous distribution of the modification. Intriguingly, we detected changes in the distribution of H3K9me3 for quiescent cells when compared with proliferating cells, although to a much lesser extent (Extended Data Fig. 2a). Together with our finding that global H3K9me3 levels increase upon persistent replication stress, the genome-wide redistribution of H3K9me3 supports the hypothesis of a stochastic accumulation of H3K9me3 at sites of fork stalling that happens across the genome in a cell population.

We next aimed to evaluate whether replication stress has a lasting impact on the epigenetic landscape[35]. We confirmed that after prolonged HU treatment cells progressively returned to proliferation after removing the drug (Extended Data Fig. 2b). To this end we derived single-cell clones from proliferating cells and allowed cells to recover from a prolonged treatment with HU (Fig. 1f). Mass spectrometry profiling of histone modifications revealed that global levels of H3K9me3 (along with other marks analysed) were restored to normal upon recovery from replication stress, making these clones appear remarkably similar to those derived from control cells (Fig. 1f and Extended Data Fig. 2c). The dynamic nature of this phenomenon suggests an active regulation by epigenetic 'writers' and 'erasers' orchestrating de novo H3K9me3 accumulation during replication stress and its removal upon recovery.

### Dynamic heterochromatin assembly and disassembly at replication forks

To gain mechanistic insights into the heterochromatin establishment pathway, we examined the response of cells exposed to short-term acute replication stress. We used super-resolution stimulated emission depletion (STED) microscopy on human lung fibroblast (MRC5) cells to observe the localization of H3K9me3 in replicating cells undergoing acute replication stress induced by 1 mM HU for 1 h. Untreated cells had a broad nuclear distribution of H3K9me3 with no specific overlap with DNA replication sites marked by short pulse (20 min) of 5-ethynyl-2′-deoxyuridine (EdU). However, in cells treated with HU a remarkable overlap between H3K9me3 and replication sites was visible (Fig. 2a and Supplementary Movies 1 and 2). This suggests that, similar to what we have observed upon chronic replication stress, there is chromatin modification at stressed replication sites. However, EdU foci comprise not one but several replication forks[36,37]. Therefore, to visualize chromatin composition directly at individual replication forks, we optimized the previously described technique of chromatin fibres[38,39] to isolate and stretch high numbers of individual chromatin fibres. This technology, which we named ChromStretch, produces high numbers of informative signals while being highly reproducible (Fig. 2b). We observed the histone H3 and accumulation of H3K9me3 mark along the single-molecule DNA fibres containing EdU-labelled replication forks/bubbles (Fig. 2b and Extended Data Fig. 3a,b). Analysis of H3K9me3 intensity along individualized chromatin fibres showed

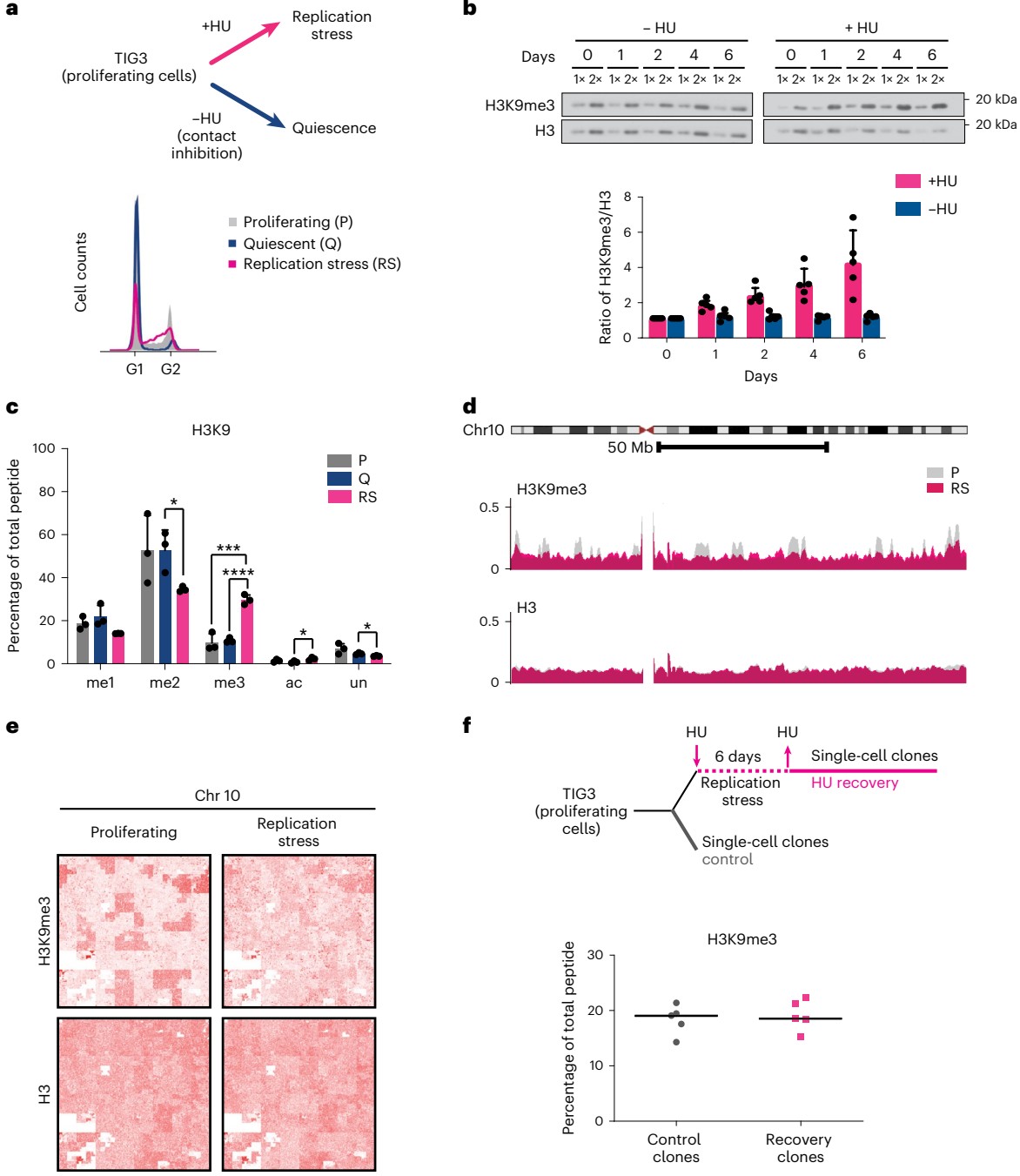

**Fig. 1 | Analysis of DNA replication and histone PTM dynamics under chronic replication stress condition. a**, Top: experimental design. TIG3 fibroblasts were cultured in the absence or presence of HU (600 µM) for at least 6 days, rendering cells quiescent due to contact inhibition or long-term exposed to replication stress, respectively. Bottom: cell cycle analysis of proliferating cells and cells treated with or without HU for 6 days. **b**, Top: time course analysis of H3K9me3 levels by immunoblotting on chromatin extracts from cells treated without (left) or with (right) HU for the indicated time. Representative western blots of five independent experiments. Histone H3 was used as a loading control for chromatin. Bottom, quantification of H3K9me3 levels relative to total H3 in chromatin extracts analysed by western blot. The graphs show the average $n = 5$ biological replicates with error bars indicating standard deviation. **c**, Analysis of H3K9 modification by mass spectrometry. Quantification of modifications on

the H3 peptide (amino acids 9–17) in proliferating (grey), quiescent (blue) and HU-treated (pink) TIG3 cells. The graph shows the average of three biological replicates with error bars indicating standard deviation. Unpaired two-sided $t$-test: \*\*\*\*$P < 0.0001$; \*\*\*$P < 0.001$; \*$P < 0.05$. For full histone PTM analysis, see Extended Data Fig. 1f. **d**, Overlay of ChIP–seq profiles at chromosome 10 for H3K9me3 and H3 in proliferating (P) and HU-treated cells (RS). **e**, Visualization of chromosome-wide profiles of ChIP–seq data for H3K9me3 and total H3 using Hilbert curves. See also Extended Data Fig. 2a. **f**, Analysis of H3K9me3 by mass spectrometry after recovery from HU. Top: experimental setup. Single-cell clones were derived from proliferating cells (control, grey) or cells allowed to recover after persistent replication stress (HU recovery, pink). Bottom: analysis by quantitative mass spectrometry. The lines represent the medians from $n = 5$ single-cell clones. For full histone PTM analysis, see Extended Data Fig. 2c.

the accumulation of this mark mainly at EdU-labelled sites undergoing replication stress (that is, HU treated) in comparison with untreated condition. Interestingly, the accumulation of H3K9me3 mark was correlated with increased levels of H3 at replicating sites undergoing replication stress (Extended Data Fig. 3c,d)[40]. This suggested an increased density of modified H3 nucleosomes and a more compact chromatin conformation at sites of replication stress, representing a fundamental feature of heterochromatin. Further, during a time course of 1 mM HU treatment, a significant increase of H3K9me3 levels at stressed replication sites could already be detected after 20 min of HU treatment and gradually increased till 1 h. After 1 h of HU treatment, most labelled replication sites were marked with H3K9me3, in contrast to untreated cells (Fig. 2c and Extended Data Fig. 4a), as confirmed by quantification of the H3K9me3 signal overlapping with replication sites showing an increase in the presence of HU (Fig. 2c, untreated versus HU conditions). We further tracked H3K9me3 modification upon fork restart by incorporating EdU for 20 min at various times after release from HU. Interestingly, we observed a significant reduction in H3K9me3 levels after 20 min of release and a full recovery of H3K9me status to match pre-HU treatment levels 30–45 min after release (Fig. 2d).

How is priming of de novo H3K9me3 at stressed forks executed? As increased levels of H3K9me1 were previously observed upon replication stress[41], we wondered if lower H3K9me modifications could be observed at sites of replication stress. Using ChromStretch, we observed a similar dynamic accumulation of H3K9me2 at EdU-labelled sites during the time course of HU treatment (Extended Data Fig. 4b). However, unlike the steeper shift in H3K9me3 signal from 30 min to 1 h HU treatment, an early saturation of signal was observed, suggesting that, rather than serving as a terminal histone mark, H3K9me2 represents a transient mark that is eventually converted into H3K9me3 (Fig. 2c and Extended Data Fig. 4b). Further, we observed a significant reduction in H3K9me2 signal upon release from HU stress, mirroring the H3K9me3 reduction observed upon fork restart (Extended Data Fig. 4c). Consistently, we observed enrichments of all three methylation states of H3K9 at the site of stressed replication forks using ChromStretch (Fig. 2e and Extended Data Fig. 4d–f), suggesting a sequential acquisition of me1, me2 and me3. To further validate these findings, we performed proximity ligation assays (PLAs) between replication sites labelled by EdU and H3K9me modifications, which were detected by high-content imaging of cells[42]. H3K9me3 as well as H3K9me1 and H3K9me2 (Fig. 3a–c) accumulated at replication sites upon HU-induced replication stress but not at ongoing (untreated condition) or restarted forks (HU release condition). Together, these data reveal that H3K9 methylation marks are transiently laid down at stressed replication forks.

Methylation of H3K9 is a sequential mechanism catalysed by histone methyltransferases (HMTs), starting with the deposition of precursor H3K9me1 and H3K9me2 marks, followed by the deposition of H3K9me3 (refs. 6,43,44). One of the main enzymes responsible for the deposition of H3K9me1/me2, the lysine methyltransferase G9a/EHMT2 (refs. 43,45), associates with replication forks[46,47]. To test whether G9a functionally affects stalled replication forks, we generated G9a knockout cells and, as an orthogonal approach, used UNC0642 (ref. 48), a highly specific and potent catalytic inhibitor of G9a/GLP, which blocks catalysis of H3K9 methylation on nucleosomes without affecting protein stability (Extended Data Fig. 4g). Both approaches showed that lack of G9a activity does not alter the cell cycle profile nor EdU incorporation efficiency (Extended Data Fig. 4h). Interestingly, we observed a drastic loss of all H3K9me1/me2/me3 marks in both G9a knockout cells as well as inhibitor (G9ai) treated conditions. This was confirmed by both PLA (Fig. 3a–c) and ChromStretch (Fig. 2e) and suggested these marks are established de novo at replication forks by G9a upon replication stress. We further noticed that the chromatin remodelling of forks upon replication stress depends on the activation of DNA replication checkpoint, as its inhibition eliminated the transient accumulation of H3K9me3 or accumulation of G9a at stressed replication sites (Fig. 3e and Extended Data Fig. 4i)[47,49].

As G9a is well known to catalyse H3K9me1/me2 more efficiently than H3K9me3 in vivo, we tested the involvement of other HMTs, such as SETDB1 or SUV39h1, which catalyse H3K9me3 (refs. 6,44). The accumulation of H3K9me3 upon HU treatment was drastically abrogated upon transient knockdown of SUV39h1 but remained unaffected by loss of SETDB1 (Fig. 3f and Extended Data Fig. 5a). Since, biochemically SUV39h1 catalyses mono-, di- and trimethylation on H3K9 (refs. 50,51), we wondered if SUV39h1 contributes to adding the lower K9me1 modifications. The substantial reduction in H3K9me1 levels at stressed forks upon transient depletion of SUV39h1 (Extended Data Fig. 5b) provides support for a model in which checkpoint-activated G9a initiates a platform of H3K9me1/me2 in conjunction with SUV39h1. This platform facilitates the 'reading' and 'writing' of lower H3K9me1 marks and catalyses the higher H3K9me3 modification on nucleosomes deposited at stressed replication sites (Fig. 3g and Extended Data Fig. 5b). We suggest here that these enzymes transiently heterochromatinize the local chromatin environment at stressed replication forks. This repressive state was further supported by transient enrichment of HDAC1 and deacetylation of lysine 16 on histone H4 (H4K16ac deacetylation)[52] observed specifically at stressed forks in contrast to untreated or HU release condition (Extended Data Fig. 5c,d). Furthermore, enrichment of both HDAC1 and deacetylated H4K16 marks at stressed forks showed dependency on the H3K9 methylation platform catalysed by G9a (Extended Data Fig. 5c,d).

**Fig. 2 | De novo H3K9me3 accumulates at stalled replication forks in a G9a-dependent manner. a**, The distributions of active replication sites (red) and H3K9me3 (green) were compared using super-resolution microscopy. Left: representative STED images of untreated (UT) and HU treated (HU) nuclei. Middle: representative intensity profile of the EdU signal (red) and H3K9me3 signal (green) extracted from STED images (left). Right: 3D reconstruction of untreated (UT) and HU treated (HU) nuclei imaged using STED microscopy and illustrating the accumulation of H3K9me3 at replication sites (yellow) upon HU treatment. $n = 5$ cells examined per condition over two independent experiments with similar results. **b**, Top: representative image of chromatin fibres acquired by ChromStretch in the absence of HU treatment (left) or after HU treatment (right) and stained for EdU (red), H3K9me3 (green) and H3 (blue). Bottom: intensity profiles of EdU (red), H3K9me3 (green) and H3 (blue) of the representative fibres indicated by the black arrows, in the absence (left) or after HU treatment (right). $n = 10$ fibres examined per condition over two independent experiments with similar results. **c**, Analysis of the dynamics of H3K9me3 at replication sites upon replication stress using ChromStretch. Top, experimental design: Cells were first labelled for 20 min with EdU and treated with 1 mM HU for the indicated amount of time. Bottom: quantification of H3K9me3 signal overlapping with

EdU ($n_{UT} = 106$, $n_{HU10} = 100$, $n_{HU20} = 104$, $n_{HU30} = 104$, $n_{HU60} = 104$ EdU tracks were analysed; ****$P \leq 0.0001$, *$P \leq 0.05$, NS, non-significant, Kruskal–Wallis test followed by Dunn's test). **d**, Analysis of the dynamics of H3K9me3 at replication sites after release from replication stress using ChromStretch. Left: experimental design. Cells were first treated with 1 mM HU for 1 h and released in medium without HU. At the indicated time post release, cells were labelled with EdU for 20 min. Single chromatin molecule was isolated using ChromStretch. Right: quantification of H3K9me3 signal at individual ($n$) replication sites ($n_{UT} = 100$, $n_{HU} = 111$, $n_{rel20} = 120$, $n_{rel30} = 100$, $n_{rel45} = 127$, $n_{rel60} = 118$ EdU tracks were analysed; ****$P \leq 0.0001$, NS, non-significant, Kruskal–Wallis test followed by Dunn's test). **e**, Quantification of H3K9me1 (left), H3K9me2 (middle) and H3K9me3 (right) at replication sites in the presence or in the absence of G9a activity (UNC0642 − and +, respectively) both at ongoing (UT) and stressed (HU) replication forks using ChromStretch. The number of replication tracks analysed was: for H3K9me1(left): $n_{UT-} = 107$, $n_{UT+} = 106$, $n_{HU-} = 131$, $n_{HU+} = 101$; H3K9me2 (middle): $n_{UT-} = 73$, $n_{UT+} = 51$, $n_{HU-} = 55$, $n_{HU+} = 88$; H3K9me3 (right): $n_{UT-} = 67$, $n_{UT+} = 68$, $n_{HU-} = 123$, $n_{HU+} = 94$ EdU tracks were analysed; ****$P \leq 0.0001$, NS, non-significant, Kruskal–Wallis test followed by Dunn's test). Source numerical data are available in Source Data.

A condensed state of heterochromatin is maintained by suppressing nucleosome turnover[53,54]. To further characterize the changes in chromatin structure in response to acute replication stress, we monitored chromatin expansion/compaction representing status of nucleosome turnover, by activating histone H2A fused to a photo-activatable version of GFP (PA-GFP)[55–58]. We generated isogenic cell lines stably expressing mCherry-tagged PCNA and PA-GFP-H2A to compare the chromatin structure of replicating versus non-replicating cells simultaneously in presence or absence of replication stress (Fig. 3h and Extended Data Fig. 5e)[59]. We compared the evolution of PA-GFP-H2A tracks in PCNA-mCherry negative (control cells) or PCNA-mCherry positive (test cells), in untreated cells and in cells undergoing acute

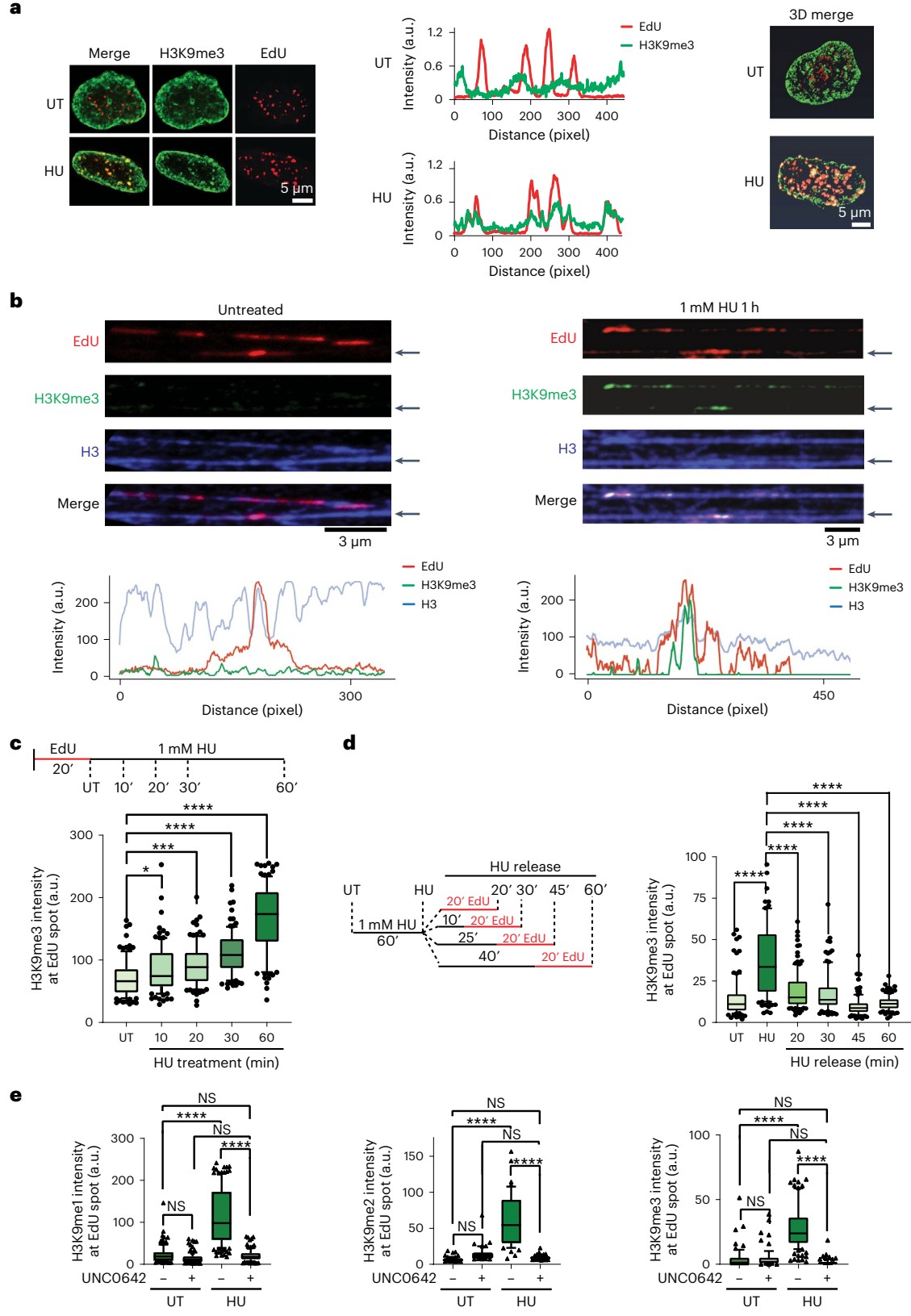

HU stress (1 mM; 1 h). We observed a gradual reduction in PA-GFP-H2A tracks area upon HU treatment in replicating cells but not in non-replicating or untreated cells. Moreover, treatment with UNC0642 before HU treatment abrogated this response in PCNA-mCherry positive cells treated with HU (Fig. 3i and Extended Data Fig. 5e). These findings are consistent with the notion that G9a-mediated H3K9me accumulation at stressed replication sites induces a compact chromatin structure in the stressed regions (Fig. 3g).

## Stalled fork-associated proteome requires heterochromatin platform

To understand how epigenetic landscape formed at replication forks in response to replication stress is critical for establishing the protein network associated with stressed replication forks, we performed isolation of proteins on nascent DNA (iPOND) coupled to stable isotope labelling with amino acids in cell culture (SILAC)-based quantitative mass spectrometry[59]. We took advantage of G9a catalytic inhibition using short treatment of UNC0642 for 2 h to investigate the direct regulation of protein homeostasis dependent upon transiently accumulated H3K9me marks. We compared protein enrichments at active replication forks as well as at stalled replication forks in the presence or absence of G9ai (Fig. 4a and Extended Data Fig. 6a). Interestingly, the enrichment of core replisome machinery such as DNA polymerases (POLD, POLA and POLE) PCNA, PCNA-interacting proteins and the RFC (1–5) complex was not changed remarkably upon G9ai (Extended Data Fig. 6b and Supplementary Table 3), while the enrichment of a set of proteins that associate with stalled replication forks was dramatically shifted (Fig. 4a and Supplementary Table 4). Among these were the fork protection factors BRCA1, BARD1, FANCD2 and RAD51, while no significant differences were observed in fork remodeller SMARCAL1, ATR-interacting proteins, canonical histones (H1–H4) or histone chaperones and nucleosome remodellers associated with replication forks such as ASF1a/b, CHD4 or the DNA replication repair MMS22L–TONSL complex. In agreement with these observations, G9ai did not affect the transient accumulation of other histone marks associated with stressed sites, such as H2AK15Ub (ref. 59) or the efficiency of incorporating new histones, H4K20me0 (ref. 60) (Extended Data Figs. 3d and 6c,d), suggesting the primary role of G9a at replication forks is to catalyse transient repressive H3K9me modification. We also observed the enrichment of proteins that do not normally associate with stalled replication forks under wild-type conditions, such as histone demethylases, RNA binding proteins and the error-prone

DNA polymerase PRIMPOL (Fig. 4a), indicating altered nascent chromatin proteome of stressed forks upon inhibition of G9a activity. We performed transcriptome analysis to test whether the changes in protein enrichments at replication forks upon G9ai could be a result of transcription deregulation. We observed very mild effects in a subset of non-DDR genes (≥1.5-fold change in expression) in conditions of either 2 h of G9ai-untreated or G9ai with HU-treated cells, whereas almost no anomalous expression was observed in either condition for a large set of DDR genes (n = 179) (ref. 59), which included both homologous recombination and non-homologous DNA end-joining DDR genes (Extended Data Fig. 6e). This suggests that the function of G9a in regulating the chromatin landscape at replication forks is unrelated to its role in transcriptional regulation in unperturbed cells. We validated our mass spectrometry data by quantifying enrichment of some of these proteins at the site of replication using PLA. In concordance with iPOND-MS, we did not observe any significant changes in PCNA or RPA (Extended Data Fig. 6f,g) levels at the replication forks upon G9ai while we observed significant reduction in RAD51 and the BARD1-BRCA1 complex associated with stalled replication forks (Fig. 4b–d). We further investigated if accumulation of H3K9me3 upon replication stress depends on fork remodelling activity: knockdown of SMARCAL1 did not significantly change H3K9me levels upon HU-induced replication stress (Fig. 4a,e), suggesting that fork remodelling is not required for H3K9me deposition by G9a.

## Chromatin compaction ensures stressed fork stability

Since fork protection proteins prevent degradation of nascent DNA by nucleases, we performed DNA fibre analysis to test whether their defective recruitment at stalled forks in the absence of H3K9me impairs fork stability. We labelled replication tracts with 5-chloro-2′-deoxyuridine (CldU) and 5-iodo-2′-deoxyuridine (IdU) followed by 3 h treatment with 4 mM HU to assess the efficiency of stalled fork protection. As previously reported[61], loss of BRCA1 resulted in stalled fork degradation, and G9ai resulted in comparable levels of nascent DNA degradation (Fig. 5a). Similarly, knockdown of SUV39h1 resulted in fork degradation upon HU treatment and was epistatic with G9ai (Figs. 3f and 5a). These data strongly suggest that the accumulation of H3K9me3 at replication forks induced by G9a and SUV39h1 upon replication stress is essential for stalled forks protection. Moreover, we noticed that, despite having a normal cell cycle (Extended Data Fig. 4h), the firing of replication origins, analysed by DNA combing to measure replication tracks labelled with CldU and IdU, was mildly

**Fig. 3 | H3K9me3, G9a and Suv39h1 accumulation at stalled replication forks is replication checkpoint dependent and results in chromatin compaction.**
**a**, Left: representative images of PLA depicting H3K9me3 presence at replication sites (H3K9me3-EdU PLA, red). Nuclei were counterstained with DAPI (blue). Right: distribution of the total intensity of all H3K9me3-EdU PLA spots per nucleus in wild-type cells (WT), G9a knockout cells (G9a−/−) and wild-type cells treated with 1 µM UNC0642 (UNC0642). Cells were labelled with EdU for 20 min and were either left untreated (UT), treated with 1 mM HU for 1 h (HU) or treated with 1 mM HU for 1 h and released from HU for 25 min and labelled with EdU for 20 min (Rel). ($n_{WT-UT}$ = 2,436, $n_{WT-HU}$ = 2,212, $n_{WT-REL}$ = 2,340, $n_{G9aKO-UT}$ = 1,038, $n_{G9aKO-HU}$ = 1,168, $n_{G9aKO-REL}$ = 1,074, $n_{UNC0642-UT}$ = 2,413, $n_{UNC0642-HU}$ = 2,328, $n_{UNC0642-REL}$ = 2,315 cells analysed). **b–d**, Same as **a** but showing the distribution of PLA spot intensity per nucleus for H3K9me1-EdU PLA ($n_{WT-UT}$ = 1,346, $n_{WT-HU}$ = 1,050, $n_{WT-REL}$ = 1,192, $n_{G9aKO-UT}$ = 1,543, $n_{G9aKO-HU}$ = 1,470, $n_{G9aKO-REL}$ = 1,630, $n_{UNC0642-UT}$ = 1,502, $n_{UNC0642-HU}$ = 1,296, $n_{UNC0642-REL}$ = 1,338 cells analysed) (**b**), H3K9me2-EdU PLA ($n_{WT-UT}$ = 1,442, $n_{WT-HU}$ = 1,431, $n_{WT-REL}$ = 1,338, $n_{G9aKO-UT}$ = 1,321, $n_{G9aKO-HU}$ = 1,381, $n_{G9aKO-REL}$ = 1,380, $n_{UNC0642-UT}$ = 1,367, $n_{UNC0642-HU}$ = 1,490, $n_{UNC0642-REL}$ = 1,411 cells analysed) (**c**) and G9a-EdU PLA ($n_{WT-UT}$ = 1,407, $n_{WT-HU}$ = 1,086, $n_{WT-REL}$ = 1,502, $n_{G9aKO-UT}$ = 1,510, $n_{G9aKO-HU}$ = 1,513, $n_{G9aKO-REL}$ = 1,510, $n_{UNC0642-UT}$ = 1,504, $n_{UNC0642-HU}$ = 1,505, $n_{UNC0642-REL}$ = 1,501 cells analysed) (**d**). **e**, Distribution of H3K9me3-EdU (left) or G9a-EdU (right) total PLA spot intensity per nucleus of wild-type cells treated (ATRi+) or not (ATRi−) with 10 µM ATR inhibitor and EdU labelled for 20 min followed by a 1 mM HU treatment for 1 h. For H3K9me3-EdU PLA:

$n_{HU-}$ = 909, $n_{HU+}$ = 931; for G9a-EdU PLA: $n_{HU-}$ = 869, $n_{HU+}$ = 1,080 cells analysed. **f**, Same as **a** but showing the distribution of H3K9me3-EdU total PLA spot intensity per nucleus for the indicated conditions ($n_{ctl-UT}$ = 1,509, $n_{ctl-HU}$ = 1,509, $n_{ctl-REL}$ = 1,506, $n_{UNC0642-UT}$ = 2,003, $n_{UNC0642-HU}$ = 1,529, $n_{UNC0642-REL}$ = 1,543, $n_{siSUV39h1-UT}$ = 1,514, $n_{siSUV39h1-HU}$ = 1,502, $n_{siSUV39h1-REL}$ = 1,500, $n_{UNC0642+siSUV39h1-UT}$ = 1,502, $n_{UNC0642+siSUV39h1-HU}$ = 1,523, $n_{UNC0642+siSUV39h1-REL}$ = 1,507, cells analysed) (note that, for **a–f**, blue dashed indicates mean of the distribution, ****P ≤ 0.0001, ***P ≤ 0.001, **P ≤ 0.01, *P ≤ 0.05, NS, non-significant, one-way analysis of variance Kruskal–Wallis test followed by Dunn's test is used for all statistical analysis). **g**, Model summarizing G9a and SUV39h1 role at stalled replication forks. Upon replication stress, checkpoint-regulated G9a activity at stressed replication forks results in transient accumulation of H3K9me1/2 allowing SUV39h1 to catalyse H3K9me3 modification. Further accumulating HDAC1 resulted in the loss of H4K16ac. Figure created with biorender.com. **h**, Representative images of the changes over time of a stripe of photo-activated GFP-H2A for the indicated conditions. This experiment was reproduced independently three times with similar results. **i**, Mean photo-activated GFP-H2A area over time relative to the area at T = 0 min in percentage ± standard deviation. In PCNA negative (black) and positive (red) for untreated cell: WT-UT (left), cells undergoing replication stress: WT + HU (middle) and cells undergoing replication stress in the absence of G9a activity (right). Unpaired two-sided t-test, ****P ≤ 0.0001, **P ≤ 0.01. For experimental design, see Extended Data Fig. 5e. n = 3 independent experiments. Source numerical data are available in Source Data.

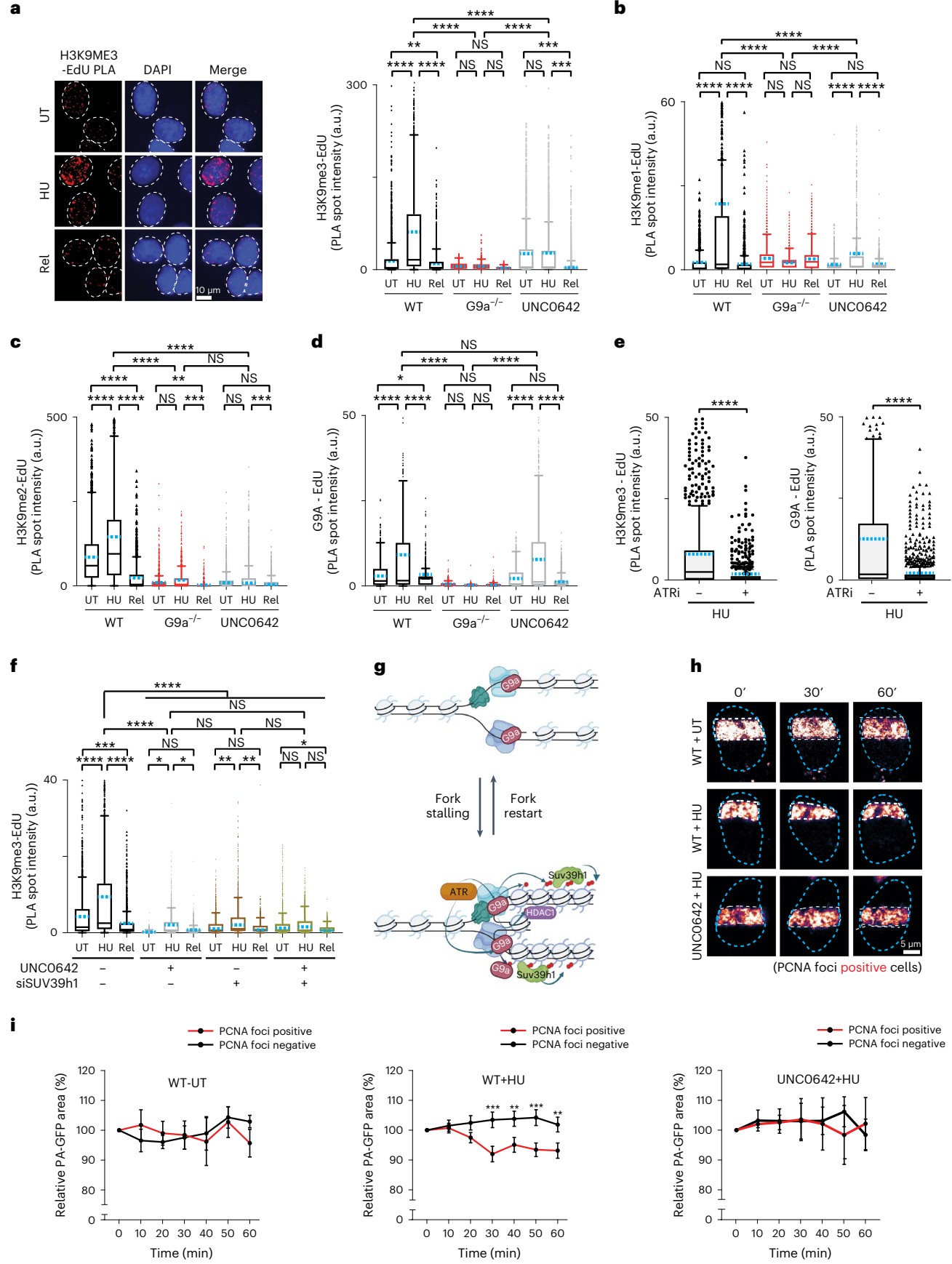

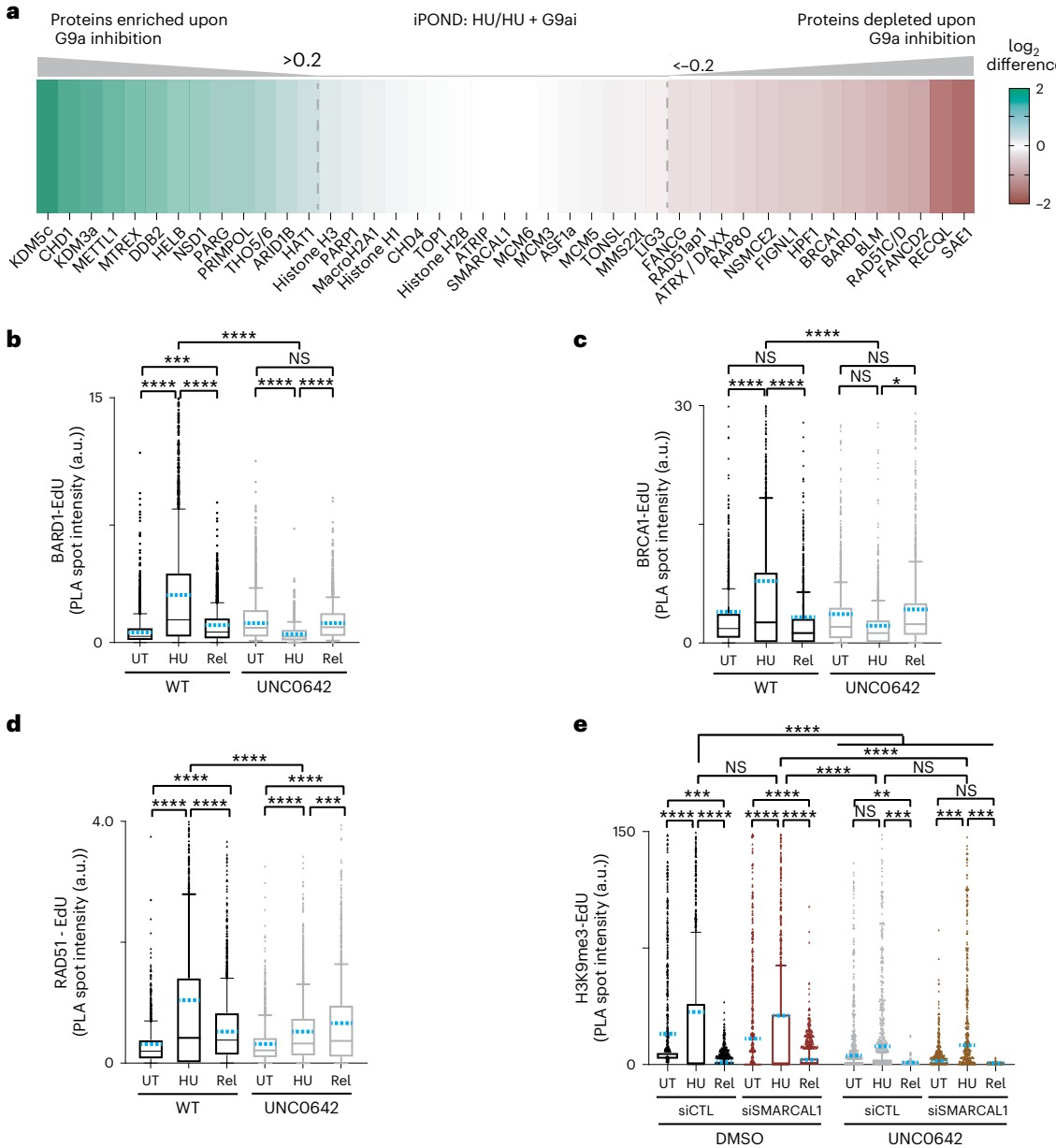

**Fig. 4 | Loss of transiently accumulated H3K9me drastically alters the chromatin landscape of stalled forks. a**, Colour-coded diagram showing a selection of proteins enriched (shades of green) or depleted (shades of red) at stalled replication fork in the absence of G9a activity. Proteins were considered enriched when the $\log_2$ ratio of HU + G9a inhibition/HU was greater than 0.2 and depleted when the $\log_2$ ratio of HU + G9a inhibition/HU was lower than −0.2. **b**–**d**, Dynamics of BARD1 (**b**), BRCA1 (**c**) and RAD51 (**d**), at replication sites in the presence (WT) and in the absence of G9a activity (UNC0642). The plots are showing the distribution of PLA spots intensity per nucleus in either unperturbed (UT), stalled (HU) and restarted (Rel) replication. BARD1-EdU ($n_{WT-UT}$ = 1,648, $n_{WT-HU}$ = 2,008, $n_{WT-REL}$ = 2,022, $n_{UNC0642-UT}$ = 2,004, $n_{UNC0642-HU}$ = 2,008, $n_{UNC0642-REL}$ = 2,002 cells analysed) (**b**), BRCA1-EdU ($n_{WT-UT}$ = 1,502, $n_{WT-HU}$ = 1,508, $n_{WT-REL}$ = 1,510, $n_{UNC0642-UT}$ = 1,511, $n_{UNC0642-HU}$ = 1,521, $n_{UNC0642-REL}$ = 1,505 cells analysed) (**c**) or RAD51-EdU ($n_{WT-UT}$ = 1,511, $n_{WT-HU}$ = 1,510, $n_{WT-REL}$ = 1,503, $n_{UNC0642-UT}$ = 1,521, $n_{UNC0642-HU}$ = 1,502, $n_{UNC0642-REL}$ = 1,505 cells analysed) (**d**). Cells were labelled with EdU for 20 min and were either left untreated (UT) or treated with 1 mM HU for 1 h (HU) or treated with 1 mM HU for 1 h and released from HU for 25 min and labelled with EdU for 20 min (Rel). **e**, Dynamics of H3K9me3 at replication sites in the presence (DMSO) and in the absence of G9a activity (UNC0642) as well as in the presence (siCTL) or absence of SMARCAL1 (siSMARCAL1). Plots showing distribution of H3K9me3-EdU PLA spots intensity per nucleus in either unperturbed (UT), stalled (HU) and restarted (Rel) replication. Cells were labelled with EdU for 20 min and were either left untreated (UT) or treated with 1 mM HU for 1 h (HU) or treated with 1 mM HU for 1 h and released from HU for 25 min and labelled with EdU for 20 min (Rel). It is interesting to note that transient accumulation of H3K9me3 at replication sites upon replication stress is independent of fork reversal activity. $n_{siCTL-UT}$ = 1,338, $n_{siCTL-HU}$ = 1,337, $n_{siCTL-REL}$ = 1,339, $n_{siCTL+UNC0642-UT}$ = 1,341, $n_{siCTL+UNC0642-HU}$ = 1,138, $n_{siCTL+UNC0642-REL}$ = 1,343, $n_{siSMARCAL1-UT}$ = 1,339, $n_{siSMARCAL1-HU}$ = 1,342, $n_{siSMARCAL1-REL}$ = 747, $n_{siSMARCAL1+UNC0642-UT}$ = 1,341, $n_{siSMARCAL1+UNC0642-HU}$ = 1,340, $n_{siSMARCAL1+UNC0642-REL}$ = 1,338 cells analysed; blue dashed line represents the mean of the distribution, ****$P \le 0.0001$, ***$P \le 0.001$, **$P \le 0.01$, *$P \le 0.05$, NS, non-significant, Kruskal–Wallis test followed by Dunn's test is used for all statistical analysis. Source numerical data are available in Source Data.

dysregulated upon G9ai (Extended Data Fig. 7a–c), which resulted in slower fork progression rates in unperturbed conditions (Extended Data Fig. 7d)[62,63]. This could be rescued by treating cells with the Cdk inhibitor, Roscovitine (Extended Data Fig. 7e). However, Roscovitine treatment could not prevent fork degradation observed upon G9ai (Extended Data Fig. 7f), suggesting that the fork protection role of G9a by establishing chromatin compaction is independent of DNA replication origin regulation.

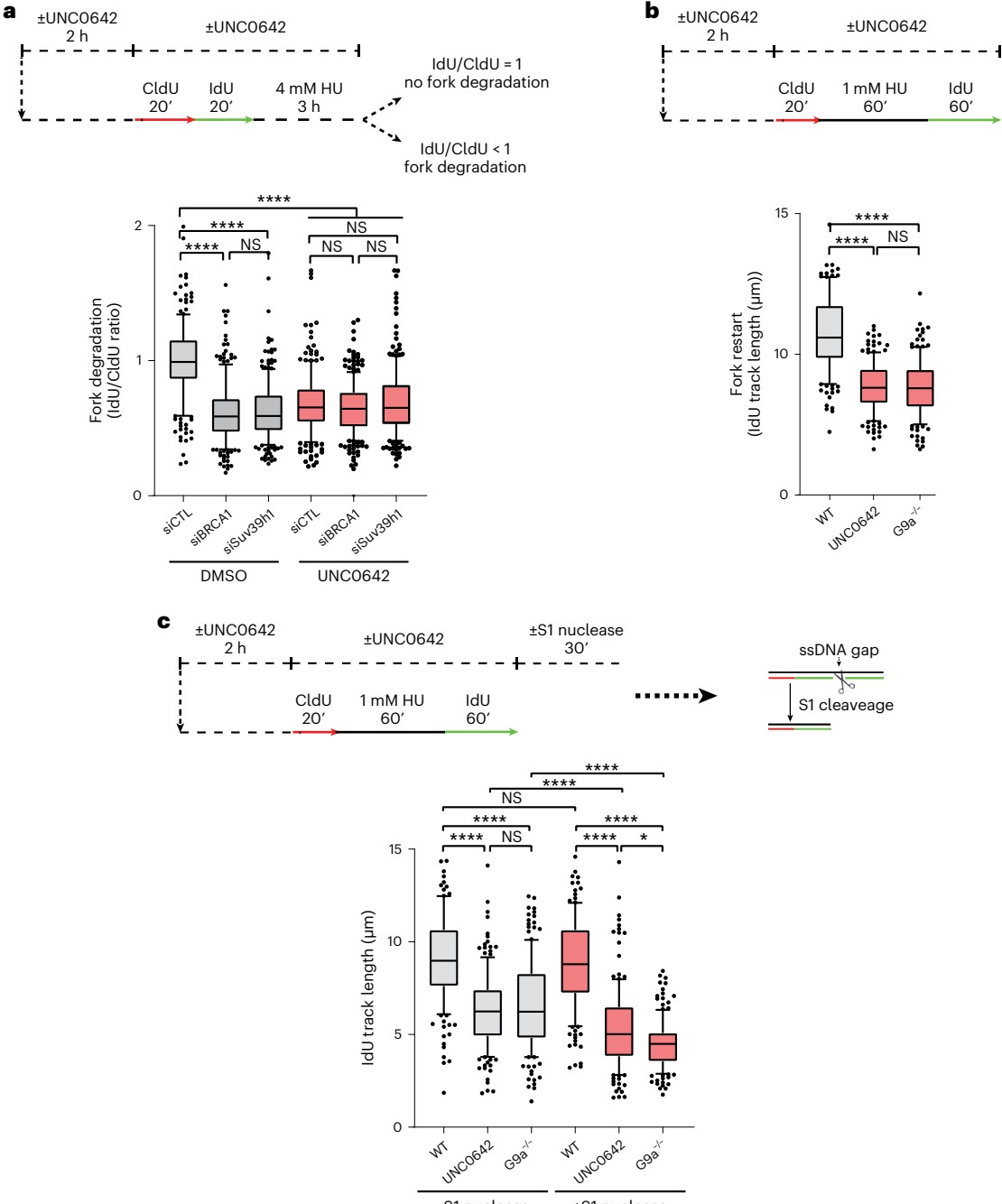

**Fig. 5 | Loss of transient H3K9me3 accumulation at stalled forks impairs replication fork stability and causes genome instability. a**, Top: schematic of replication fork degradation assay with CldU and IdU labelling. Bottom: ratio of IdU to CldU tract length was plotted for the indicated conditions. ($n_{siCTL}$ = 207, $n_{siBRCA1}$ = 214, $n_{siSUV39h1}$ = 213, $n_{siCTL+UNC0642}$ = 207, $n_{siBRCA1+UNC0642}$ = 239, $n_{siSUV39h1+UNC0642}$ = 203 replication tracks analysed; ****$P \leq 0.0001$, NS, non-significant, Kruskal–Wallis test followed by Dunn's test). **b**, Top: schematic of the fork restart assay. Bottom: the IdU track length (μm) was plotted to show fork

restart ($n_{WT}$ = 150, $n_{UNC0642}$ = 150, $n_{G9aKO}$ = 150; ****$P \leq 0.0001$, NS, non-significant, Kruskal–Wallis test followed by Dunn's test). **c**, Top: schematics of ssDNA gap accumulation. Bottom: the IdU track length (μm) was plotted to assess the accumulation of ssDNA behind the forks for the indicated conditions ($n_{WT\_S1-}$ = 151, $n_{UNC0642\_S1-}$ = 157, $n_{G9aKO\_S1-}$ = 151, $n_{WT\_S1+}$ = 153, $n_{UNC0642\_S1+}$ = 153, $n_{G9aKO\_S1+}$ = 153 replication tracks analysed; ****$P \leq 0.0001$, ***$P \leq 0.001$, **$P \leq 0.01$, NS, non-significant, Kruskal–Wallis test followed by Dunn's test). Source numerical data are available in Source Data.

We further tested if H3K9me3 establishment at stalled forks is also critical for proper fork restart once replication stress is alleviated. We used a DNA fibre assay to assess the efficiency of fork restart. Cells treated with UNC0642 or G9a knockout cells restarted replication more slowly (shorter IdU tracks) than untreated cells after release from HU (Fig. 5b), suggesting that G9a activity is required for timely restart of replication. Interestingly, when cells were treated with single-stranded

DNA (ssDNA)-specific S1 nuclease to determine whether IdU tracks after release from HU contained ssDNA gaps[64], we observed a significant shortening of the replication tracks upon G9ai. These data suggest that restarted forks accumulate ssDNA behind the replication forks in absence of G9a activity (Fig. 5c). Such accumulation of ssDNA gaps generated upon replication stress contributes to genome instability[65,66]. Consistently, cells lacking G9a were highly sensitive to replication

stress-inducing, DNA-damaging drugs olaparib (poly(ADP-ribose) polymerase inhibitor, PARPi) and cisplatin (Extended Data Fig. 7g,h).

### JMJD1A/KDM3A disassembles heterochromatin for proper fork restart

The primase polymerase PRIMPOL can facilitate restart of replication but leaves ssDNA gaps behind the fork. Interestingly, in our quantitative iPOND-MS dataset, PRIMPOL was enriched at stressed replication forks upon G9ai (Fig. 4a). To test if PRIMPOL was responsible for ssDNA gaps accumulated behind the forks in G9ai cells, we depleted PRIMPOL using small interfering RNA (siRNA). Interestingly, upon depletion of PRIMPOL, G9ai cells showed significantly fewer ssDNA gaps accumulated behind the forks (Fig. 6a), but the defective fork restart observed in G9ai cells was enhanced (Fig. 6b), suggesting that PRIMPOL-mediated re-priming is required for DNA synthesis in G9ai cells even if it is at the expense of genome stability. Upon PRIMPOL overexpression[67], ssDNA gaps accumulated (+S1 nuclease condition) upon HU treatment, even more so in combination with G9ai, suggesting that loss of H3K9me3 at nascent DNA exposes forks to be accessed by PRIMPOL (Extended Data Fig. 7i). Together these findings suggest that de novo heterochromatin formation at nascent DNA denies access to PRIMPOL to maintain genome integrity (Figs. 4a, 5c and 6a and Extended Data Fig. 7i).

We noted that Jumonji domain-containing protein 1A (JMJD1A)/Lysine (K)-Specific Demethylase 3A (KDM3A) was enriched at stalled replication forks upon G9ai (Fig. 4a). We wondered if this enrichment could accelerate the demethylation of H3K9, thus leaving forks unprotected. Transient depletion of JMJD1A/KDM3A in G9ai cells rescued ssDNA gap accumulation, similar to siPRIMPOL, suggesting that loss of H3K9me3 assembly at replication forks provides access to PRIMPOL. Furthermore, we observed a significant delay in fork restart ability upon loss of JMJD1A/KDM3A in untreated cells and upon G9ai, suggesting that KDM3A distorts heterochromatin upon release of replication stress to ensure canonical restart of forks (Fig. 6b). Transient depletion of KDM3A fully restored fork protection in G9ai cells (Fig. 6c), suggesting that untimely action of KDM3A allowed PRIMPOL and DNA nucleases to access de-heterochromatinized forks in G9ai cells. Consistent with the rescue of both fork degradation and ssDNA gap accumulation, we observed restoration of H3K9me3 at stalled forks in cells lacking both KDM3A and G9a activity, probably by intact activity of SUV39h1 (Fig. 6d and Extended Data Fig. 5b). Even though fork restart is delayed in the absence of KDM3A and G9a activity, it allows the restoration of a canonical fork restart pathway that may restore genome stability in G9ai cells. We performed clonogenic assays to test cellular sensitivity of cells lacking G9a. We observed a complete rescue of cellular sensitivity towards cisplatin and PARPi initially observed in the absence of G9a, suggesting restoration of genome stability (Fig. 6e,f). This shows that

dynamic involvement of epigenetic 'writers' and 'erasers' balances the amount of heterochromatin marks to ensure genome integrity upon replication stress.

G9a is overexpressed in various cancers and promotes metastasis[68,69]. Higher levels of H3K9 methylation as well as higher G9a/GLP levels correlate with poor prognosis in patients with high-grade serous ovarian cancer[48,70,71]. Independently, our analysis on G9a/GLP levels in patients with ovarian cancer indicated a correlation with poor response to chemotherapy as well as poor survival (Fig. 7a,b). The importance of the dynamic changes in the chromatin landscape for the stability of stalled replication fork could explain why accumulation of H3K9me3 and hypoacetylation is observed in many cancers.

## Discussion

Our study uncovers a previously unidentified role of H3K9me1/me2/me3 'writers' and 'erasers' in dynamically remodelling chromatin at replication forks to maintain fork stability upon replication stress. Using quantitative proteomics, genomics and high-resolution single-molecule chromatin visualization, we have revealed a dynamic checkpoint regulated de novo heterochromatin assembly mechanism at replication forks catalysed by G9a/EHMT2 in concert with SUV39h1. Disassembly of heterochromatin is rapidly catalysed by the H3K9-demethylase JMJD1A/KDM3A at restarted forks upon release from replication stress. Our data suggest that the compaction of stressed replicating regions is required to establish a chromatin environment associated with fork protection, while its timely disassembly is required to allow canonical fork restart, preventing ssDNA gap accumulation. Both these processes are tightly regulated to maintain genetic as well as epigenetic stability in cells undergoing replication stress.

First, using chronic replication stress conditions where cells were cultured in low dose of HU for several days, we observed genome-wide accumulation of H3K9me3 chromatin marks, as previously reported[21,35,72]. Interestingly, cells submitted to acute replication stress mediated by high dose of HU for a short amount of time (1–2 h) showed a similar transient accumulation of H3K9me3 at stressed replication forks. H3K9me1/me2 levels increased at replication stress sites under acute stress conditions, unlike chronic stress condition, suggesting that lower H3K9 modifications would have been converted to H3K9me3 upon prolonged replication stress. This assumption is supported by the genome-wide spread of H3K9me3 observed during prolonged replicative stress. This supports a stepwise mechanism[31] for establishment of H3K9me3 upon replication stress. There may be a small percentage of dormant origins showing EdU labelling that may already lie in an existing heterochromatin region and show H3K9me3 signal. However, throughout our ChromStretch analysis, we avoided the bias arising from constitutive (approximately hundreds of

---

**Fig. 6 | Loss of KDM3A rescues fork degradation, ssDNA gap accumulation and drug sensitivity of cells lacking G9a activity. a**, Top: schematics of ssDNA gap accumulation. Bottom: IdU track length (µm) distribution for the indicated conditions. $n = 100$ replication forks analysed per condition ($n_{siCTL\_S1-} = 206$, $n_{siPRIMPOL\_S1-} = 91$, $n_{siKDM3\_S1-} = 206$, $n_{siCTL+UNC0642\_S1-} = 201$, $n_{siPRIMPOL+UNC0642\_S1-} = 202$, $n_{siKDM3+UNC0642\_S1-} = 201$, $n_{siCTL\_S1+} = 206$, $n_{siPRIMPOL\_S1+} = 206$, $n_{siKDM3\_S1+} = 209$, $n_{siCTL+UNC0642\_S1+} = 209$, $n_{siPRIMPOL+UNC0642\_S1+} = 203$, $n_{siKDM3+UNC0642\_S1+} = 205$, replication tracks analysed; ****$P \le 0.0001$, ***$P \le 0.001$, **$P \le 0.01$, NS, non-significant, Kruskal–Wallis test followed by Dunn's test). **b**, Top: schematics of the Fork restart assay. Bottom: IdU track length (µm) distribution ($n_{siCTL} = 650$, $n_{siPRIMPOL} = 376$, $n_{siKDM3} = 316$, $n_{siCTL+UNC0642} = 502$, $n_{siPRIMPOL+UNC0642} = 369$, $n_{siKDM3+UNC0642} = 302$ replication tracks analysed; ****$P \le 0.0001$, ***$P \le 0.001$, **$P \le 0.01$, *$P \le 0.05$, NS, non-significant, Kruskal–Wallis test followed by Dunn's test). **c**, Fork degradation performed as Fig. 5a. Ratio of IdU to CldU tract length was plotted for the indicated conditions ($n_{siCTL} = 161$, $n_{siBRCA1} = 164$, $n_{siKDM3} = 161$, $n_{siCTL+UNC0642} = 162$, $n_{siBRCA1+UNC0642} = 163$, $n_{siKDM3+UNC0642} = 176$ replication tracks analysed; ****$P \le 0.0001$, NS, non-significant, Kruskal–Wallis test followed by Dunn's test). **d**, Dynamics of H3K9me3 at replication sites in the presence (DMSO) or in the absence of G9a

activity (UNC0642) and in the presence (siCTL) or absence of KDM3A (siKDM3). Distribution of H3K9me3-EdU PLA spots intensity per nucleus upon unperturbed (UT), stressed (HU) and restarted (Rel) replication. Cells were labelled with EdU for 20 min and were either left untreated (UT) or treated with 1 mM HU for 1 h or treated with 1 mM HU for 1 h and released for 25 min before labelling with EdU for 20 min (Rel). Blue dashed indicates mean of the distribution, $n_{siCTL-UT} = 1,509$, $n_{siCTL-HU} = 1,509$, $n_{siCTL-REL} = 1,506$, $n_{siCTL+UNC0642-UT} = 2,003$, $n_{siCTL+UNC0642-HU} = 1,529$, $n_{siCTL+UNC0642-REL} = 1,543$, $n_{siKDM3-UT} = 1,516$, $n_{siKDM3-HU} = 1,504$, $n_{siKDM3-REL} = 1,524$, $n_{siKDM3+UNC0642-UT} = 1,505$, $n_{siKDM3+UNC0642-HU} = 1,536$, $n_{siKDM3+UNC0642-REL} = 1,514$ cells analysed; ****$P \le 0.0001$, ***$P \le 0.001$, **$P \le 0.01$, *$P \le 0.05$, NS, non-significant, Kruskal–Wallis test followed by Dunn's test is used for all statistical analysis. **e,f**, Colony survival assay. Mean survival in wild type (WT) and cells lacking G9a (G9a−/−), in the presence (siCTL) or absence of KDM3A (siKDM3) and treated with different concentrations of olaparib (PARPi, **e**) or cisplatin (**f**). Data are normalized to the 0 dose of the corresponding condition. Error bars represent ± standard deviation ($n = 3$ independent experiment) (****$P \le 0.0001$, **$P \le 0.01$, NS, non-significant, ordinary two-way analysis of variance was used for multiple comparisons). Source numerical data are available in Source Data.

kilobases) long heterochromatic regions to show the effects of replication stress in de novo heterochromatin assembly at EdU-labelled replicating sites in euchromatic regions. These regions specifically show that H3K9me3 upon HU treatment generally overlaps with the EdU signal and does not extend beyond it (Fig. 2b and Extended Data Fig. 3a,b). This mechanism is distinct from the role of histone chaperone ATRX/DAXX in maintaining H3.3-mediated heterochromatin assembly at G4 structures, independent of checkpoint activation[73]. ATRX remains associated with G4-repeats-containing regions to maintain them in condensed state, whereas checkpoint-activated G9a catalyses de novo H3K9me1/me2/me3 at a majority of stressed

forks (80–85% stressed forks), suggesting a general fork protection mechanism. However, we do not rule out the possibility that ATRX/DAXX and G9a act in concert to prevent replication stress, especially as DAXX deposits H3.3 carrying H3K9me3 (ref. 74). Moreover, ATRX and DAXX were slightly depleted from stalled replication forks in absence of G9a activity, suggesting an interplay between these two pathways. In parallel with H3K9me3 accumulation, we observed transient H3K9me3-dependent accumulation of HDAC1 resulting in local histone deacetylation at stressed replication sites. Consistent with the well-established role of deacetylated nucleosomes and H3K9me3 in reduced nucleosome turnover and increased chromatin compaction,

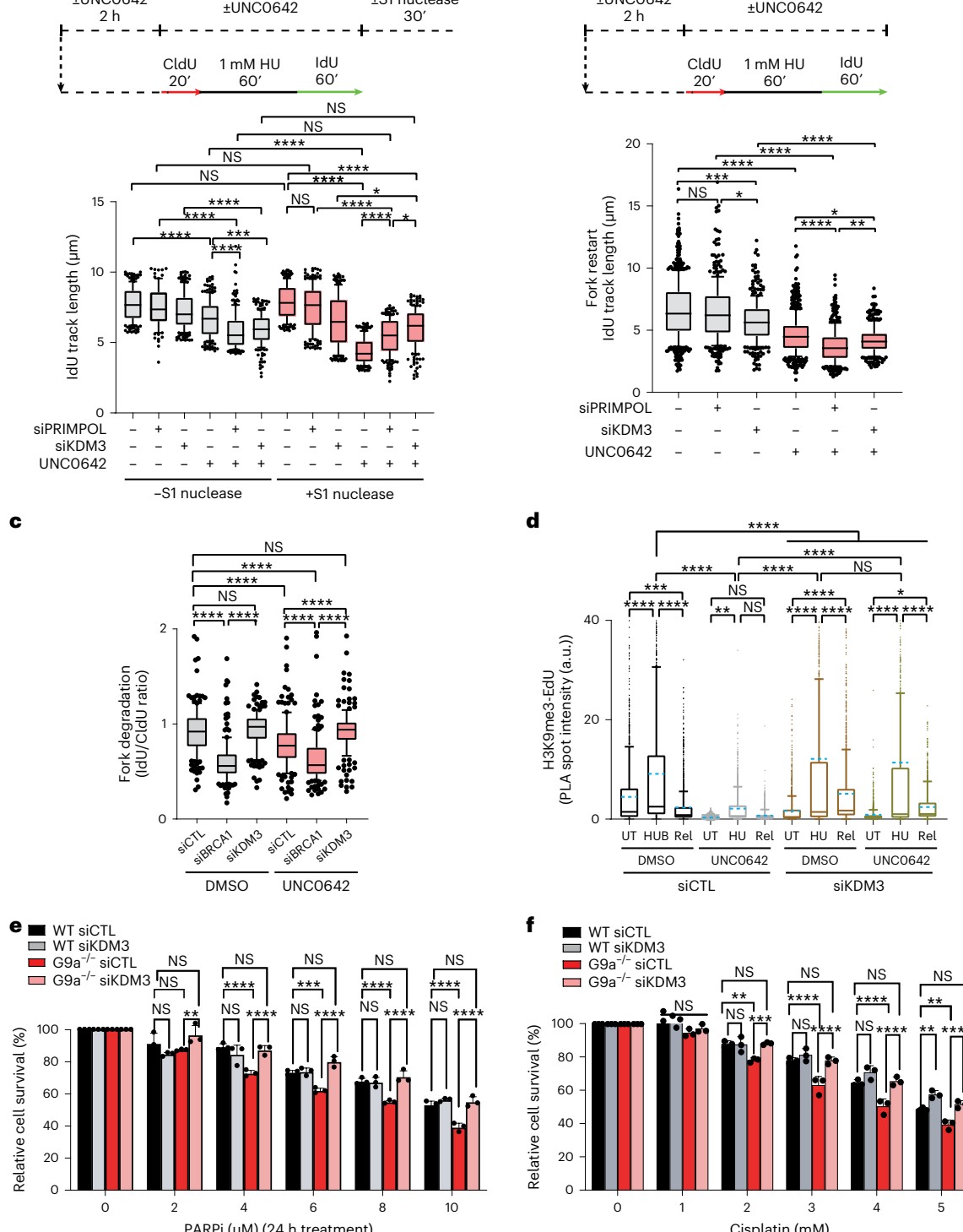

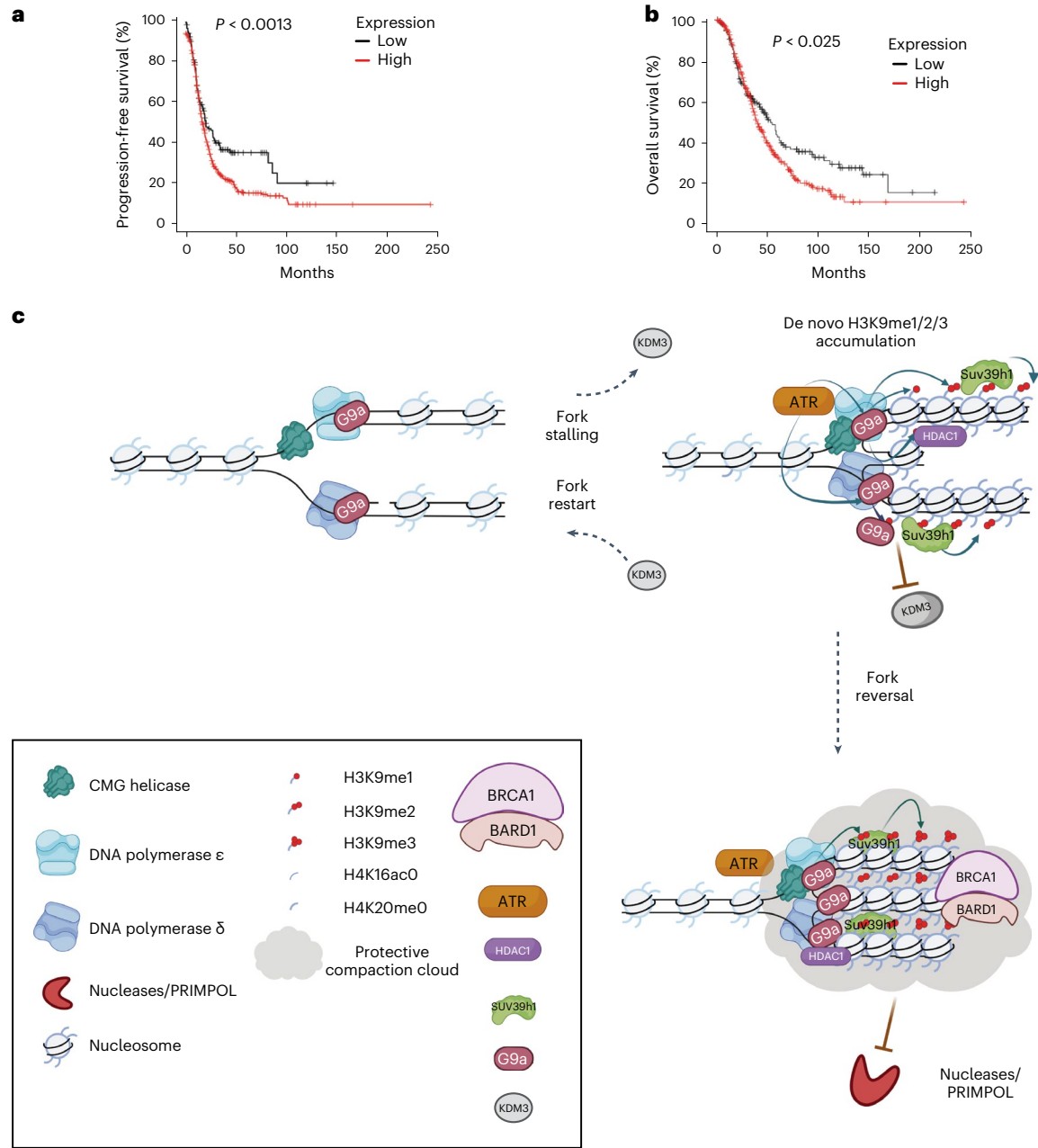

**Fig. 7 | G9a overexpression correlates with poor prognosis in ovarian cancer, highlighting the importance of a timely accumulation of de novo H3K9me1/2/3 marks and its disassembly catalysed by 'writers' and 'erasers' at stressed replication forks to maintain fork stability. a,b,** Combined mean expression was calculated to distinguish TCGA patients with ovarian cancer with low or high GLP/G9a expression[71,95,96]. Kaplan–Meier curves were generated against progression-free survival (**a**) and overall patient survival (**b**) (n = 614 patients). P values were calculated with the use of a two-sided log-rank test. **c,** G9a/EHMT2 associated with replication forks is activated by canonical DNA replication checkpoint pathway to catalyse H3K9me1/me2 at replication forks upon replication stress. Activated G9a generates a platform of H3K9me1/me2/me3 in concert with Suv39h1 at the site of stressed replication forks, which subsequently recruits histone deacetylase, HDAC1 to deacetylate the nucleosomes. Such closed chromatin conformation may create a protective compaction bubble that protects replication forks by (1) promoting efficient recruitment of fork protection factors, BARD1·BRCA1; and (2) such a conformation may also prevent the access to DNA nucleases and other detrimental factors, such as PRIMPOL that can lead to accumulation of ssDNA gaps behind the replication forks. Furthermore, synergistic activity of G9a and Suv39h1 further prevents the substrate, H3K9me1/me2 nucleosomes, availability to H3K9-demethylase, JMJD1A/KDM3A, timely assembly of which facilitates the disassembly of heterochromatin to promote their fork restart. Figure created with biorender.com. Source numerical data are available in Source Data.

our data suggest chromatin condensation exclusively in replicating cells exposed to HU (Figs. 2b and 3h,i and Extended Data Fig. 3a–d). The role of HDACs in maintaining a closed chromatin conformation upon replication stress has been described in fission yeast as the 'chromsfork pathway'[75]. However, this HDAC-dependent pathway is independent of checkpoint regulation, unlike the mechanism identified in this study where G9a enrichment as well as H3K9me3 accumulation at stressed replication forks are regulated by checkpoint activation. These studies together argue that chromatin compaction upon replication stress are conserved protective responses.

Although our live-imaging assays did not provide resolution to compare chromatin accessibility between replicating versus non-replicating region within a cell, these data along with observations from ChromStretch fibres showing higher nucleosome density at stressed replication sites indicate a change in chromatin compaction in response to HU stress. Adapting deep-sequencing-based high-resolution assays to measure nucleosomal occupancy or chromatin accessibility[76–78] at stressed forks could help advance our understanding of fork chromatin structure and protection. In parallel with H3K9me3 accumulation, our comprehensive profiling of PTMs also revealed induction of H3K36me2 in a replication stress-dependent manner. H3K36me2 has been implicated in DNA repair through non-homologous DNA end-joining[79,80] as well as linked to DNA replication checkpoint activation in fission yeast[81]. The increased levels of H3K36me2 and its writers have been reported in various cancers[82,83]. The higher enrichment of 'writer' of H3K36me2, NSD1, upon G9ai provides an exciting avenue to follow for future studies.

Replication checkpoint-activated G9a initiates stepwise accumulation of H3K9me1/me2/me3 in concord with SUV39h1. Our iPOND data suggest that an important part of G9a function at stalled forks is to prevent the untimely action of JMJD1A/KDM3A to prevent precarious restart of replication fork. The synergistic action of these HMTs may accelerate the catalytic reactions, leading to chromatin compaction during replication stress, as suggested by the significant accumulation of H3K9me2/me3 levels within 20–30 min of HU treatment (Fig. 2c and Extended Data Fig. 4b). The fast accumulation of heterochromatin may ensure that nascent DNA at stressed forks is protected from the action of nucleases, primases or the transcription machinery, to maintain genome stability.

We speculate that synergistic action of histone modifiers, G9a and SUV39h1 at stressed forks would also prevent demethylases such as KDM3A to gain access to the common substrate for their binding. KDM3A seems to play a role in the timely restart of replication forks, suggesting that, upon release and de-activation of the replication checkpoint, the balance is shifted towards KDM3A accessing stalled forks and disassembling heterochromatin by demethylating H3K9me marks. However, in the absence of G9a activity, untimely demethylation by KDM3A provides access to nucleases, causing degradation of forks, or to PRIMPOL to promote DNA synthesis at the expense of genome stability. In the absence of G9a activity and of the opposing activity of KDM3A, full access is given to SUV39h1 to form heterochromatin at stalled forks. However, upon HU release, forks show significant delay in restart, although this happens through canonical pathways as ssDNA gaps no longer accumulate. This suggest that, in the absence of KDM3A, it takes longer to dissolve the heterochromatin structures upon release from replication stress, yet normal restart can take place. Whether this fine-tuned interplay between chromatin modifiers requires additional regulation or synergistic action of multiple demethylases of JMJD1/2 family members[84,85] remains to be investigated.

How G9a-dependent heterochromatin ensures selective entry of fork protection proteins, while restricting access to nucleases/PRIMPOL, remains to be understood. BARD1 is a reader of H2AK13/15Ub and H4K20me0 marks, which facilitate recruitment of BARD1–BRCA1 complex at DNA-damaged sites[60,86]. These marks remain intact upon G9ai, yet we observed defective BARD1–BRCA1 complex enrichment at stalled forks. Compact chromatin conformation established by hypoacetylated H3K9me3 nucleosomes might bring the nucleosomes containing epigenetic marks, H2AK13/15Ub and H4K20me0, spatially closer to facilitate BARD1-BRCA1 recruitment at the stressed forks. Alternatively, previous studies implicate a direct binding of BARD1-BRCA1 complex with H3K9me3-modified nucleosomes[87,88]. These findings must be further investigated in the context of stalled replication forks. Importantly, the synthetic lethality of BRCA1/BARD1 with loss of H3K9me2 in *Caenorhabditis elegans*[89], together with our findings, raises intriguing possibilities for therapeutic treatment of BRCA1-mutated cancer.

Altogether, our results show that the chromatin environment is dramatically remodelled upon both persistent and acute replication stress by accumulation of H3K9me3. We elucidated the detailed molecular mechanism of dynamic assembly and disassembly of heterochromatin at stressed replication forks. Similar chromatin dynamics may occur in cancer cells that proliferate under persistent endogenous replication stress due to oncogene activation. Our findings may provide an explanation to the increased enrichment of heterochromatin observed in various cancers[21,90–94] that correlates with the poor response to chemotherapy, probably due to stabilized condensed replication forks. A combination therapy targeting the proteins mediating these epigenetic aberrations, such as G9a/GLP or SUV39h1/h2, may be worth exploring for its potential to reduce resistance to chemotherapy and cancer relapse risk.

## Online content

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

[1]Department of Molecular Genetics, Erasmus University Medical Center, Erasmus MC Cancer Institute, Rotterdam, the Netherlands. [2]Oncode Institute, Erasmus University Medical Center, Erasmus MC Cancer Institute, Rotterdam, the Netherlands. [3]Novo Nordisk Foundation Center for Protein Research (CPR), Faculty of Health and Medical Sciences, University of Copenhagen, Copenhagen, Denmark. [4]Biotech Research and Innovation Centre (BRIC), Faculty of Health and Medical Sciences, University of Copenhagen, Copenhagen, Denmark. [5]Department of Biochemistry & Molecular Biology, VILLUM

Centre for Bioanalytical Sciences and Centre for Epigenetics, University of Southern Denmark, Odense, Denmark. [6]Department of Pathology, Erasmus Optical Imaging Centre, Erasmus Medical Center, Rotterdam, the Netherlands. [7]Cancer Science Institute of Singapore, Yong Loo Lin School of Medicine, National University of Singapore (NUS), Singapore, Singapore. [8]Department of Molecular Genetics, King's College London, London, UK. [9]Laboratory of Biochemistry and Molecular Biology, National Cancer Institute, National Institutes of Health, Bethesda, MD, USA. [10]Department of Cell and Molecular Biology, Manipal School of Life Sciences, Manipal Academy of Higher Education, Manipal, India. [11]Proteomics Center and Department of Biochemistry, Erasmus University Medical Centre, Rotterdam, the Netherlands. [12]Present address: Department of Biochemistry, Albert Einstein College of Medicine, Bronx, NY, USA. [13]Present address: Institute of Functional Epigenetics (IFE), Helmholtz Zentrum Munchen, Neuherberg, Germany. [14]These authors contributed equally: Vincent Gaggioli, Calvin S. Y. Lo. ✉e-mail: n.taneja@erasmusmc.nl

## Methods

### Cell line sources

MRC5 sv40 immortalized human fibroblast and mouse embryonic stem cells (mESCs) were generated in Nitika Taneja's lab[59].

Stable TIG-3 human fibroblast was generated in Anja Groth's lab[31].

### Cell culture

MRC5 human fibroblasts were cultured in a 1:1 ratio of Dulbecco's modified Eagle medium and Ham's F10 (Invitrogen) supplemented with 10% foetal calf serum (Biowest) and 1% penicillin–streptomycin (Sigma-Aldrich) at 37 °C and 5% $CO_2$ in a humidified incubator.

TIG3 cells were grown in Dulbecco's modified Eagle medium containing 10% FBS and 1% penicillin–streptomycin supplemented with MEM non-essential amino acid mix. Quiescent cells were obtained by contact inhibition. SA-β-galactosidase assay was performed using Senescence β-Galactosidase Staining Kit from Cell Signaling, following manufacturer instructions.

mESCs were maintained in 2i medium deficient in lysine, arginine and L-glutamine (PAA) at 37 °C and 5% $CO_2$ in a humidified incubator. For SILAC labelling, cells were grown in a medium containing 73 µg ml$^{-1}$ light [$^{12}C_6$]-lysine and 42 µg ml$^{-1}$ [$^{12}C_6$, $^{14}N_4$]-arginine (Sigma-Aldrich) or similar concentrations of heavy [$^{13}C_6$]-lysine and [$^{13}C_6$, $^{15}N_4$]-arginine (Cambridge Isotope Laboratories).

### Cell line generation

Plasmid transfections for MRC5 cell line were performed using X-tremeGENE 9 DNA transfection agent (Roche) according to the manufacturer's protocol. To generate MRC5 G9a$^{-/-}$ cells, MRC5 WT cells were transfected with pLentiCRISPR-V2 plasmid (addgene #52961) containing a guide RNA sequence targeting exon 1 of G9a, followed by puromycin selection (1 µg ml$^{-1}$).

### Transient overexpression

PRIMPOL, was transiently overexpressed in MRC-5 cell upon transfection of pcDNA3.1_nV5-DEST-WT-PRIMPOL (ref. [67]) using X-treme Gene 9 DNA transfection reagent (Roche) and experiments were performed 48 h after transfection. Transfection efficiency was checked by immunofluorescence.

### Drugs and chemicals

TIG3 cells were treated with 600 µM HU. For recovery, we washed out HU and added fresh medium.

UNC0642 (MedChemExpress) was systematically added at a concentration of 1 µM 2 h before the beginning of the experiment.

Roscovitine (Sigma-Aldrich) was added at a concentration of 10 µM for 4 h before the beginning of the experiment.

### siRNA

siRNA smart pool for the indicated gene were purchase from Dharmacon and transfection were done with lipofectamine RNAiMAX (ThermoFisher) according to the manufacturer's protocol for two consecutive days. Knockdown efficiency was checked by immunoblot.

### Protein extraction and cell fractionation

For whole cell extracts, after lysis with RIPA buffer supplemented with protease inhibitor (Roche), samples were mixed with 2× Laemmli sample buffer (Supelco) and heated at 95 °C for 5 min.

For total soluble extracts, cells were washed twice with ice-cold phosphate-buffered saline (PBS) and soluble proteins extracted by incubation for 30 min with NP-40 buffer (50 mM Tris, pH 7.8, 300 mM NaCl, 0.5% NP-40 and 0.5 mM ethylenediaminetetraacetic acid (EDTA)) supplemented with protease and phosphatase inhibitors (1 mM dithiothreitol (DTT), 5 mM Na fluoride, 0.2 mM sodium vanadate, 10 µg ml$^{-1}$ leupeptin, 10 µg ml$^{-1}$ pepstatin and 0.1 mM phenylmethylsulfonyl fluoride, Sigma). Insoluble material was collected by centrifugation

at 16,000$g$ for 10 min, and washed once with NP-40 buffer. Insoluble pellet was boiled for 15 min in urea buffer (1% SDS, 9 M urea, 25 mM Tris−HCl pH 6.8, 1 mM EDTA and 100 mM DTT) for the extraction of the chromatin fraction.

### DNA fibre analysis

Cells were sequentially pulse labelled with 30 µM CldU (MP Biomedicals) and 250 µM IdU (Sigma-Aldrich) according to the schematic in each figure. After labelling, cells were collected and resuspended in PBS at $2.5 × 10^5$ cells ml$^{-1}$. Spreading and labelling of the DNA was performed as in ref. [59] with the following conditions for the primary antibodies. CldU was detected using Anti-BrdU (BU1/75 (ICR1)) (ab6326, Abcam) diluted 1:100 in Blocking Buffer (PBS, 2% bovine serum albumin (BSA) and 0.1% Tween-20); IdU was detected using Anti-BrdU (Clone B44) (347580, BD Bioscience) diluted 1:100 in Blocking Buffer.

The DNA fibre assay with the ssDNA-specific S1 nuclease (S1 fibre), was performed as described[64]. Briefly, cells were pulse labelled with 30 µM CldU for 20 min, then treated with 1 mM HU for 1 h and released from HU in the presence of 250 µM IdU for 1 h. Cells were then permeabilized with CSK100 (100 mM NaCl, 10 mM MOPS pH 7, 3 mM $MgCl_2$, 300 mM sucrose and 0.5% Triton X-100 in water) for 10 min at room temperature, treated with the S1 nuclease (Thermo Fisher Scientific) at 20 U ml$^{-1}$ in S1 buffer (30 mM sodium acetate pH 4.6, 10 mM zinc acetate, 5% glycerol and 50 mM NaCl in water) for 30 min at 37 °C, and collected in PBS with 0.1% BSA with cell scraper. Nuclei were then pelleted at ~7,000 r.p.m. for 5 min at 4 °C, then resuspended in PBS. Spreading and labelling of the DNA was performed as in ref. [59]. Fibres were visualized and imaged with a Metafer slide scanner (Metasystem) using a 40× Plan-Neofluar 0.75 numerical aperture (NA) air objective. ImageJ software was used for the quantification.

### Chromatin fibre analysis (ChromStretch)

Chromatin fibres were prepared mostly as described in refs. [38,39] with the following modifications. After the treatments, a minimum of $3 × 10^6$ cells were collected and washed twice in cold 1× PBS. To facilitate chromatin isolation and spreading, the cellular membrane were lysed for 5 min on ice in 10 mM HEPES pH 7.9, 10 mM KCl, 1.5 mM $MgCl_2$, 0.34 M sucrose, 10% glycerol, 1 mM DTT and protease inhibitor (cOmplete, mini, EDTA-free Protease Inhibitor Cocktail, Roche). The resulting nuclei were collected by centrifugation (1,500$g$ for 5 min) 4 °C and resuspended in hypotonic buffer (3 mM EDTA, 0.2 mM egtazic acid, 1 mM DTT and protease inhibitor). The nuclei were then spotted on a Superfrost microscope slides and allowed to settle for 5 min in a humid chamber. The slides were then tilted to remove the excess buffer and were allowed to dry for a maximum of 5 min before being transferred in a lysis chamber containing lysis buffer at pH 7 and incubated for a total of 10 min. The stretching of the chromatin fibres was facilitated by flowing the lysis buffer out of the lysis chamber at a constant flow using an equipment that was design and built in the lab. Stretched fibres were finally fixed in 4% formaldehyde for 15 min. Slides were washed three times in PBS, EdU was labelled with Alexa Fluor 594 azide according to the manufacturer protocol for 30 min, slides were washed once in PBS and blocked in 1× PBS 5% BSA for 1 h and incubated in primary antibodies over night at 4 °C. Primary antibodies were rabbit monoclonal antibody to H3K9me3 (abcam ab176916, 1:1,000), mouse monoclonal antibody to H3K9me2 (abcam ab1220, 1:1,000), rabbit monoclonal antibody to H3K9me1 (abcam ab176880, 1:1,000). Primary antibodies were then labelled with the appropriate anti rabbit or anti mouse antibody conjugated with Alexa Fluor 488 diluted 1:1,000 in blocking buffer for 1 h at room temperatures. Chromatin was counterstained using rabbit polyclonal anti H3 antibody (ab1791, 1:1,000) or mouse monoclonal anti H3 antibody (ab195277, 1:1,000) in blocking buffer for 1 h at room temperature followed by a 1 h incubation at room temperature in anti-rabbit conjugated with Alexa Fluor 647 (1:1,000) or in anti-mouse conjugated with Alexa Fluor 647 (1:1,000).

Chromatin fibres were visualized using a Leica ST5 confocal microscope equipped with an oil immersion 63× (HC PL APO CS2, NA 1.4) objective. Quantification of the H3K9me1/2/3 signal overlapping with EdU signal was performed using ImageJ.

## DNA combing

Cells were sequentially pulse labelled with 30 µM CldU (MP Biomedicals) and 250 µM IdU (Sigma-Aldrich) for 20 min each. Cells were collected, washed twice in PBS and resuspended in PBS at a concentration of $1.6 × 10^6$ cells ml$^{-1}$. DNA was extracted after encapsulation of cells in low-melting-point agarose blocks at 70,000 cells per plug and combed on silanized coverslips as described[97]. Detection of IdU and CldU labels was performed as described in the DNA fibre analysis procedure. Total DNA was labelled for 1 h with anti-ssDNA antibody (AB_10805144, DSHB, 1:50), followed by 1 h incubation in the dark with anti-mouse Alexa Fluor 350 (1:50) (Invitrogen). DNA fibres were then visualized and imaged as described above (DNA fibre analysis).

## Immunoblot and antibodies

Samples were loaded on 4–12% NuPAGE Bis-Tris Gel (Novex Life Technologies) and transferred to a polyvinylidene difluoride membrane (0.45 µm, Immobilon). Membranes were blocked with 5% BSA in PBS for 1 h at room temperature and incubated with primary antibodies diluted 1:1,000 in blocking buffer overnight at 4 °C. Primary antibodies were: H3K9me1 (Upstate, 07-450), H3K9me3 (Millipore, 07-442) mouse anti-H3 (Abcam, ab10799), γH2A.X (Millipore, 05-636), Chk1p (Cell Signaling, 2344), Chk1 (DCS-310 (ref. 98)), p53-S15p (Cell Signaling, 12571), p53 (Sigma-Aldrich, mouse monoclonal antibody, clone DO-1) and β-actin (Sigma-Aldrich, A5316). Membranes were washed in 0.1% Tween-20 in PBS on the following day, followed by incubation with secondary antibody coupled to near-infrared dyes CF 680/CF 770 (1:10,000). Antibodies were visualized using an Odyssey CLx infrared scanner (LiCor).

## Immunofluorscence staining for STED microscopy

Cells were labelled with EdU (10 µM) for 20 min. For HU-treated samples, EdU is labelled before the HU treatment. After the treatments, cells were pre-extracted with 0.1% Triton X-100 in ice-cold CSK buffer for 5 min at 4 °C and fixed in 4% formaldehyde in PBS for 15 min at room temperature. Samples were then washed thoroughly in PBS and permeabilized in 0.1% Triton-X 100 in PBS for 10 min, and blocked with 5% BSA in PBS. Samples were subsequently stained with a rabbit anti H3K9me3 antibody (abcam ab176916, 1:1,000) diluted in blocking buffer, followed by incubation in an anti-rabbit antibody conjugated to abberior star 635p (1:1,000). EdU was visualized with a Click-it reaction using abberior STAR 580 (abberior) according to the manufacturer's protocol. Samples were washed with PBS and incubated with YoYo-1 for 15 min. ProLong Gold antifade mountant (Invitrogen) was used to mount the samples on the glass slides for coverslip samples.

Imaging was performed on a Leica SP8 confocal/STED microscope equipped with a white light laser and a pulsed 775 nm depletion laser using a water immersion 86× (HC PL APO STED, NA 1.2) objective with a motorized coverslip correction ring (motCORR™). The sample was excited with 561 nm and 633 nm, respectively, and emission was filtered appropriately (570–620 nm, 650–700 nm) and gated for lifetime between 0.3 and 6.0 ns. The coverslip correction ring and STED beam were adjusted before imaging.

## High-content PLA

PLA experiments were performed as described in ref. 59. Cells were grown on cover slips until 60% confluency. Primary antibodies used for PLA are: Anti-Biotin antibody (A150-109A, Bethyl Laboratories), Anti-Biotin antibody (AB_2339006, JacksonImmunoResearch), Anti-H3K9me3 (EPR16601) (Ab176916, Abcam), Anti-H3K9me2 (Ab1220, Abcam), Anti-H3K9me1 (EPR16989) (Ab176880, Abcam), Anti-G9a (EPR18894) (Ab 185050, Abcam), Anti-HDAC1 (Ab19845, Abcam), Anti-BRCA1 (D-9)

(SC6954, Santa Cruz Biotechnology), Anti-BARD1 (A300-263A, Bethyl), Anti-RPA32/RAP2 (9H8) (Ab2175, Abcam), Anti-PCNA (PC10) (ab29, Abcam), Anti-H4K20me0 (EPR22116) (Ab227804, Abcam), Anti-H4K16ac (EPR1004) (Ab109463, Abcam), Anti-RAD51 (70-002, Bio Academia), Anti-H2AK15ub (EDL H2AK15-4) (MABE1119, Millipore). All primary antibodies were diluted 1:1,000 in PBS, 5% BSA.

After washes with PBS with 0.1% Tween-20 (PBST), cells were incubated with anti-mouse minus and anti-rabbit plus PLA probes (Sigma-Aldrich) at 37 °C for 1 h. Following the manufacturer's instructions, the PLA reaction was performed with the Duolink In Situ Detection Reagents. Cells were stained with 4′,6-diamidino-2-phenylindole (DAPI) and mounted on slides using ProLong Gold. Images were captured using Metafer5 and quantified using MetaSystem. Images were captured using Metafer5 and quantified using MetaSystem. PLA spot intensity (a.u.) is calculated as the product of number of spots and the mean intensity of the spots per nucleus.

## ChIP–seq

**Sample preparation, library preparation and sequencing.** TIG3 cells for the indicated conditions were cross-linked for 10 min in 1% formaldehyde and chromatin was fragmented by sonication using Bioruptor Sonicator (Diagenode). Chromatin immunoprecipitation was performed as previously described[99] with antibodies against H3K9me3 (5 µg, Abcam ab8898) and H3 (2 µg, Abcam ab10799). The immunoprecipitated DNA was quantified by Qubit fluorometer (Life Technologies). DNA library for Illumina sequencing was prepared from 10 ng DNA, using NEBNext ChIP-Seq Library Prep Master Mix Set for Illumina (New England Biolabs) and following the manufacturer's instructions. Equimolar amounts of samples, with compatible indexes, were pooled for multiplex sequencing. For all samples, single-end sequences were generated on the Illumina HiSeq2000 platform at the Danish National High-throughput DNA Sequencing Centre.

**Data analysis.** ChIP–seq data are available at the Gene Expression Omnibus (PRJNA897702). Raw reads were aligned to the human genome (hg19 assembly excluding non-canonical chromosomes that is random, unknown and haplotype variant chromosomes) using Bowtie version 0.12.7 with default parameters except '-S -m1', which excludes reads mapping to multiple chromosomal positions. Peak detection was performed with MACS2 version 2.0.9 (20111102) using default settings except for parameters '–broad–nomodel–shiftsize=110'. The shift size of 110 bp was calculated as the median over all Phantom Peak[100] shift estimates for our H3K9me3 samples. When running differential peak detection between two H3K9me3 samples in MACS2 the additional parameter '–shift-control' was specified. Bigwig files were generated using the UCSC Kent utilities[101]. We allowed only one read per chromosomal position thus eliminating potential spurious spikes, and each remaining read was extended from its 5′-end to a total length of 250 bases, before converting to bedGraph format, scaling to mapped reads per million and final conversion to bigwig format. Individual BigWig files were uploaded to the UCSC browser for visualization[101,102]. To generate chromosome-wide landscapes of H3K9me3 and H3 we used the mean as the combining function and a smoothing window of 4 pixels. Overlay plots were generated by creation of a track hub at UCSC browser[103], where individual BigWig tracks were combined into a multi-Wig display with two coloured transparent graphs overlaid in the same vertical space. We used the integrative analysis tools from the Cistrome platform[104] to calculate Pearson correlation coefficients for multiple signal profiles on a whole-genome scale using non-overlapping windows of 250 bp. The association of H3K9me3 peaks with annotated genomic features was calculated using the Cis-regulatory Element Annotation System (CEAS) package[105]. Hilbert curve visualization of ChIP–seq data was generated using the HilbertVis application[34]. Hilbert plots allow the visualization of linear sequence data in two-dimensional space. Each coloured spot in the figure correspond to a peak where the

area of the spot is proportional to the width of the peak and the intensity of the spot corresponds to the height of the peak.

**RNA extraction and RNA sequencing.** Total RNA was extracted using the ReliaPrep RNA Miniprep Systems (Promega) according to the manufacturer's instructions. Five-hundred nanograms of total RNA was used for mRNA sequencing preparation using the Quantseq 3′mRNA kit following the manufacturer's protocol. NGS (next-generation sequencing) short reads were aligned to the GRCh38 human genome using the Star aligner. The log$_2$ fold change in gene expression relative to wild type for each sample was computed from read counts using DEGSeq, and box plots were produced using the R packages.

**Histone extraction, digestion and mass spectrometry analysis.** Total histones from TIG3 cells were isolated by acid extraction. Digestion and mass spectrometry analyses were performed as described in ref. 31. The relative quantification for a given peptide was obtained by dividing its quantification by the sum of all quantifications of all peptides sharing the same amino acid sequence. The mass spectrometry raw data are available upon request.

**iPOND-SILAC mass spectrometry.** For iPOND experiments, heavy lysine- and arginine-labelled mESCs were pre-treated with UNC0642 at a concentration of 1 μM 2 h before the beginning of the experiment. Light lysine- and arginine-labelled mESCs were pre-treated with same amount of dimethyl sulfoxide (DMSO) at the same time. Both light- and heavy-labelled mESCs were then incubated with 10 μM EdU for 10 min, with and without treatment of 4 mM HU (Sigma-Aldrich) for 3 h to stall the DNA replication forks. After labelling and treatment cells were cross-linked with 1% formaldehyde for 10 min at room temperature, quenched with 0.125 M glycine, washed with PBS and collected using cell scrapper. Samples were then treated with Click-it reaction containing 25 μM biotin-azide, 10 mM (+) sodium L-ascorbate and 2 mM CuSO$_4$ and rotated at 4 °C for 1 h. Samples were then centrifuged to pellet down the cells; supernatant was removed and replaced with 1 ml Buffer-1 (B1) containing 25 mM NaCl, 2 mM EDTA, 50 mM Tris–HCl, pH 8.0, 1% IGEPAL and protease inhibitor and rotated again at 4 °C for 30 min This step was repeated twice. Samples were centrifuged to pellet down the cells; supernatant was removed and replaced with 500 μl of B1 and sonicated using a Bioruptor Sonicator (Diagenode) using cycles of 20 s on, 90 s off 30 times at high amplitude. Samples were centrifuged, and supernatant was transferred to fresh tubes and incubated for 1 h with 200 μl of Dynabeads MyOne C1 (Sigma-Aldrich) for the streptavidin biotin capture step. Proteins were eluted, and mass spectrometry was performed. At least two peptides were required for protein identification. Quantitation is reported as the log$_2$ of the normalized heavy/light ratios with respect to mcm6. SILAC data were analysed using Proteome Discoverer (ThermoFisher).

**Clonogenic survival assay.** Cells were seeded in triplicate in 10 cm culturing dish and treated with different concentrations of olaparib throughout the whole experimental process, or different concentrations of cisplatin, for 4 h before being washed off and replaced with new medium.

After 1 week, colonies were fixed and stained in a mixture of 43% water, 50% methanol, 7% acetic acid and 0.1% Brillant Blue R (Sigma-Aldrich) and subsequently counted with Gelcount (Oxford Optronix). The survival was plotted as the mean percentage of colonies detected following the treatment normalized to the mean number of colonies from the untreated samples.

**Flow cytometry**
**TIG3 cells.** For quantification of γH2AX and H3S10p by fluorescence-activated cell sorting (FACS) cells were collected by trypsinization, fixed in 70% ethanol and permeabilized in 0.2% Triton

X-100. Fixed cells were stained with primary antibodies diluted in PBS–1% FBS (mouse-anti-γ H2AX (1:500; Millipore, 05-636) or rabbit-anti-phospho-H3S10 antibody (1:500; Millipore, 06-570)) for 1 h followed by 1 h incubation with anti-mouse or anti-rabbit secondary antibody conjugated with Alexa488 (1:1,000; Invitrogen). For quantification of DNA-replicating cells by FACS, cells were pulse labelled with 40 μM BrdU before collection and ethanol fixation. For detection of total BrdU incorporation in double-strand DNA, fixed cells were treated with 2 M HCl (30 min) to denature DNA before a 2 h staining with mouse-anti-BrdU antibody (1:20; BD Biosciences, 347580) diluted in PBS–1% FBS followed by 1 h incubation with anti-mouse secondary antibody conjugated with Alexa488 (1:100; Invitrogen). DNA was stained using 0.1 mg ml$^{-1}$ propidium iodide supplemented with RNase A (20 μg ml$^{-1}$) for 30 min at 37 °C. Flow cytometry analysis was performed on FACSCalibur using CellQuest Pro software (BD). Quantification and analysis of cell cycle profiles were obtained using FlowJo (version 7.2.2; Tree Star, Inc.).

**MRC-5 cells.** Cells were grown to 70–80% confluency in a 10 cm culturing dish. Cells were labelled with EdU for 30 min followed by fixation for 10 min in 4% formaldehyde in PBS at room temperature. Cells were then washed with 1% BSA/PBS and permeabilized in 0.5% saponin buffer in 1% BSA/PBS. Incorporated EdU were labelled with the Click-it reaction using Alexa Fluor 594 azide according to the manufacturer's protocol (Invitrogen). DAPI was used to stain the DNA. Single nuclei were selected using SSC-A versus FSC-A, followed by FSC-H versus FSC-W and SSC-H versus SSC-W.

**Analysis of patient survival using ovarian cancer datasets.** Patient survival analysis was performed using microarray datasets of ovarian tumours from The Cancer Genome Atlas (TCGA)[95,96] (https://link.springer.com/article/10.1007/s11357-023-00742-4/tables/1), and KM-plotter was used to generate the Kaplan–Meier plot. Mean expression of probes for GLP and G9a was calculated, and combined GLP/G9a expression was used to identify patients with high and low expression and plotted for overall survival ($n = 655$) and progression-free survival ($n = 614$) using KM-plotter.

### Statistics and reproducibility
Experimental data were plotted and analysed using either Microsoft Excel or GraphPad Prism 9.4.1 (GraphPad Software) built-in tests, and are indicated in the figure legends, unless otherwise indicated. All box plots show plain horizontal line representing the median and when present, and the blue dashed line represent the mean of the dataset. The box contains the 25th to 75th percentiles of the dataset, the whiskers mark the 10th and 90th percentiles and values beyond these upper and lower bounds are considered outliers and marked with a dot. The number of samples analysed per experiment are reported in the respective figure legends. All experiments were independently repeated at least two times with similar results obtained.

### Reporting summary
Further information on research design is available in the Nature Portfolio Reporting Summary linked to this article.

## Data availability
Deep-sequencing (ChIP–seq and RNA sequencing) data that support the findings of this study have been deposited as a Bioproject under accession code PRJNA845122, for the RNA sequencing data, and PRJNA897702, for the ChIP–seq data. Mass spectrometry data have been deposited in ProteomeXchange with the primary accession codes PXD041742 for silac data and PXD041914 for the proteomics analysis of histone PTM levels. The human ovarian cancer data analysed in this study were from the TCGA datasets[95,96] (https://link.springer.com/article/10.1007/s11357-023-00742-4/tables/1). Source data are provided

with this paper. All other data supporting the findings of this study are available from the corresponding author on reasonable request.

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

## Acknowledgements

We thank T. Sixma, K. Luger and X. Zhang for stimulating discussions; H. van Attikum and M. Luijsterberg for kindly providing PAGFP-H2A construct and for sharing technical information; P. Krawczyk for sharing technical information on UNC0642; A. Vindigni for sharing PRIMPOL-overexpression plasmid; and J. Vilstrup Johansen for providing bioinformatics support for analysing ChIP–seq data. Funding was provided by Dutch Cancer Society (NWO) Women in STEM Incentive Grant ENW/01054017/12412, Vidi funding (project no. 114122) and ERC funding (grant no. 101078750/#114168) support to N.T. This study was supported by the Oncode Institute, which is partly financed by the Dutch Cancer Society (R.K.). We thank the Josephine Nefkens Cancer Program for infrastructure support. Danish National Research Foundation to the Center for Epigenetics (DNRF82 to S.S., A.T. O.N.J. and A.G.). Proteomics and mass spectrometry research at University of Southern Denmark was supported by VILLUM Center for Bioanalytical Sciences (VILLUM Foundation grant no. 7292 to O.N.J.) and by PRO-MS, Danish National Mass Spectrometry Platform for Functional Proteomics (grant no. 5072-00007B to O.N.J.).

## Author contributions

N.T., V.G. and C.S.Y.L. designed and analysed the majority of experiments. ChromStretch technology is developed in collaboration with R.K. N.T. conceived and supervised the project. C.S.Y.L. performed iPOND-coupled SILAC-based mass spectrometry, and J.D. analysed iPOND-MS data. D.W., S.S. and Y.C. analysed genomics data. V.G. performed all DNA combing and DNA fibre analysis. H.D., H.N. and S.U. generated cell lines, and performed and analysed all PLA and ChromStretch assays under supervision of V.G., and C.S.Y.L. and J.A.S. assisted in STED imaging. C.S.Y.L., S.U. and E.M.M. performed colony survival assays. S.C., and G.S.C.S.G, performed TCGA analysis. The project was executed in collaboration with A.G., supervising N.R.-G., and Z.J. performed ChIP–seq experiment, and histone PTM mass spectrometry experiments. O.N.J. supervised S.S. and A.T. in analysing histone PTM mass spectrometry data. N.T., V.G., C.S.Y.L., A.G. and N.R.-G. co-wrote the manuscript with input from all authors.

## Competing interests

A.G. is co-founder and CSO in Ankrin Therapeutics. N.T., V.G. and R.K. report an international patent filed under PCT/NL2023/050120 for ChromStretch technology. No other authors have competing interests.

## Additional information

**Extended data** is available for this paper at https://doi.org/10.1038/s41556-023-01167-z.

**Correspondence and requests for materials** should be addressed to Nitika Taneja.

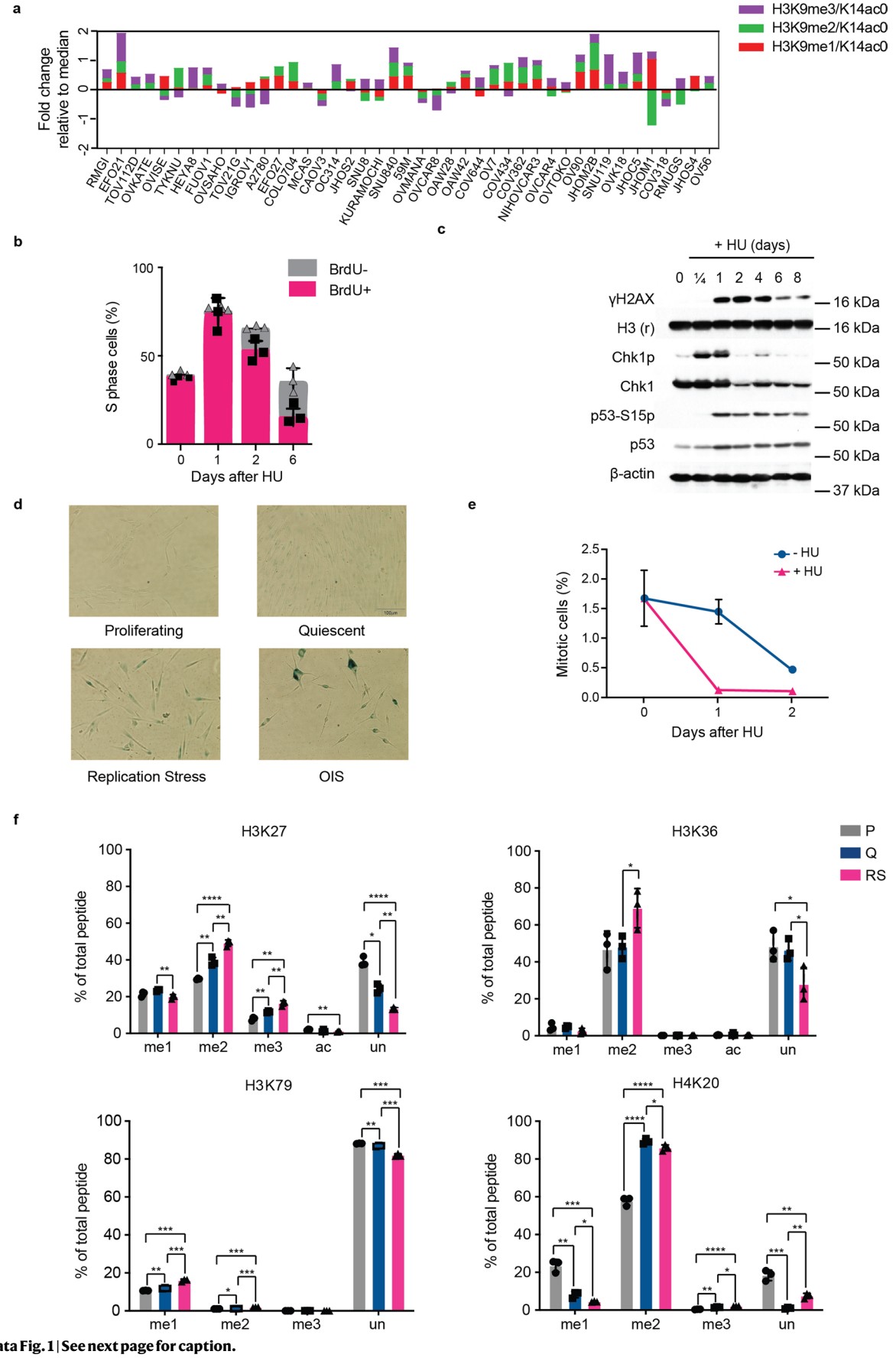

**Extended Data Fig. 1 | See next page for caption.**

**Extended Data Fig. 1 | Extended Data Figure. 1 related to Fig. 1. H3K9me3 accumulates in cancer cells and upon persistent replication stress condition.** (**a**) Bar chart representing global chromatin profiling for enrichment of deacetylated H3K9me1/K14ac0 (red), H3K9me2/K14ac0 (green) and H3K9me3/K14ac0 (purple) epigenetic marks in >40 different ovarian cancer cell lines (the name of each cell line is indicated on the x axis) were analyzed from CCLE database 23. For each chromatin mark, the fold change relative to median value of the respective ovarian cancer cell lines is shown. (**b**) Analysis of S phase cells undergoing active replication. Cells were pulse-labeled with BrdU for 45 min at the indicated time. BrdU incorporating fractions of S-phase cells were determined by flow cytometry. The percentages of BrdU positive cells are indicated as the mean +/- SD of three independent experiments. (**c**) Immunoblot analysis of DNA damage and checkpoint signaling. Phosphorylated histone H2AX (γH2A.X), Ser317-phosphorylated Chk1 (Chk1p) and Ser15-phosphorylated p53 (p53-S15p). This experiment was reproduced independently three times with similar results. (**d**) Persistent replication stress induces senescence, as demonstrated by high senescence-associated-β-galactosidase (SA-β-Gal) activity. TIG3 cells expressing the oncogene B-RAF (OIS: Oncogene induced senescence) were used as positive control for the presence of SA-β-Gal positive cells. This experiment was reproduced independently three times with similar results. (**e**) Analysis of mitotic cells. Histone H3 serine 10 phosphorylation (H3S10p) was analyzed by flow cytometry of TIG3 cells treated as indicated. The percentages of H3S10p positive cells are indicated as the mean +/- SD of three independent experiments. (**f**) Analysis of histone PTM levels in proliferating (grey), quiescent (blue) and HU-treated (pink) TIG3 fibroblasts. The graph show the average of three biological replicates with error bars indicating SD. Unpaired two-sided t-test: (****) P < 0.0001; (***) P < 0.001; (**) P < 0.01; (*) P < 0.05. Source numerical data and unprocessed blots are available in source data.

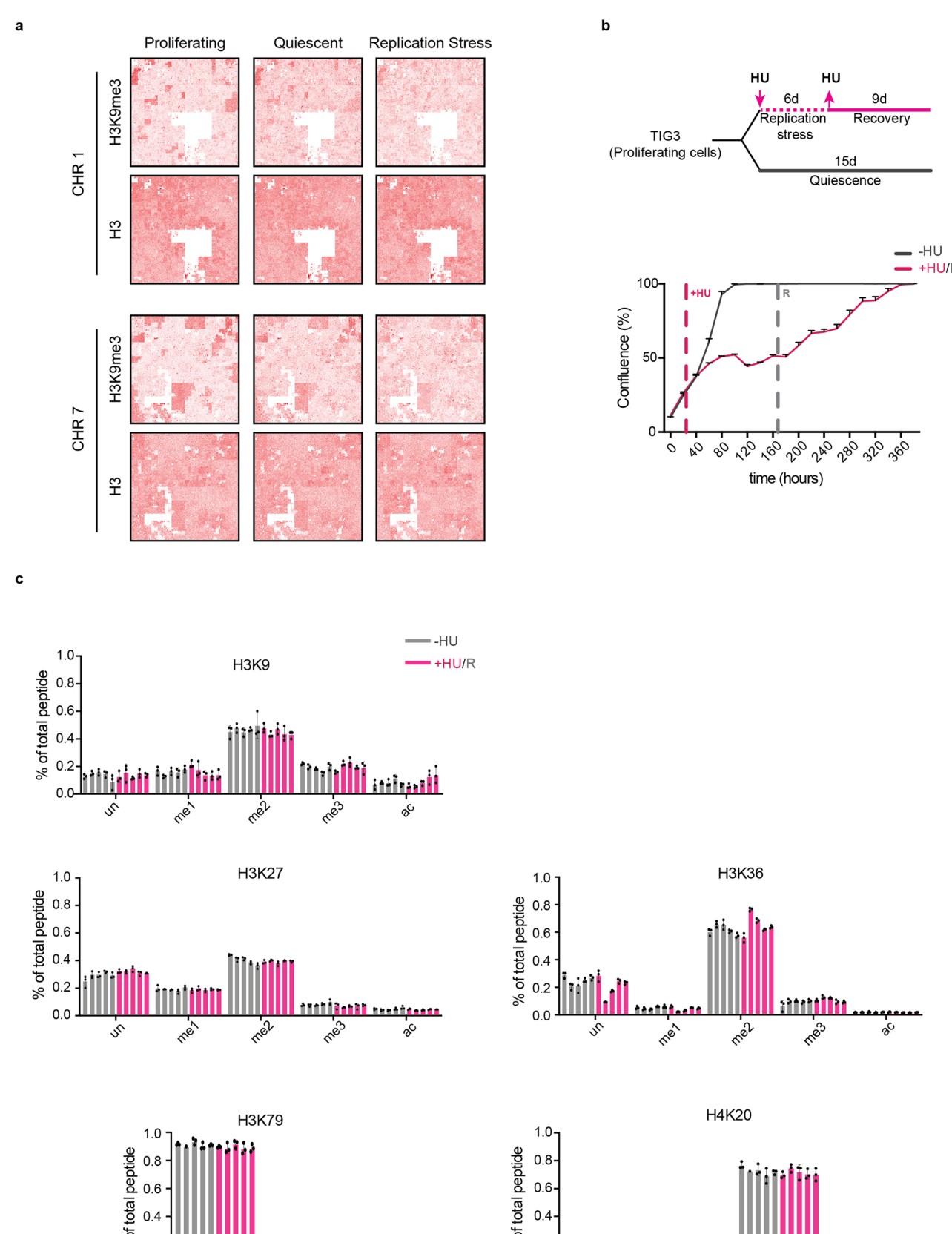

**Extended Data Fig. 2 | See next page for caption.**

**Extended Data Fig. 2 | Extended Data Figure. 2 related to Fig. 1. Profiling of histone PTMs by quantitative mass spectrometry upon recovery from replication stress.** (**a**) Chromosome-wide profiles of ChIP-seq data for H3K9me3 and total H3 visualized Hilbert curves. Profiles for chromosomes 1, 7 for proliferating (P), quiescent (Q) and HU-treated (RS) cells in two biological replicates are shown. (**b**) Experimental design (top). TIG3 cells were treated for 6 days with 600 μM HU and allowed to recover for 9 days after removal of the drug (+HU/R) or cultured in the absence of HU for the whole period (-HU).

Analysis of cell proliferation by high-content live-cell imaging (bottom). The graphs show the mean confluence (%) +range from two technical replicates and are representative of two independent biological replicates. (**c**) Analysis by quantitative mass spectrometry of histone PTMs in single cell clones derived from proliferating cells (control) or cells allowed to recover after persistent replication stress (HU recovery). Five clones were analyzed for each condition. The graphs show the average of three technical replicates with error bars indicating SD.

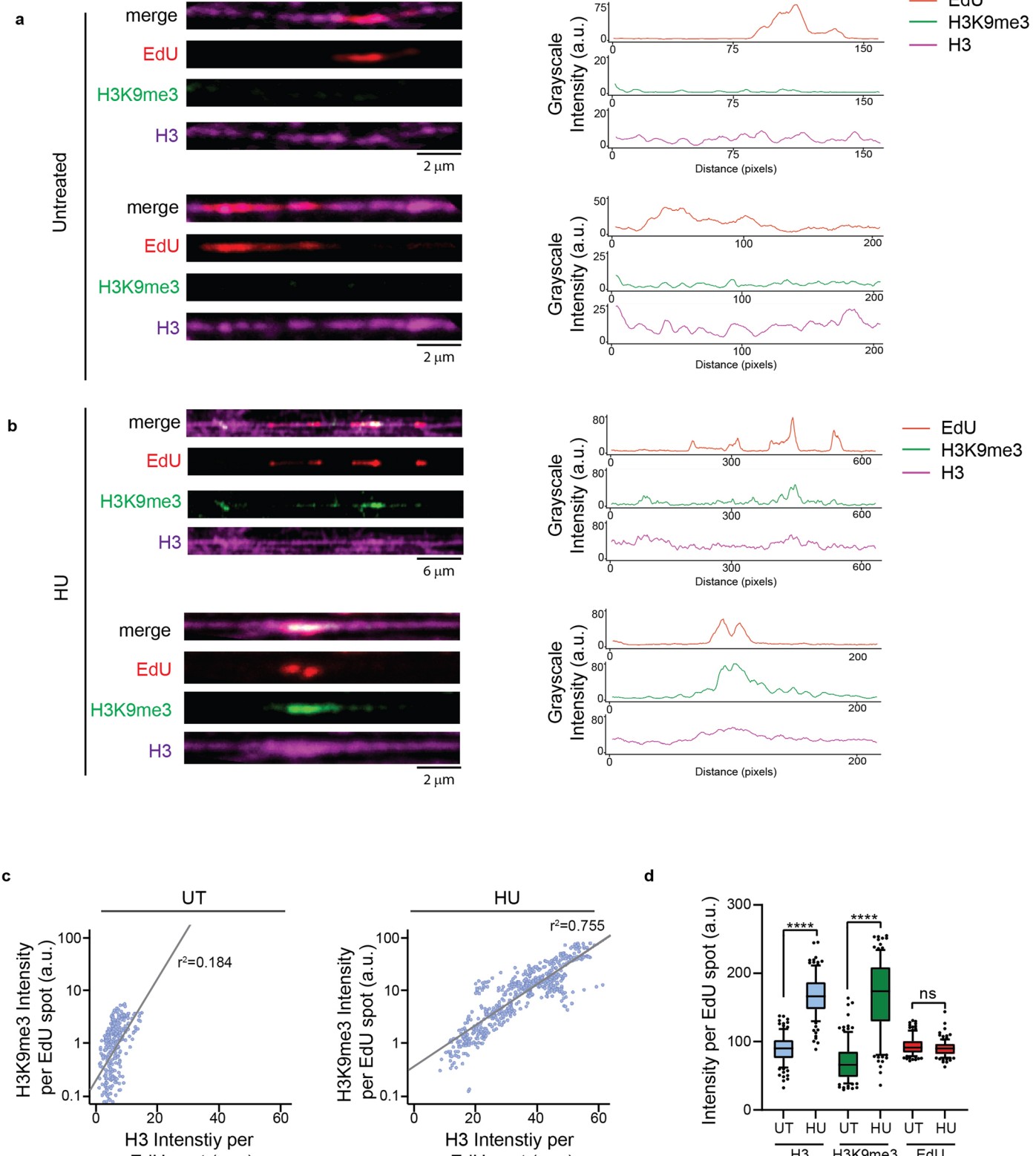

**Extended Data Fig. 3 | Extended Data Figure. 3 related to Fig. 2. Single molecule analysis of H3K9me3 accumulation at stressed replication forks.** (**a**, **b**) Left, representative chromatin fibers in the absence of treatment (**a**) and after a 1 hr incubation in the presence of 1 mM HU (**b**). Right, intensity profiles of the each representative fibers. The intensity profiles of EdU (red), H3K9me3 (green) and H3 (blue) have been plotted. These experiments were reproduced independently 3 times with similar outcomes. (**c**) Correlation analysis of H3K9me3 and H3 at EdU spot is shown both for untreated (left) and for cells treated with 1 mM HU for 1 hr (right). R2 indicate the correlation coefficient between H3K9me3 and H3 intensity distribution. (**d**) Analysis of ChromStretch fibers. Quantification of the intensity of H3, H3K9me3 and EdU both at ongoing (UT) and stalled (HU) replication forks. ($n_{UT} = 106$, $n_{HU} = 104$ individual replication tracks analyzed; **** = P ≤ 0.0001, ns = non-significant, One-way ANOVA). Source numerical data are available in source data.

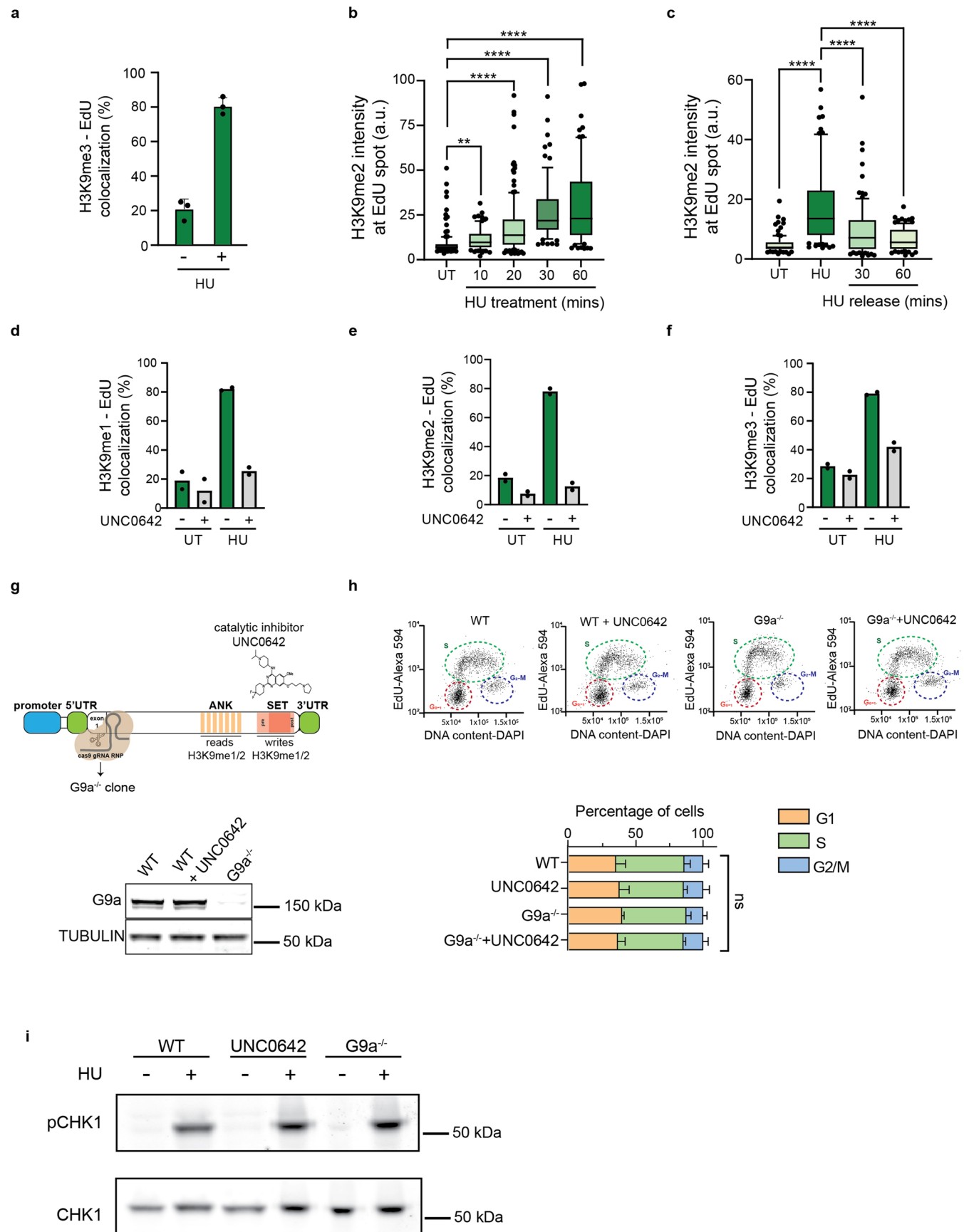

**Extended Data Fig. 4 | See next page for caption.**

**Extended Data Fig. 4 | Extended Data Figure. 4 related to Fig. 2. Transient accumulation of H3K9me and G9a at stalled replication forks is dependent upon checkpoint activation.** (**a**–**f**) ChromStretch analysis assessing: (**a**) Dynamics of H3K9me3 at replication sites at ongoing (UT) and stalled (HU) replication forks. Mean ± SD percentage of colocalization of H3K9me3 and EdU is shown as a bar plot. (n = 3 independent experiments). (**b**) Dynamics of H3K9me2 at replication sites upon replication stress. For experimental design see Fig. 2c. Quantification of H3K9me2 at EdU sites for the indicated conditions. ($n_{UT}$ = 132, $n_{HU10}$ = 74, $n_{HU20}$ = 140, $n_{HU30}$ = 75, $n_{HU60}$ = 86; **** = P ≤ 0.0001, ** = P ≤ 0.01, Kruskal-Wallis test followed by Dunn's test). (**c**) Dynamics of H3K9me2 at replication sites after release from replication stress. For experimental design see Fig. 2d. Quantification of H3K9me2 at EdU sites for the indicated conditions. ($n_{UT}$ = 100, $n_{HU}$ = 92, $n_{rel30}$ = 92, $n_{rel60}$ = 93; **** = P ≤ 0.0001, Kruskal-Wallis test followed by Dunn's test). (**d**–**f**) Dynamics of H3K9 PTM at replication sites in the presence (UNC0642-) or in the absence of G9a activity (UNC0642+) at ongoing (UT) and stalled (HU) replication forks. Bar plot of the mean of the percentages of H3K9me1 (**d**), H3K9me2 (**e**), H3K9me3 (**f**) colocalization with EdU. (n = 2 independent experiments). (**g**) Top: Schematic representation of the G9a isoform A. Exon1 was targeted using CRISPR/Cas9 to generate a G9a knock-out. UNC0642 binds G9a SET domain preventing G9a catalytic activity. Bottom: Immunoblot showing G9a levels in wild type cells (WT), wild type where G9a activity was inhibited with UNC0642 (WT + UNC0642) for 2hrs and a G9a knockout clone (G9a-/-). Tubulin is used as a loading control. (**h**) Top: Cell cycle profile of the indicated cells. Bottom: Mean percentage of cells in various phases of the cell cycle ± standard deviation from 3 independent experiments. (ns = non-significant, One-way ANOVA). (**i**) Immunoblot showing pCHK1 levels in wild type cells (WT), wild type where G9a activity was inhibited with UNC0642 (WT + UNC0642) and a G9a knock out clone (G9a-/-), in the presence or in the absence of replication stress. CHK1 is used as a loading control. This experiment was reproduced independently three times with similar results. Source numerical data and unprocessed blots are available in source data.

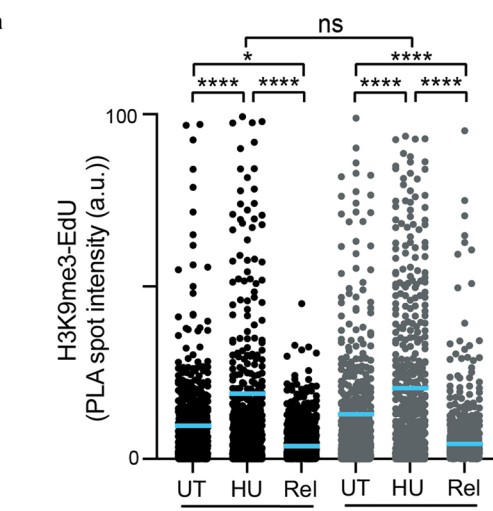

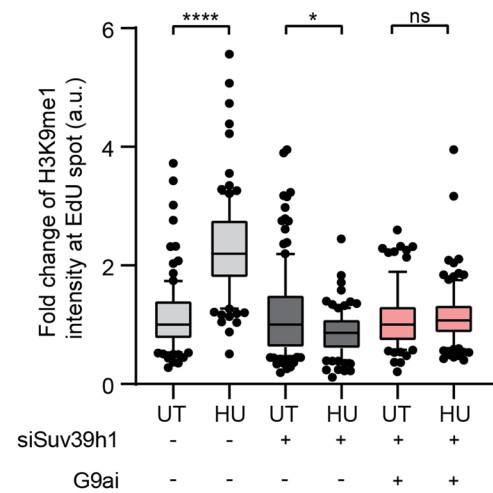

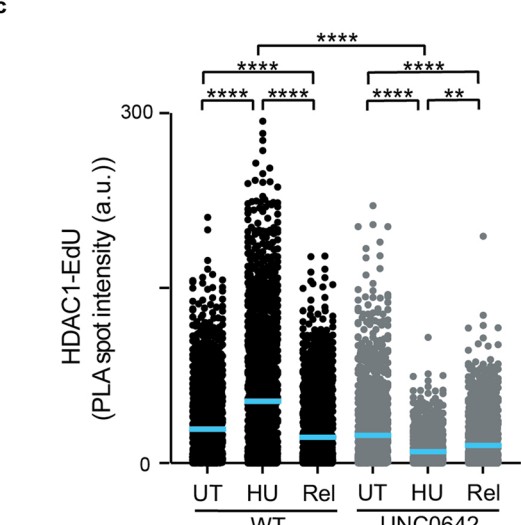

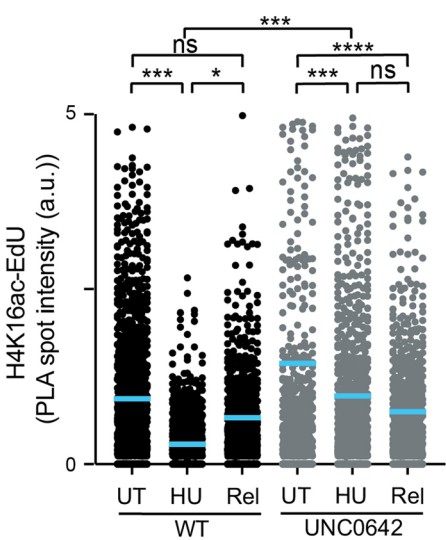

**e**

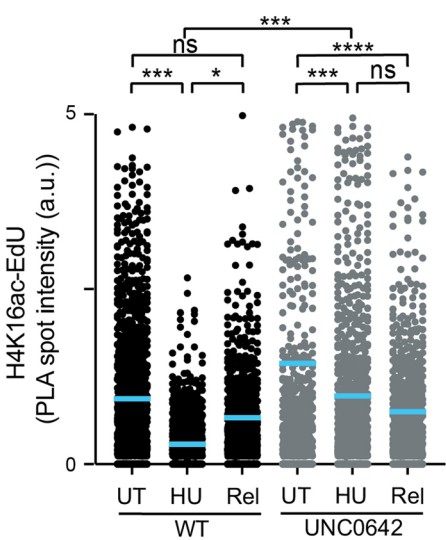

**Extended Data Fig. 5 | See next page for caption.**

**Extended Data Fig. 5 | Extended Data Figure. 5 related to Fig. 3. Recruitment of HDAC1 and H4K16 deacetylation at stalled replication forks is H3K9me dependent.** (**a**) Plot showing distribution of H3K9me3-EdU total PLA spot intensity per nucleus for the indicated conditions. ($n_{siCTL-UT}$ = 925, $n_{siCTL-HU}$ = 951, $n_{siCTL-REL}$ = 990, $n_{siSETDB1-UT}$ = 1071, $n_{siSETDB1-HU}$ = 1040, $n_{siSETDB1-REL}$ = 1138 cells analyzed; blue dashed line represents the mean of the distribution, **** = P ≤ 0.0001, * = P ≤ 0.05, ns = non-significant, Kruskal-Wallis test followed by Dunn's test). (**b**) Analysis of ChromStretch fibers to assess the dynamics of H3K9me1 at replication upon replication stress. Quantification of H3K9me1 signal overlapping with EdU for the indicated condition. The signal is represented as a fold increase compared to the mean H3K9me1 signal of the untreated condition. ($n_{siCTL-UT}$ = 100, $n_{siCTL-HU}$ = 100, $n_{siSUV39h1-UT}$ = 132, $n_{siSUV39h1-HU}$ = 100, $n_{siSUV39h1+UNC0642-UT}$ = 87, $n_{siSUV39h1+UNC0642-HU}$ = 100 cells analyzed; **** = P ≤ 0.0001, ns = non-significant, Kruskal-Wallis test followed by Dunn's test). (**c**) Plot showing distribution of HDAC1-EdU total

PLA spot intensity per nucleus for the indicated conditions. ($n_{WT-UT}$ = 1691, $n_{WT-HU}$ = 1871, $n_{WT-REL}$ = 1771, $n_{UNC0642-UT}$ = 1534, $n_{UNC0642-HU}$ = 1798, $n_{UNC0642-REL}$ = 1652 cells analyzed; blue dashed line represents the mean of the distribution, **** = P ≤ 0.0001, ** = P ≤ 0.01, Kruskal-Wallis test followed by Dunn's test). (**d**) Plot showing distribution of H4K16ac-EdU total PLA spot intensity per nucleus for the indicated conditions. ($n_{WT-UT}$ = 1507, $n_{WT-HU}$ = 1254, $n_{WT-REL}$ = 1489, $n_{UNC0642-UT}$ = 1187, $n_{UNC0642-HU}$ = 1488, $n_{UNC0642-REL}$ = 1365 cells analyzed; **** = P ≤ 0.0001, ** = P ≤ 0.01, * = P ≤ 0.05, ns = non-significant, Kruskal-Wallis test followed by Dunn's test). (**e**) Chromatin compaction can be followed in replicating (PCNA positive) and non-replicating (PCNA negative) cells in which a stripe of photo-activable GFP-H2A has been activated. Adding HU and/or UNC0642 immediately after the activation of GFP-H2A allow to measure over time the impact of these drugs on chromatin compaction. Figure created with biorender.com. Source numerical data are available in source data.

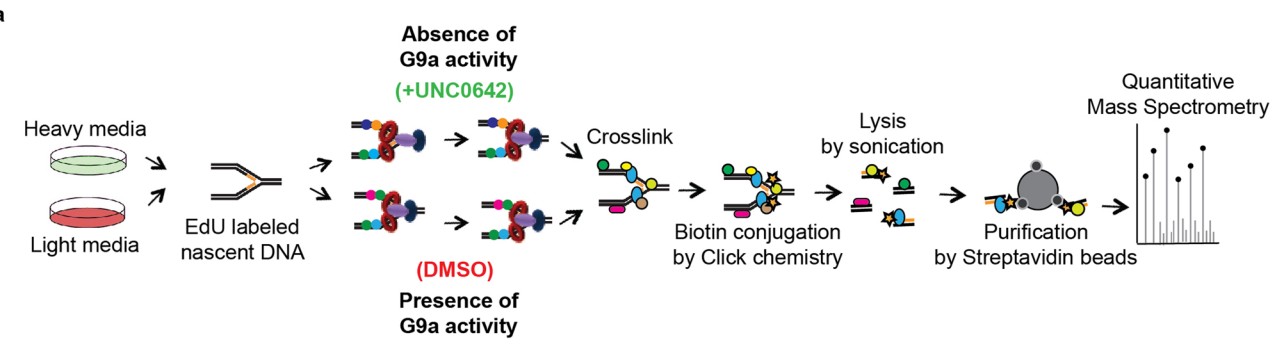

**a**

Heavy media

Light media

EdU labeled nascent DNA

Absence of G9a activity (+UNC0642)

Presence of G9a activity (DMSO)

Crosslink

Biotin conjugation by Click chemistry

Lysis by sonication

Purification by Streptavidin beads

Quantitative Mass Spectrometry

**b**

iPOND: UT / UT+G9ai

proteins enriched upon G9a inhibition >0.2

unchanged

proteins depleted upon G9a inhibition < -0.2

$Log_2$ difference

PRIM1, BAZ1B, POLA1, POLD2, FEN1, WIZ, SUPT16, LIG3, SMARCAD1, PCNA, MCM2, PRIM2, RFC5, CHAF1A, GINS1, RFC3, H1, POLA2, RFC2, CHAF1B, H3, TOP1, MCM7, MMS22L, POLR1B, CDK1, MCM6, DNMT1, UHRF1, POLR1E, HAT1, H2A, GINS4, POLD1, RFC4, POLE, MCM5, MCM3, MCM4, POLR2L, POLE2, ASF1a, ASF1b

**c**

H2AK15Ub-EdU (PLA spot intensity (a.u.))

ns
****
****    ****
****    **    ****

UT HU Rel | UT HU Rel
WT | UNC0642

**d**

H4K20me0-EdU (PLA spot intensity (a.u.))

*
****    ns
****    ****    ****    ****

UT HU Rel | UT HU Rel
WT | UNC0642

**e**

GeneType: Dysregs, DDR

UNC0642 vs WT

abs(log2FC)

T-test, P<2.2e$^{-16}$
N=95    N=179

Dysregs    DDR
GeneType

UNC0642+HU vs WT+HU

abs(log2FC)

T-test, P<2.2e$^{-16}$
N=57    N=179

Dysregs    DDR
GeneType

**f**

PCNA - EdU (PLA spot intensity (a.u.))

**
****    ns
****    ****    ****    ****

UT HU Rel | UT HU Rel
WT | UNC0642

**g**

RPA - EdU (PLA spot intensity (a.u.))

*
****    ns
****    ****    ****    ****

UT HU Rel | UT HU Rel
WT | UNC0642

**Extended Data Fig. 6 | See next page for caption.**

**Article** https://doi.org/10.1038/s41556-023-01167-z

**Extended Data Fig. 6 | Extended Data Figure. 6 related to Fig. 4. Chromatin landscape of active replication forks remains unaltered in absence of G9a activity.** (**a**) Schematic representation of the iPOND-SILAC-MS experiment comparing the protein present at replication fork when G9a is active (DMSO) vs when G9a is inactive (+UNC0642). This comparison was done at stalled replication fork (Fig.4a) or ongoing replication fork (Extended Data Fig. 6b). (**b**) Diagram showing a selection of the protein that are enriched (shades of green) or depleted (shades of red) at ongoing replication fork in the absence of G9a activity. Proteins considered enriched when $\log2$ (ratio of UT + G9a inhibition/ UT) $\geq 0.2$. Proteins considered depleted when $\log2$ (ratio of UT+G9a inhibition/ UT) $\leq -0.2$. (**c**) Distribution of H2AK15Ub-EdU total PLA spot intensity per nucleus assessing the level of H2AK15Ub at replication sites for the indicated conditions. ($n_{WT\text{-}UT} = 1337$, $n_{WT\text{-}HU} = 1312$, $n_{WT\text{-}REL} = 1370$, $n_{UNC0642\text{-}UT} = 308$, $n_{UNC0642\text{-}HU} = 1408$, $n_{UNC0642\text{-}REL} = 1426$ cells analyzed; blue dashed line = mean of the distribution, **** = $P \leq 0.0001$, ** = $P \leq 0.01$, ns = non-significant, Kruskal-Wallis test followed by Dunn's test). (**d**) Distribution of H4K20me0-EdU total PLA spot intensity per nucleus assessing the level of H4K20me0 at replication sites for the indicated conditions. ($n_{WT\text{-}UT} = 1504$, $n_{WT\text{-}HU} = 1400$, $n_{WT\text{-}REL} = 1374$, $n_{UNC0642\text{-}UT} = 1358$, $n_{UNC0642\text{-}HU} = 1096$, $n_{UNC0642\text{-}REL} = 1210$ cells analyzed; blue dashed line = mean of the distribution, **** = $P \leq 0.0001$, * = $P \leq 0.05$, ns = non-significant, Kruskal-Wallis test followed by Dunn's test). (**e**) Fold change in transcript levels of dysregulated genes (red) and DNA damage repair (DDR) genes (blue) in wild type cells treated with UNC0642 normalized to untreated wild type cells. Left: In the absence of replication stress. Right: In the presence of replication stress (1 mM HU 1 hr), N = number of genes, unpaired two-sided t-test. (**f**) Distribution of PCNA-EdU total PLA spot intensity per nucleus assessing the level of PCNA at replication sites for the indicated conditions. ($n_{WT\text{-}UT} = 732$, $n_{WT\text{-}HU} = 628$, $n_{WT\text{-}REL} = 391$, $n_{UNC0642\text{-}UT} = 723$, $n_{UNC0642\text{-}HU} = 763$, $n_{UNC0642\text{-}REL} = 688$ cells analyzed; blue dashed line = mean of the distribution, **** = $P \leq 0.0001$, * = $P \leq 0.05$, ns = non-significant, Kruskal-Wallis test followed by Dunn's test). (**g**) Distribution of RPA-EdU total PLA spot intensity per nucleus assessing the level of RPA at replication sites for the indicated conditions. ($n_{WT\text{-}UT} = 1344$, $n_{WT\text{-}HU} = 1437$, $n_{WT\text{-}REL} = 1352$, $n_{UNC0642\text{-}UT} = 965$, $n_{UNC0642\text{-}HU} = 793$, $n_{UNC0642\text{-}REL} = 514$ cells analyzed; blue dashed line = mean of the distribution, **** = $P \leq 0.0001$, * = $P \leq 0.05$, ns = non-significant, Kruskal-Wallis test followed by Dunn's test). Source numerical data are available in source data.

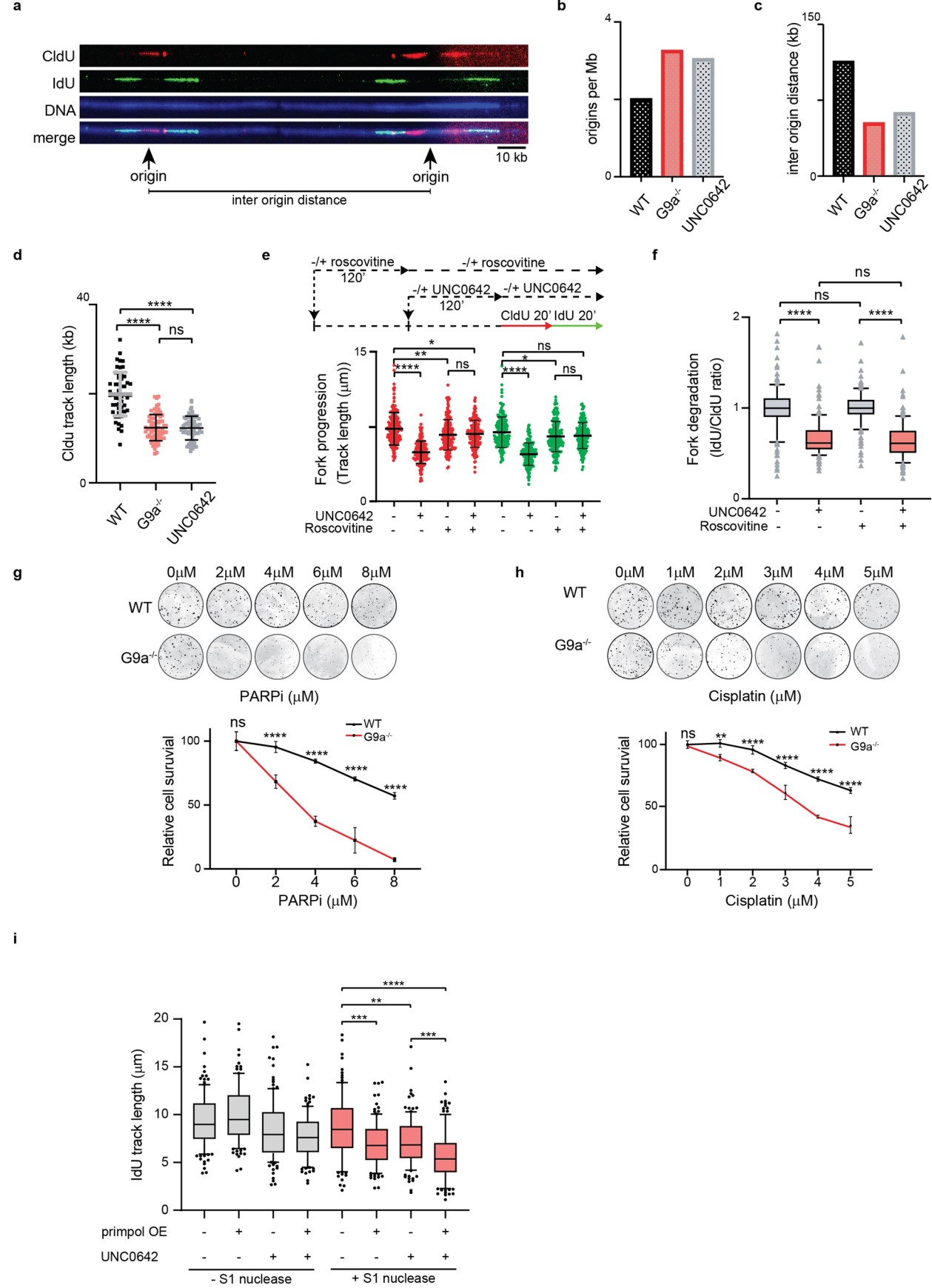

**Extended Data Fig. 7 | See next page for caption.**

**Extended Data Fig. 7 | Extended Data Figure. 7 related to Fig. 5. Loss of G9 activity causes genome instability.** (**a**) Representative image of a combed DNA molecule labelled with CldU and IdU. This experiment was independently reproduced two times with similar results. (**b**) Bar chart showing the number of origin of replication per megabase (Mb) of DNA analyzed in the indicated conditions. (**c**) Bar chart showing the average inter-origin distance in kilobase (Kb) in the indicated conditions. (**d**) Plot showing the distribution of CldU track length in Kb in the indicated conditions. Mean ± SD of the track length distribution is shown. ($n_{WT}$ = 55, $n_{G9aKO}$ = 69, $n_{UNC0642}$ = 65 CldU tracks; **** = P ≤ 0.0001, *** = P ≤ 0.001, ns = non-significant, One-way ANOVA Kruskal-Wallis test followed by Dunn's test). (**a**–**d**) This experiment was independently reproduced two times with similar results. (**e**) Top panel: Schematic of replication fork progression assay using CldU and IdU labeling. Bottom panel: CldU (red) and IdU (green) track length (μm) distribution for the indicated conditions. Mean ± SD of the track length distribution is shown. ($n_{UT}$ = 158, $n_{UNC0642}$ = 163, $n_{roscovitin}$ = 161, $n_{UNC0642+roscovitin}$ = 165 CldU and IdU tracks analyzed; **** = P ≤ 0.0001, ** = P ≤ 0.01, * = P ≤ 0.05, ns = non-significant, One-way ANOVA Kruskal-Wallis test followed by Dunn's test). (**f**) Fork degradation assay using DNA fiber methodology.

The distribution of the ratio of IdU to CldU track length (μm) was plotted for the given conditions. ($n_{UT}$ = 155, $n_{UNC0642}$ = 152, $n_{roscovitin}$ = 154, $n_{UNC0642+roscovitin}$ = 154 tracks analyzed; **** = P ≤ 0.0001, ns = non-significant, One-way ANOVA Kruskal-Wallis test followed by Dunn's test). (**g**, **h**) Representative images (top) and Quantification (bottom) of colony survival assay. Mean survival ± SD from 3 independent experiments in wild type (WT) and cells lacking G9a (G9a-/-) treated with different concentrations of olaparib (PARPi, **g**) or cisplatin (**h**) is shown. (**** = P ≤ 0.0001, ** = P ≤ 0.01, ns = non-significant, unpaired two-sided t-test). (**i**) Primpol was over-expressed in MRC-5 cells 48 h prior to the experiment and accumulation of ssDNA behind the replication forks upon primpol over-expression (primpol OE) and G9a inhibition (UNC0642) was assess using S1 nuclease. Right: IdU track length (μm) distribution for the indicated conditions. ($n_{UT\_S1-}$ = 120, $n_{PrimpolOE\_S1-}$ = 113, $n_{UNC0642\_S1-}$ = 120, $n_{PrimpolOE+UNC0642\_S1-}$ = 100, $n_{UT\_S1+}$ = 101, $n_{PrimpolOE\_S1+}$ = 110, $n_{UNC0642\_S1+}$ = 112, $n_{PrimpolOE+UNC0642\_S1+}$ = 114 tracks analyzed; **** = P ≤ 0.0001, *** = P ≤ 0.001, ** = P ≤ 0.01, * = P ≤ 0.05, ns = non-significant, Kruskal-Wallis test followed by Dunn's test). Source numerical data are available in source data.

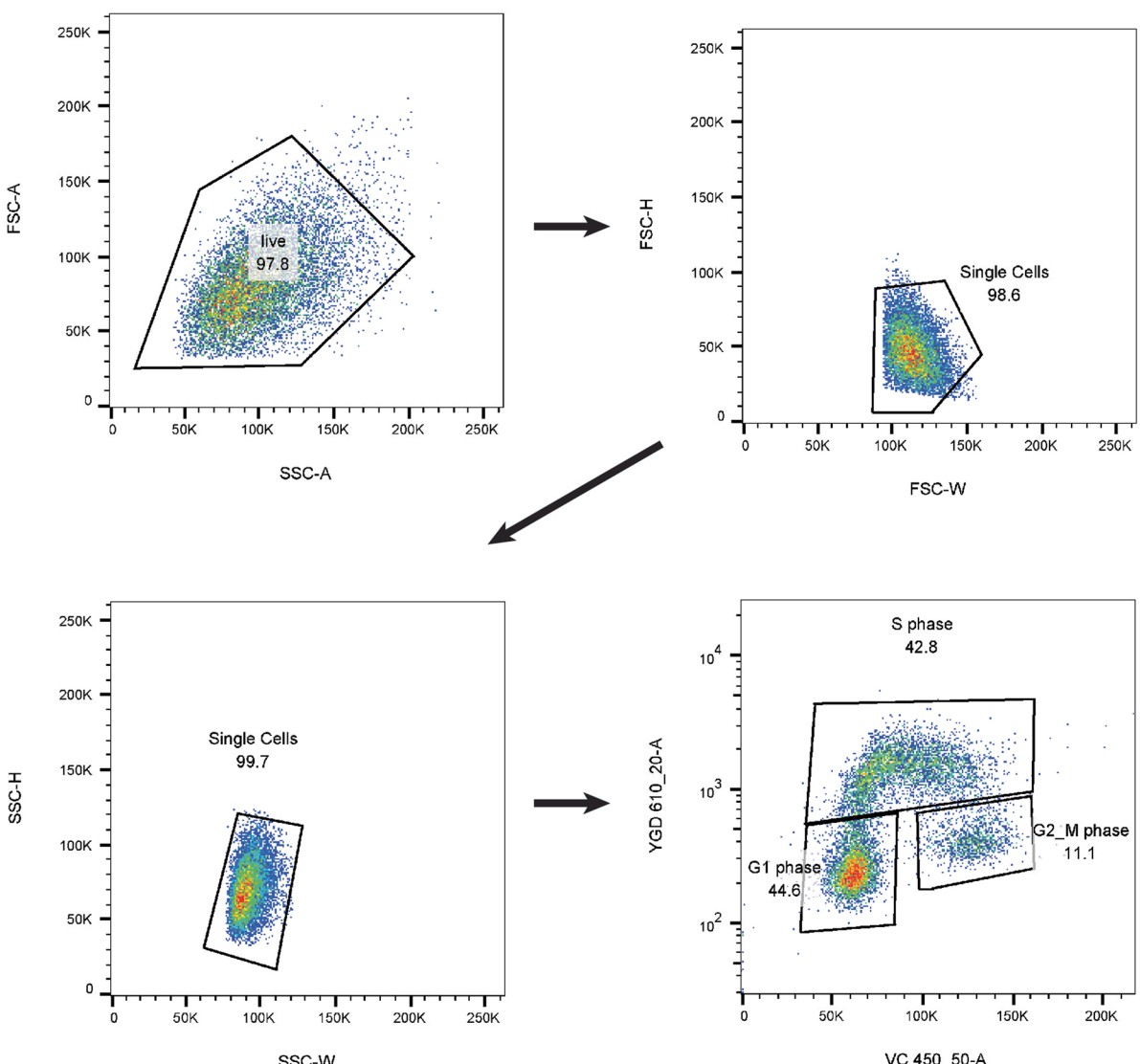

**Extended Data Fig. 8 | Extended Data Figure. 8 related to Extended Data Fig. 4. Flow Cytometry Gating Strategy.** Single nuclei were selected using FSC-A vs SSC-A, followed by FSC-H vs FSC-W and SSC-H vs SSC-W in the flow cytometry analysis. G1 phase, S phase and G2/M phase of the cell cycle were determined based on the intensity of the EdU and the DAPI channel.

# Reporting Summary

## Statistics

For all statistical analyses, confirm that the following items are present in the figure legend, table legend, main text, or Methods section.

| n/a | Confirmed | |
|---|---|---|
| ☐ | ☒ | The exact sample size (*n*) for each experimental group/condition, given as a discrete number and unit of measurement |
| ☒ | ☐ | A statement on whether measurements were taken from distinct samples or whether the same sample was measured repeatedly |
| ☐ | ☒ | The statistical test(s) used AND whether they are one- or two-sided<br>*Only common tests should be described solely by name; describe more complex techniques in the Methods section.* |
| ☐ | ☒ | A description of all covariates tested |
| ☒ | ☐ | A description of any assumptions or corrections, such as tests of normality and adjustment for multiple comparisons |
| ☐ | ☒ | A full description of the statistical parameters including central tendency (e.g. means) or other basic estimates (e.g. regression coefficient) AND variation (e.g. standard deviation) or associated estimates of uncertainty (e.g. confidence intervals) |
| ☐ | ☒ | For null hypothesis testing, the test statistic (e.g. *F*, *t*, *r*) with confidence intervals, effect sizes, degrees of freedom and *P* value noted<br>*Give P values as exact values whenever suitable.* |
| ☒ | ☐ | For Bayesian analysis, information on the choice of priors and Markov chain Monte Carlo settings |
| ☒ | ☐ | For hierarchical and complex designs, identification of the appropriate level for tests and full reporting of outcomes |
| ☐ | ☒ | Estimates of effect sizes (e.g. Cohen's *d*, Pearson's *r*), indicating how they were calculated |

*Our web collection on statistics for biologists contains articles on many of the points above.*

## Software and code

Policy information about availability of computer code

| Data collection | Illumina HiSeq2000, Metafer 5 , Leica ST5 confocal microscope |
|---|---|
| Data analysis | Graphpad Prism 9.4.1, imageJ 1.53t, Metasystem, R studio 1.2.5019, Flowjo v10.8.1, Spotfire Analyst 12.0, FastQC v0.11.7, Proteome Discoverer (version 2.5.0.400 ) |

For manuscripts utilizing custom algorithms or software that are central to the research but not yet described in published literature, software must be made available to editors and reviewers. We strongly encourage code deposition in a community repository (e.g. GitHub). See the Nature Portfolio guidelines for submitting code & software for further information.

## Data

Policy information about availability of data

All manuscripts must include a data availability statement. This statement should provide the following information, where applicable:
- Accession codes, unique identifiers, or web links for publicly available datasets
- A description of any restrictions on data availability
- For clinical datasets or third party data, please ensure that the statement adheres to our policy

Deep-sequencing (ChIP-seq, and RNA–seq) data that support the findings of this study have been deposited as a Bioproject under accession code PRJNA845122, for the RNA-sequencing data, and PRJNA897702, for the ChIP-sequencing data.
Mass spectrometry data have been deposited in ProteomeXchange with the primary accession codes PXD041742, for silac data and PXD041914 for the proteomics

analysis of histone PTM levels

The human ovarian cancer data analyzed in this study were from the TCGA datasets102,103 (https://link.springer.com/article/10.1007/s11357-023-00742-4/tables/1).

Source data have been provided in Source Data. All other data supporting the findings of this study are available from the corresponding author on reasonable request.

## Human research participants

Policy information about studies involving human research participants and Sex and Gender in Research.

| | |
|---|---|
| Reporting on sex and gender | N/A |
| Population characteristics | N/A |
| Recruitment | N/A |
| Ethics oversight | N/A |

Note that full information on the approval of the study protocol must also be provided in the manuscript.

# Field-specific reporting

Please select the one below that is the best fit for your research. If you are not sure, read the appropriate sections before making your selection.

☒ Life sciences  ☐ Behavioural & social sciences  ☐ Ecological, evolutionary & environmental sciences

For a reference copy of the document with all sections, see nature.com/documents/nr-reporting-summary-flat.pdf

# Life sciences study design

All studies must disclose on these points even when the disclosure is negative.

| | |
|---|---|
| Sample size | N/A. Sample size for samples was chosen according to or exceeding standards in the field |
| Data exclusions | N/A, no data exclusion |
| Replication | Experimental assays were performed two - three independent replicates, with similar results. |
| Randomization | This is not applicable, as our sample groups were grown under the same condition and collect randomly when given treatment without any bias. |
| Blinding | Blinding was not needed as data is collected by imaging software which yield unbiased, objective measurments |

# Reporting for specific materials, systems and methods

We require information from authors about some types of materials, experimental systems and methods used in many studies. Here, indicate whether each material, system or method listed is relevant to your study. If you are not sure if a list item applies to your research, read the appropriate section before selecting a response.

## Materials & experimental systems

| n/a | Involved in the study |
|---|---|
| ☐ | ☒ Antibodies |
| ☐ | ☒ Eukaryotic cell lines |
| ☒ | ☐ Palaeontology and archaeology |
| ☒ | ☐ Animals and other organisms |
| ☒ | ☐ Clinical data |
| ☒ | ☐ Dual use research of concern |

## Methods

| n/a | Involved in the study |
|---|---|
| ☐ | ☒ ChIP-seq |
| ☐ | ☒ Flow cytometry |
| ☒ | ☐ MRI-based neuroimaging |

## Antibodies

| | |
|---|---|
| Antibodies used | Anti-BrdU (Clone B44)(347580, BD Bioscience),<br>Anti-Chk1, (G4) (sc8408, Santa Cruz Biotechnology), |

Anti-Chk1 (DCS-310,(Sørensen et al. 2003))
Anti-BARD1(A300-263A, Bethyl),
BRCA1 (D-9)(SC6954, Santa Cruz Biotechnology),
Anti-RPA32/RAP2 [9H8] (Ab2175, Abcam),
Anti-BrdU [BU1/75 (ICR1)] ( ab6326, Abcam),
Anti-PCNA [PC10] (ab29, Abcam),
Anti-H3K9me1 [EPR16989] (Ab176880, Abcam),
Anti-H3K9me2 (Ab1220, Abcam),
Anti-H3K9me3 [EPR16601](Ab176916, Abcam),
Anti-G9a [EPR18894](Ab 185050, Abcam), ),
Anti-H4K20me0 [EPR22116] (Ab227804, Abcam),
Anti-H4K16ac [EPR1004] (Ab109463, Abcam),
Anti-Phospho-Chk1 (Ser345) (133D3)(#2348, Cell Signaling),
Anti-H3 (Ab1791, Abcam),
Anti-H3 [1B1B2] (Ab195277, Abcam),
Anti-HDAC1 (Ab19845, Abcam),
Anti-RAD51 (70-002, Bio Academia),
Anti-H2AK15ub (EDL H2AK15-4) (MABE1119, Millipore),
Anti-γ H2AX (clone JBW301)(05-636, Millipore),
Anti-phospho-H3S10 (06-570, Millipore),
Anti-H3K9me3 (Ab8898, Abcam),
Anti-H3 (Ab10799, Abcam),
Anti-H3K9me3 (07-442, Millipore),
Anti-H3K9me1 (Upstate, 07-450),
Anti-p53-S15p (D4S1H) (12571, Cell Signaling),
Anti-p53 (clone DO-1) (MABE327, Sigma-Aldrich),
Anti-b-actin (clone AC-74) (A5316, Sigma-aldrich),
Anti-ssDNA antibody (AB_10805144, DSHB)
Anti-Biotin antibody (A150-109A, Bethyl Laboratories)
Anti-Biotin antibody (AB_2339006, JacksonImmunoResearch)

| Validation | All antibodies used are commercially available and have been validated by the manufacturers. |
|---|---|

# Eukaryotic cell lines

Policy information about cell lines and Sex and Gender in Research

| Cell line source(s) | MRC5 sv40 immortalized human fibroblast and mESCs were generated in Nitika Taneja's lab (Lo et al, Science Advances, 2021)<br>Stable TIG-3 human fibroblast was generated in Anja Groth's lab (Alabert et al, Genes Dev, 2015) |
|---|---|
| Authentication | None of the cell line used were authenticated |
| Mycoplasma contamination | cell lines are constantly tested for mycoplasma contamination, and were all negative. |
| Commonly misidentified lines<br>(See ICLAC register) | No commonly misidentified cell lines were used in the study |

# ChIP-seq

## Data deposition

☒ Confirm that both raw and final processed data have been deposited in a public database such as GEO.

☒ Confirm that you have deposited or provided access to graph files (e.g. BED files) for the called peaks.

| Data access links<br>*May remain private before publication.* | PRJNA897702 |
|---|---|
| Files in database submission | PRI_D43J_1_0_H3K9_GCCAAT_L006_R1.fastq.gz, PRI_D43J_2_0_H3K9_CTTGTA_L006_R1.fastq.gz,<br>PRI_EETD_1_plus_H3K9_GCCAAT_L008_R1.fastq.gz, PRI_EETD_2_plus_H3K9_CTTGTA_L008_R1.fastq.gz,<br>PRI_QNRF_1_minus_H3K9_GCCAAT_L007_R1.fastq.gz, PRI_QNRF_2_minus_H3K9_CTTGTA_L007_R1.fastq.gz,<br>PRI_CDKD_1_0_H3_GCCAAT_L005_R1.fastq.gz, PRI_CDKD_2_0_H3_CTTGTA_L005_R1.fastq.gz,<br>PRI_FLC4_1_plus_H3_GCCAAT_L007_R1.fastq.gz, PRI_FLC4_2_plus_H3_CTTGTA_L007_R1.fastq.gz,<br>PRI_AXQA_1_minus_H3_GCCAAT_L006_R1.fastq.gz, PRI_AXQA_2_minus_H3_CTTGTA_L006_R1.fastq.gz, |
| Genome browser session<br>(e.g. UCSC) | https://genome.ucsc.edu/s/nazaret/Gaggioli%20et%20al_2022_initial_submission |

## Methodology

| Replicates | 2 |
|---|---|

| Sequencing depth | Provided as a separate file"Gaggioli et al_Sequencing depth information.pdf" |
|---|---|
| Antibodies | Anti-H3K9me3 (Ab8898, Abcam), Anti-H3 (Ab10799, Abcam) |
| Peak calling parameters | During analysis of the ChIP seq experiments, peak calling was performed according to the following parameter: Peak detection was performed with MACS2 version 2.0.9 (20111102) using default settings except for parameters '--broad --nomodel --shiftsize=110'. The shift size of 110 bp was calculated as the median over all Phantom Peak. |
| Data quality | FASTQC, FASTQCScreen |
| Software | ChIP-seq data are available at the Gene Expression Omnibus (GEO) (PRJNA897702). Raw reads were aligned to the human genome (hg19 assembly excluding non-canonical chromomes i.e. random, unknown and haplotype variant chromosomes) using Bowtie version 0.12.7 with default parameters except '-S -m 1', which excludes reads mapping to multiple chromosomal positions. Bigwig files were generated using the UCSC Kent utilities (Kent et al. 2010).. We allowed only one read per chromosomal position thus eliminating potential spurious spikes, and each remaining read was extended from its 5'-end to a total length of 250 bases, before converting to bedGraph format, scaling to mapped reads per million and final convertion to bigwig format. Individual BigWig files were uploaded to the UCSC browser for visualization(Kent et al. 2002;2010). To generate chromosome-wide landscapes of H3K9me3 and H3 we used the mean as the combining function and a smoothing window of 4 pixels |

## Flow Cytometry

### Plots

Confirm that:

☒ The axis labels state the marker and fluorochrome used (e.g. CD4-FITC).

☒ The axis scales are clearly visible. Include numbers along axes only for bottom left plot of group (a 'group' is an analysis of identical markers).

☒ All plots are contour plots with outliers or pseudocolor plots.

☒ A numerical value for number of cells or percentage (with statistics) is provided.

### Methodology

| Sample preparation | Cells were grown to 70–80% confluency in a 10cm culturing dish. Cells were labeled with EdU for 30minutes followed by fixation for 10minutes in 4% formaldehyde in PBS at room temperature. Cells were then washed with 1% BSA/PBS and permeabilized in 0.5% saponin buffer in 1% BSA/PBS. Incorporated EdU were labelled with the click-it reaction using Alexa Fluor® 594 azide according to the manufacturer's protocol (Invitrogen). DAPI was used to stain the DNA. |
|---|---|
| Instrument | BD LSRFortessa Cell Analyzer |
| Software | Flowjo v10.8.1 |
| Cell population abundance | The single cells population after SSC and FSC gating was around 60%-90% of the total event. |
| Gating strategy | Gating was done using SSC-A vs FSC-A, followed by FSC-H vs FSC-W and SSC-H vs SSC-W to select single nuclei. |

☒ Tick this box to confirm that a figure exemplifying the gating strategy is provided in the Supplementary Information.

