## [Peer Review File · Nature Cell Biology]

Peer Review Information

Journal: Nature Cell Biology

Manuscript Title: Dynamic de novo heterochromatin assembly and disassembly at replication forks ensures fork stability

Corresponding author name(s): Dr Nitika Taneja

Editorial Notes:

Reviewer Comments & Decisions:

Decision Letter, initial version:

*Please delete the link to your author homepage if you wish to forward this email to co-authors.

Dear Dr Taneja,

Thank you for submitting your manuscript, "Dynamics of de novo heterochromatin assembly and disassembly at replication forks ensures fork stability", to Nature Cell Biology and thank you for your patience with the peer review process. It has now been seen by 3 referees, who are experts in heterochromatin, DNA replication (Referee #1); heterochromatin, H3K9 methylation (Referee #2); and DNA replication stress (Referee #3). As you will see from their comments (attached below), they found this work of potential interest but have raised substantial concerns, which in our view would

need to be addressed with considerable revisions before we can consider publication in Nature Cell Biology.

As per our standard editorial process, we have now discussed the referee reports in detail within the editorial team, including the chief editor, to identify key referee points that should be addressed with priority, as opposed to requests that are overruled as being beyond the scope of the current study. To guide the scope of the revisions, I have listed these points below. Our standard revision period is six months, and we are committed to providing a fair and constructive peer-review process, so please feel free to contact me if you would like to discuss any of the referee comments further or if you anticipate any delays or issues addressing the reviews.

In particular, it would be essential to:

- more comprehensively analyze the changes in histone marks and carry out further analyses guided by Rev#1 point #1 on H3K9me2 and Rev#3 points #3-4-5 on H3K36me2 and H3K9me1 and me2. Please also address Rev#3's concerns about the ChromStretch data detecting overlap between H3K9me3 and EdU signals (point #2).
- address the reviewers' questions and suggestions around the enzymes involved with adequate discussion and edits at a minimum (Rev#1 point #2; Rev#2 points #1, #3)
- strengthen the model linking heterochromatin formation to PrimPOL, addressing Rev#3's suggestion in point #8 to test whether PrimPOL overexpression increases ssDNA gaps under HU conditions with or without G9a inhibition.
- All other referee concerns pertaining to strengthening existing data, providing controls, methodological details, clarifications and textual changes, should also be addressed.
- Finally, please pay close attention to our guidelines on statistical and methodological reporting (listed below) as failure to do so may delay the reconsideration of the revised manuscript. In particular please provide:
 - a Supplementary Figure including unprocessed images of all gels/blots in the form of a multi-page pdf file. Please ensure that blots/gels are labeled and the sections presented in the figures are clearly indicated.
 - a Supplementary Table including all numerical source data in Excel format, with data for different figures provided as different sheets within a single Excel file. The file should include source data giving rise to graphical representations and statistical descriptions in the paper and for all instances where the figures present representative experiments of multiple independent repeats, the source data of all repeats should be provided.

We would be happy to consider a revised manuscript that would satisfactorily address these points, unless a similar paper is published elsewhere, or is accepted for publication in Nature Cell Biology in the meantime.

In contrast, although we agree with Rev#2 that inducing replication stress in a site-specific manner

would provide valuable insights, we also appreciate that it may be technically challenging and thus, in our view, addressing it experimentally will not be strictly necessary for reconsideration of the manuscript at this journal.

- ensure that it conforms to our format instructions and publication policies (see below and <https://www.nature.com/nature/for-authors>).
- provide a point-by-point rebuttal to the full referee reports verbatim, as provided at the end of this letter.
- provide the completed Reporting Summary (found here <https://www.nature.com/documents/nr-reporting-summary.pdf>). This is essential for reconsideration of the manuscript will be available to editors and referees in the event of peer review. For more information see <http://www.nature.com/authors/policies/availability.html> or contact me.

When submitting the revised version of your manuscript, please pay close attention to our [href="https://www.nature.com/nature-portfolio/editorial-policies/image-integrity">Digital Image Integrity Guidelines](https://www.nature.com/nature-portfolio/editorial-policies/image-integrity). and to the following points below:

Nature Cell Biology is committed to improving transparency in authorship. As part of our efforts in this direction, we are now requesting that all authors identified as 'corresponding author' on published papers create and link their Open Researcher and Contributor Identifier (ORCID) with their account on the Manuscript Tracking System (MTS), prior to acceptance. ORCID helps the scientific community achieve unambiguous attribution of all scholarly contributions. You can create and link your ORCID from the home page of the MTS by clicking on 'Modify my Springer Nature account'. For more information please visit www.springernature.com/orcid.

This journal strongly supports public availability of data. Please place the data used in your paper into a public data repository, or alternatively, present the data as Supplementary Information. If data can only be shared on request, please explain why in your Data Availability Statement, and also in the correspondence with your editor. Please note that for some data types, deposition in a public repository is mandatory - more information on our data deposition policies and available repositories appears below.

[Redacted]

We hope that you will find our referees' comments and editorial guidance helpful. Please do not hesitate to contact me if there is anything you would like to discuss. Thank you again for considering NCB for your work.

Best wishes,

Melina

Melina Casadio, PhD
Senior Editor, Nature Cell Biology
ORCID ID: <https://orcid.org/0000-0003-2389-2243>

Reviewers' Comments:

Reviewer #1:

Remarks to the Author:

The submitted manuscript is an excellent piece of work that explores the changes in H3K9methylation that accompany replication stress. The authors see an increase in H3K9methylation (and no increase in other abundant histone modifications) and trace it to the upregulation and recruitment of G9a, which primarily deposits H3K9me1 and me2. They document in part the impact of losing this modification during replication stress, and suggest that chromatin condensation is triggered by the widespread methylation of H3K9. The methylation and its removal support efficient fork restart. Overall the paper makes a significant and original contribution to understanding chromatin changes in response to replication stress and the replication checkpoint.

There are a few points that I think must be addressed before acceptance.

1. G9a – as shown invitro and in vivo – primarily deposits H3K9me2. The di-methyl can be modified to trimethyl by SETDB1 or Suv39H1 or H2, but H3K9me2 is sufficient to bind HP1 (as tightly as H3K9me3) and it has been shown to silence genes in other organisms. Thus it is surprising that the authors only show the data for H3K9me3 in figures 1d, 1e, Figure 2, and extended data Figure 2a. I think the paper needs to show the changes in H3K9me2 as well, especially as they implicate G9a as the key modifier responsible for the HU sensitivity. They do show in extended figure 2c that H3K9me2 appears to be more abundant than me3 and I assume they have antibodies that recognize H3K9me2 specifically. Why is it not shown (other than in ext. data figure 1, where it is obvious that H3K9me2 is

- as increased as me3 albeit not always at the same sites). What distinguishes regions that acquire H3K9me3 vs H3K9me2 ? It may be repeat elements that drive the dimethyl to tri methyl, and this would be important to show. Both H3K9me1, and H3K9me2 show strong G9a-dependent colocalization with DNA synthesis signals in a manner that is even more pronounced than H3K9me3 (Figure 2e).
2. Just because they detect some H3K9me3 in the absence of SUV39H1, this does not mean that G9a is depositing me3. It may be SUV39H2 or SETDB1. Thus the strong statement that G9a may itself trimethylated should be softened.
 3. The authors use Parp inhibitors and cis-platin treatment to argue that loss of G9a affects survival of replication stress. These results are somewhat dubious (figure 5d). The G9a deficient strain undoubtedly also misregulates a lot of genes (G9a is the primary H3K9HMT that targets genes in differentiated cells). Thus the hypersensitivity to PARPi or cis-platin may not be due to the absence of H3K9me2 at sites of damage. Moreover, PARPi is not known to primarily cause replication stress, since it blocks many other types of repair (notably BER, ss nick or ds break repair). Cis-platin also provokes a complicated repair pathway that depends on base excision, and not simply "fork arrest". Thus the conclusions, while significant, are not as clearly indicative of a role for H3K9me at sites of replication stress, as the authors state.
 4. The argument that H3K9me mediates compaction at sites of replication stress is even less compelling (Figure 3h). It is fine to include it but I would put this data as extended data, as I do not find it compelling. Better would be to do ATAC-seq or DAM-methylase accessibility, to show that the absence of H3K9me at sites of replication arrest are less compact.
 5. Finally, the IPond data and implication of BRCA1 in the fork degradation events is intriguing. The authors should cite earlier data showing synthetic lethality of Brca1 and Bard1 with loss of H3K9me2 in *C. elegans* as it reinforces the current observation (Padeken 2019, gad.322495.118v1).

The paper has so much data in it, that removal of some of the less compelling results (Parpi synergy, compaction assay) and inclusion of H3K9me2 (with revision of the text if the me2 and me3 diverge in localization) would streamline and strengthen the conclusions made.

Reviewer #2:

Remarks to the Author:

In this manuscript, Gaggioli et al. describe changes in chromatin structure at stressed replication forks. Their main findings include (1) histone H3K9 methyltransferase EHMT2/G9a assemble heterochromatin at stressed replication forks in a checkpoint-dependent manner, (2) G9a works together with another H3K9 methyltransferase SUV39H1 to induce histone H3K9me1/me2/me3 at stressed replication forks, (3) this leads to a closed conformation that stabilizes stalled replication forks and is opposed by the KDM3A H3K9 demethylase, and (4) heterochromatin disassembly by KDM3A enables PRIMOL access and replication restart. They also provide evidence that G9a^{-/-} cells are more sensitive to chemotherapeutic drugs and that this is partially rescued by depletion of KDM3A in G9a^{-/-} cells.

The findings are novel and interesting and provide new insight into the role of H3K9me and chromatin dynamics in protection of stressed replication forks. The main conclusions of the study are further supported by a series of thoughtful and well-executed experiments. The manuscript is suitable for publication in *Nature Cell Biology*.

I only have minor comments for the authors to consider.

1. The authors focus on KDM3A which is understandable based on their iPOND results. I am curious whether the other major H3K9 demethylases particularly KDM4A, B, C also play roles in stalled fork restart. KDM4B has been shown to be critical for the repair of site-specific double strand DNA breaks which also involves H3K9 recruitment (PMID: 32494005). The results in this study are also relevant to the authors findings and they may wish to discuss them.
2. The authors study is based on induction of global replication stress using hydroxy urea. They should consider inducing replication stress in a site-specific manner to test whether they obtain similar results. I would understand if this were beyond the scope of their present study but it may be achievable using dCas9 and/or engineering binding sites for protein shown to block replication fork progression (such as the budding yeast Fob1 protein).
3. Could the authors hypothesize why both SUV39H1 and G9a are required since SUV39H1 can catalyze all three modification states?
4. Some of the experimental methods need to be more fully described, either in the results section or in the methods, for example the rationale for using CIdu and IdU to examine fork restart rates. I don't think either thymidine analog is described anywhere in the paper.

Altogether, this study presents a wealth of interesting results regarding changes in proteins bound to stressed replication forks and should be of great interest to the field.

Reviewer #3:

Remarks to the Author:

This is an interesting and nicely executed study showing the action of the histone methyl-transferase EHMT2/G9a in heterochromatin formation at stressed replication forks. After realizing of the high density of repressive epigenetics marks in cancer cells, whose hallmark is replication stress, the authors explored whether there was a correlation between both events, to show that after HU treatment cells increased the levels of H3K9me3. ChIP-seq analysis of this mark revealed a genome-wide re-distribution of H3K9me3 upon replication stress (HU) that the authors attribute to heterochromatinization at sites of replication fork stalls. Using super-resolution STED microscopy authors found a significant overlap between H3K9me3 and replication sites in the presence of HU, that was not observed in their absence. Then, using an improved ChromStretch technique, the authors observed the accumulation of H3K9me3 at stalled replication forks in HU, a result that was extended for H3K9me1 and H3K9me2. To determine a cause-effect relationship of these epigenetic marks and histone methyl-transferases, authors show, by using G9a KO cells and a catalytic inhibitor of G9a, a drastic reduction in methylation of H3K9. Further analyses by iPOND and SILAC allowed them to identify a number of factors enriched or reduced at replication forks that were corroborated by PLA. Among the lost ones are known fork-protection factors whereas among the enriched factors are FANCD2, RAD51, MCM, SMARCAL1, PrimPOL or the KDM3A demethylase. In the last part of the study the authors show that under conditions lacking H3K9 methylation DNA replicates faster and leaves stretched of ssDNAs that are consistent with a restart by PrimPOL. In agreement, forks are degraded (not protected) and re-start is slow supporting that H3K9methylation is essential for fork protection and also for fork re-start.

Altogether, they conclude that fork protection involves transient chromatin compaction at forks.

The study is solid and provides a rational and convincing model on the methylation of histone H3K9 as a mark of chromatin condensation. Authors propose that this is a transient process, as it would be reverted by the JMJD1A/KDM3A demethylase, that is associated with stressed replication forks, distinct to those previously shown for ATRX/DAXX, as in this study the condensation is checkpoint dependent in contrast to that of ATRX. It is a sound study that deserves publication and will be of great interest in the field of genome integrity and replication stress. Few suggestions and questions are indicated below that may serve to strengthen the conclusions and clarify some points.

Specific comments:

1. The experiments of colocalization of EdU and H3K9me3 have the caveat of not being a single fork (as they discuss) but also, they may identify ongoing forks and not just stalled forks. Upon nucleotide depletion, forks stall and this causes the firing of extra dormant or late origins (see for instance PMID: 12914702). Thus, one possibility not to be dismissed is that new origins in heterochromatin regions fire and this would explain the colocalization with H3K9me3. Discuss.
2. The ChromStretch technique reveals an overlap between H3K9me3 and EdU signals specifically when cells are treated with HU. However, the intensity of H3K9me3 is not further increased respect to the intensity of H3 (Figure 3g). This could mean that the increase of methylation detected is just a consequence of higher histone levels. Since these are chromatin fibers, it is important to clarify the meaning of higher histone intensity. Could it mean that there is more signal per DNA length or that fibers are not stretched enough in that region? Can authors rule out this last possibility? This is relevant, because if that was the case, the observations could be interpreted by a different structure of the chromatin fiber that impairs stretching rather than an accumulation of certain mark. In this regard, it does not look like that the genome-wide experiment is informative since the signals detected are not quantitative and the distribution profiles of total H3 look quite similar with or without replicative stress.
3. Page 3. Line 25. It is not clear why H3K9me is looked at specifically from the beginning. It seems that the reason is that previous studies that have reported an increment in H3K9me3 levels upon oncogene-induced replicative stress conditions (stated in line 44) and because they have previously reported higher levels of H3K9me1 upon replication stress (ref 39) rather than because of the increase in cancer cell lines (Extended Data Fig 1 A). Indeed, Extended Data Fig 1 A shows the analysis of ovarian cancers but experiments are then performed in TIG3 cells (lung fibroblasts). What happens in other cancers? Why are these cells used? It is also not clear or explained why H3K9 methylation levels are looked together with H3K14ac0. Clarify.
4. Page 4. Line 5. The observed increase in H3K36me2 that is detected is not discussed or studied any further. Why? Is it also detected in cancer databases?
5. Page 5. Line 45. Were H3K9me1 and me2 increased levels not detected at the masspec analysis upon HU? In that case, why?
6. Extended Data Fig. 3g-h should be in main figures as these data are crucial to state the dependency on the DNA replication checkpoint and to draw ATR in the model in Figure 3f and the final model.
7. Page 6. Line 34. The model includes HDAC1 and deacetylation of lysine 16 on histone H4 but this is

not based on any data from this manuscript. The author state here that 'replication fork restart alleviated these effects (Extended Data Fig. 4c-d)' but the effects only refer to H3K9me. It is not clear why the authors add HDAC1 acetylation to their model without experiments that support this role.

8. Authors propose that heterochromatin facilitates the loading of fork protection factors whereas it reduces that of PrimPOL and other proteins that may compromise genome integrity. This is a case in which PrimPOL is suggested to play a mutagenic role due to its potential to leave unreplicated ssDNA stretches behind the fork. Would PrimPOL overexpression increase ssDNA gaps under HU conditions and/or under G9a inhibition?

Minor comments:

- Page 6. Line 25. The authors state that 'combining knock down of Suv39h1 with inhibition of G9a did not significantly increase the accumulation defect of H3K9me3, suggesting that both G9a and Suv39h1 act in the same pathway (Fig. 3e).' However, the defect is already suppressed at all in both single knock down and inhibition conditions (and of course in the combination of both) and therefore it is difficult to draw any genetic conclusion regarding the pathways.
- Page 5. Line 30. Figure 2e is cited before 2d.
- Figure 4d-e axis legends are misspelled.

AUTHOR AFFILIATIONS – should be denoted with numerical superscripts (not symbols) preceding the

names. Full addresses should be included, with US states in full and providing zip/post codes. The corresponding author is denoted by: "Correspondence should be addressed to [initials]."

Methods should be written concisely, but should contain all elements necessary to allow interpretation and replication of the results. As a guideline, Methods sections typically do not exceed 3,000 words. The Methods should be divided into subsections listing reagents and techniques. When citing previous methods, accurate references should be provided and any alterations should be noted. Information must be provided about: antibody dilutions, company names, catalogue numbers and clone numbers for monoclonal antibodies; sequences of RNAi and cDNA probes/primers or company names and catalogue numbers if reagents are commercial; cell line names, sources and information on cell line identity and authentication. Animal studies and experiments involving human subjects must be reported in detail, identifying the committees approving the protocols. For studies involving human

subjects/samples, a statement must be included confirming that informed consent was obtained. Statistical analyses and information on the reproducibility of experimental results should be provided in a section titled "Statistics and Reproducibility".

All Nature Cell Biology manuscripts submitted on or after March 21 2016 must include a Data availability statement as a separate section after Methods but before references, under the heading "Data Availability". For Springer Nature policies on data availability see <http://www.nature.com/authors/policies/availability.html>; for more information on this particular policy see <http://www.nature.com/authors/policies/data/data-availability-statements-data-citations.pdf>. The Data availability statement should include:

- Accession codes for primary datasets (generated during the study under consideration and designated as "primary accessions") and secondary datasets (published datasets reanalysed during the study under consideration, designated as "referenced accessions"). For primary accessions data should be made public to coincide with publication of the manuscript. A list of data types for which submission to community-endorsed public repositories is mandated (including sequence, structure, microarray, deep sequencing data) can be found here <http://www.nature.com/authors/policies/availability.html#data>.
- Unique identifiers (accession codes, DOIs or other unique persistent identifier) and hyperlinks for datasets deposited in an approved repository, but for which data deposition is not mandated (see here for details <http://www.nature.com/sdata/data-policies/repositories>).
- At a minimum, please include a statement confirming that all relevant data are available from the authors, and/or are included with the manuscript (e.g. as source data or supplementary information), listing which data are included (e.g. by figure panels and data types) and mentioning any restrictions on availability.
- If a dataset has a Digital Object Identifier (DOI) as its unique identifier, we strongly encourage including this in the Reference list and citing the dataset in the Methods.

We recommend that you upload the step-by-step protocols used in this manuscript to the Protocol Exchange. More details can found at www.nature.com/protocolexchange/about.

All imaging data should be accompanied by scale bars, which should be defined in the legend. Cropped images of gels/blots are acceptable, but need to be accompanied by size markers, and to retain visible background signal within the linear range (i.e. should not be saturated). The boundaries of panels with low background have to be demarked with black lines. Splicing of panels should only be

considered if unavoidable, and must be clearly marked on the figure, and noted in the legend with a statement on whether the samples were obtained and processed simultaneously. Quantitative comparisons between samples on different gels/blots are discouraged; if this is unavoidable, it should only be performed for samples derived from the same experiment with gels/blots were processed in parallel, which needs to be stated in the legend.

The total number of Supplementary Figures (not including the “unprocessed scans” Supplementary Figure) should not exceed the number of main display items (figures and/or tables (see our Guide to Authors and March 2012 editorial <http://www.nature.com/ncb/authors/submit/index.html#suppinfo>; <http://www.nature.com/ncb/journal/v14/n3/index.html#ed>). No restrictions apply to Supplementary Tables or Videos, but we advise authors to be selective in including supplemental data.

GUIDELINES FOR EXPERIMENTAL AND STATISTICAL REPORTING

REPORTING REQUIREMENTS – We are trying to improve the quality of methods and statistics reporting in our papers. To that end, we are now asking authors to complete a reporting summary

that collects information on experimental design and reagents. The Reporting Summary can be found here <https://www.nature.com/documents/nr-reporting-summary.pdf>) If you would like to reference the guidance text as you complete the template, please access these flattened versions at <http://www.nature.com/authors/policies/availability.html>.

Author Rebuttal to Initial comments
--

Response to Reviewers' comments:

We thank the reviewers for their suggestions and encouraging remarks. In response to the reviewers' comments, we have thoroughly addressed the issues raised either by performing additional experiments or providing additional explanation. In doing so, we have further strengthened the evidence supporting our conclusions and we thank the reviewers for their help. The textual changes are highlighted in the revised manuscript.

Point-by-point response

Comments by the reviewers, which are indicated in blue, are addressed as follows:

Reviewer #1:

The submitted manuscript is an excellent piece of work that explores the changes in H3K9methylation that accompany replication stress. The authors see an increase in H3K9methylation (and no increase in other abundant histone modifications) and trace it to the upregulation and recruitment of G9a, which primarily deposits H3K9me1 and me2. They document in part the impact of losing this modification during replication stress, and suggest that chromatin condensation is triggered by the widespread methylation of H3K9. The methylation and its removal support efficient fork restart. Overall the paper makes a significant and original contribution to understanding chromatin changes in response to replication stress and the replication checkpoint.

We greatly appreciate the reviewer's positive comments.

1. " G9a – as shown invitro and in vivo – primarily deposits H3K9me2. The di-methyl can be modified to trimethyl by SETDB1 or Suv39H1 or H2, but H3K9me2 is sufficient to bind HP1 (as tightly as H3K9me3) and it has been shown to silence genes in other organisms. Thus it is surprising that the authors only show the data for H3K9me3 in figures 1d, 1e, Figure 2, and extended data Figure 2a. I think the paper needs to show the changes in H3K9me2 as well, especially as they implicate G9a as the key modifier responsible for the HU sensitivity."

As per the reviewer's suggestion, we have performed a thorough analysis, analogous that performed for H3K9me3 (**Fig. 2c-d**), of H3K9me2 dynamics upon replication stress using single molecule chromatin fibers. This analysis reveals dynamics of H3K9me2 accumulation resembling that of H3K9me3 with a gradual increase in H3K9me2 levels at replication forks upon replication stress. However, unlike the steeper shift in H3K9me3 signal from 30 mins to 1 hour HU treatment, an early saturation of signal is observed that is consistent with the hypothesis that H3K9me2, rather than serving as a terminal histone mark, represents a transient mark that is eventually converted into H3K9me3. This new observation supports our model that stepwise accumulation of H3K9 modifications from lower to higher states occurs upon replication stress. We have included this data in the revised manuscript as **Extended Data Fig. 4b**. Further, we have examined the dynamics of H3K9me2 removal from forks upon release from HU. Here we observed a significant reduction in H3K9me2 signal upon release from HU stress, mirroring the H3K9me3 reduction observed upon fork restart (**Fig. 2d**). We have also included this data in revised manuscript as **Extended Data Fig. 4c**.

"Why is it not shown (other than in ext. data figure 1, where it is obvious that H3K9me2 is as increased as me3 albeit not always at the same sites). What distinguishes regions that acquire H3K9me3 vs H3K9me2?.."

We thank the reviewer for their comment and apologize for not clearly articulating this. The data in **Extended Data Figure 1a** represents the total level of H3K9me1/me2/me3 modified peptides detected upon global chromatin profiling in different ovarian cancer cell lines, (data adapted from Ghandi et al, 2022; PMID: 31068700) as mentioned in the figure legend. We apologize for inadvertently omitting the names of cancer cell lines in the x-axis of the plot and we have corrected this figure by adding the names. This data implies that most of these cancer cell lines show increased levels of either all or some of the H3K9me modifications, but the plot does not represent the co-localization of different H3K9me modifications at any particular genomic locus.

Furthermore, the data from single molecule chromatin fibers showing the H3K9me signal at a single EdU labelled spot reveals a significant colocalization of EdU and H3K9me1, me2 and me3 signals (~80-90% in all cases) and similar dynamics of H3K9me3 or H3K9me2 accumulation during the time course of HU treatment. This suggests that these modifications co-occur at the given replication site (**Fig. 2c-e, new Extended Data Fig. 4b-f**). Due to technical issues, e.g. antibody origin and fixation method, we were unable to co-stain for different H3K9me marks in conjunction with EdU labelling. However, the significant similarity in colocalization events of each H3K9me modification with EdU implies that they co-occur at any given replicating site.

2. “Just because they detect some H3K9me3 in the absence of SUV39H1, this does not mean that G9a is depositing me3. It may be SUV39H2 or SETDB1. Thus the strong statement that G9a may itself trimethylated should be softened.”

We appreciate the reviewer’s suggestion and have changed the text accordingly in the revised manuscript.

3. “The authors use Parp inhibitors and cis-platin treatment to argue that loss of G9a affects survival of replication stress. These results are somewhat dubious (figure 5d). The G9a deficient strain undoubtedly also misregulates a lot of genes (G9a is the primary H3K9HMT that targets genes in differentiated cells). Thus the hypersensitivity to PARPi or cis-platin may not be due to the absence of H3K9me2 at sites of damage. Moreover, PARPi is not known to primarily cause replication stress, since it blocks many other types of repair (notably BER, ss nick or ds break repair). Cis-platin also provokes a complicated repair pathway that depends on base excision, and not simply “fork arrest”. Thus the conclusions, while significant, are not as clearly indicative of a role for H3K9me at sites of replication stress, as the authors state.”

We agree with the reviewer that although the *de novo* establishment of H3K9me by G9a plays a crucial role in maintaining replication fork stability upon replication stress and these effects can be alleviated by depleting demethylase KDM3 (Fig. 6a-f), we cannot rule out the possibility that the hypersensitivity towards replication-stress-inducing drugs could be a synergistic effect of the loss of replication fork stability, impaired DNA repair and misregulation of certain genes in the absence of G9a activity. Therefore, considering this data as supporting evidence of our finding that G9a helps to maintain replication fork stability, we have moved this data from **Fig. 5d-e to Extended Data Fig. 7g-h**.

4. “The argument that H3K9me mediates compaction at sites of replication stress is even less compelling (Figure 3h). It is fine to include it but I would put this data as extended data, as I do not find it compelling. Better would be to do ATAC-seq or DAM-methylase accessibility, to show that the absence of H3K9me at sites of replication arrest are less compact.”

Measuring chromatin accessibility in the vicinity of replication forks has been challenging and observing the compaction specifically upon replication stress even more so due to the limited number of replicating cells (40-45% S phase cells) and the dynamic aspect of DNA replication itself. Using single cell live imaging data of photoactivated H2A-GFP in combination with the expression of an S-phase cell marker, PCNA-GFP, we have shown chromatin compaction in replicating cells undergoing replication stress which is dependent upon the histone methyltransferase activity of G9a that establishes a platform of H3K9me3, essential for heterochromatin formation. Due to the single cell resolution, we are able to observe the compaction in replicating cells, an effect which otherwise would likely be diluted if observed in a pool of cells. We believe this is an important finding as the compaction could be a way to deny nucleases and other factors access to the otherwise highly exposed open structure of the replication fork. Nevertheless, we agree that future studies can benefit from the development of more sensitive methods to investigate the dynamics of heterochromatin formation at slowing / stalled replication forks at specific genomic sites. We have discussed the lack of proper methods to address these problems and have suggested some possible strategies in our revised manuscript.

As suggested by the reviewer, we have performed ATAC-Seq on replicating cells from two independent clones that endogenously express PCNA-GFP. The cells were FACS sorted to separate S phase cells from the non-S phase population. Not unexpectedly due to the pooling of cells, the resolution of nucleosome occupancy was insufficient to show significant changes upon replication stress. However, investigating chromatin accessibility at the transcription start sites (TSS) where the nucleosome free regions are easily resolved by the assay, we observed a small degree of compaction upon HU treatment in both the clones (see attached Figure A). We consider that the limited effect in this assay reflect that replication forks stall at random places in the genome, making population-based analysis of global chromatin unsuited. Repli-ATAC and MINCEseq were developed to track chromatin accessibility immediately post-replication, however as nascent chromatin is generally depleted for accessible regions these technologies are mainly useful to track how accessibility is restored post-replication when

Figure A. Trend plot showing the read intensity (in rpm) of the ATAC-Seq signal over 5 kb centered on the TSSs of all genes both untreated (UT), and HU treated (HU) replicating cells (PCNA-GFP positive) for 2 independent clones (clone1 and clone2)

transcription restarts (Stewart-Morgan et al, 2019; PMID: 31126739 and Ramachandran et al, 2016; PMID: 27062929).

5. *“Finally, the IPOND data and implication of BRCA1 in the fork degradation events is intriguing. The authors should cite earlier data showing synthetic lethality of Brca1 and Bard1 with loss of H3K9me2 in C. elegans as it reinforces the current observation (Padeken 2019, gad.322495.118v1).”*

We thank the reviewer for pointing out this study and apologize for missing it in our earlier version. We have cited this in the revised manuscript.

Reviewer #2:

“In this manuscript, Gaggioli et al. describe changes in chromatin structure at stressed replication forks. Their main findings include (1) histone H3K9 methyltransferase EHMT2/G9a assemble heterochromatin at stressed replication forks in a checkpoint-dependent manner, (2) G9a works together with another H3K9 methyltransferase SUV39H1 to induce histone H3K9me1/me2/me3 at stressed replication forks, (3) this leads to a closed conformation that stabilizes stalled replication forks and is opposed by the KDM3A H3K9 demethylase, and (4) heterochromatin disassembly by KDM3A enables PRIMOL access and replication restart. They also provide evidence that G9a^{-/-} cells are more sensitive to chemotherapeutic drugs and that this is partially rescued by depletion of KDM3A in G9a^{-/-} cells.

The findings are novel and interesting and provide new insight into the role of H3K9me and chromatin dynamics in protection of stressed replication forks. The main conclusions of the study are further supported by a series of thoughtful and well-executed experiments. The manuscript is suitable for publication in Nature Cell Biology..”

We greatly appreciate reviewer’s positive comments.

I only have minor comments for the authors to consider.

1. *The authors focus on KDM3A which is understandable based on their iPOND results. I am curious whether the other major H3K9 demethylases particularly KDM4A, B, C also play roles in stalled fork restart. KDM4B has been shown to be critical for the repair of site-specific double strand DNA breaks which also involves H3K9 recruitment (PMID: 32494005). The results in this study are also relevant to the authors findings and they may wish to discuss them.”*

We agree with the reviewer that, as in the case of HMTs, there may be redundancy in demethylases in order to accelerate the demethylation of H3K9 upon fork restart. The family of KDM4, especially member KDM4B, has been shown previously to play a critical role in DNA repair by demethylating H3K9me at DSBs. Unfortunately, we were unable to detect KDM4B specific peptides (i.e., Mascot score is 0) and/or changes of other family proteins above the set threshold (**Table S4**) in our iPOND-Mass spec of HU treatment in absence of G9 activity. In addition to KDM3A, we detected KDM5c which is involved in demethylating H3K4me marks. Therefore, we focused on KDM3A which is involved in demethylating methylated H3K9. Depleting KDM3A led to increased accumulation of H3K9me at forks and delayed fork restart defects and loss of KDM3A partly rescued ssDNA gap accumulation behind the fork upon G9a inhibition. These results suggest that the dynamics of maintaining H3K9me levels at forks are crucial to the process of maintaining fork stability. It is plausible that redundancy between these modifiers fine-tunes H3K9me levels at forks to maintain replication forks stability and the ability to restart forks in a timely manner via canonical restart mechanisms.

2. *“The authors study is based on induction of global replication stress using hydroxy urea. They should consider inducing replication stress in a site-specific manner to test whether they obtain similar results. I would understand if this were beyond the scope of their present study but it may be achievable using dCas9 and/or engineering binding sites for protein shown to block replication fork progression (such as the budding yeast Fob1 protein).”*

This is a great suggestion by the reviewer. In fact, our lab has been working to set up a method to induce replication stress in a site-specific manner using a lacO-lacI based approach. We have thus far confirmed that a H3K9me3 signal appears on almost all EdU labelled foci, and frequently on lacO marked with lacI-GFP foci undergoing replication mainly upon HU treated conditions in comparison to the untreated condition. However, due to the highly dynamic nature of the replication process, the opportunity to capture the chromatin dynamics established upon replication stress at a particular region of interest is available only in a limited population of cells. Therefore, this system is far from trivial and beyond what can be achieved within the scope of this current study.

3. “Could the authors hypothesize why both SUV39H1 and G9a are required since SUV39H1 can catalyze all three modification states?”

Although *in vitro* both Suv39h1 and G9a have been shown to catalyze all three modification states of H3K9me, earlier work has suggested that G9a mainly primes lower H3K9me modifications that primes for Suv39h1 to produce the H3K9me3 mark (Rice et al, 2003, PMID: 14690610; Rea et al, 2000, PMID: 10949293). Further there are many other examples of these HMTs collaborating in maintaining heterochromatin to keep chromatin repressed. We now extend this concept of closed chromatin conformation to maintain replication fork stability upon replication stress. Though our work does not address the underlying nature of this collaboration between HMTs, but it is clear from our data that checkpoint activated G9a is critical to deposit H3K9me marks at replication forks. These marks may prime Suv39h1 to accumulate at stressed replication forks, and the synergistic action of these HMTs may accelerate the catalytic reactions to establish compact chromatin, as suggested by the significant accumulation of H3K9me2/me3 levels within 20-30 mins of HU treatment (**Fig. 2 c and new Extended Data Fig. 4b**). The fast accumulation of heterochromatin may ensure that nascent DNA at stressed forks is protected from the action of nucleases, PRIMPOL, or the transcription machinery, in order to maintain genome stability. Consistently, we observed a higher enrichment of these factors at stressed forks in the absence of G9a activity using quantitative iPOND-MS (**Figs. 4a, 5a-c and Tables S4**). Secondly, in addition to induction of chromatin compaction at stressed forks, these HMTs may protect the substrate by binding H3K9me to exclude untimely accumulation of demethylases, such as KDM3A, which also recognize methylated H3K9 for their binding and can lead to untimely restart of stressed forks via PRIMPOL-mediated non-canonical pathways.

4. *Some of the experimental methods need to be more fully described, either in the results section or in the methods, for example the rationale for using Cldu and IdU to examine fork restart rates. I don't think either thymidine analog is described anywhere in the paper.*

Altogether, this study presents a wealth of interesting results regarding changes in proteins bound to stressed replication forks and should be of great interest to the field.

We apologize for the lack of clarity. We have now updated the information on the rationale and use of these thymidine analogs for DNA fiber assays in the manuscript.

Reviewer #3:

Remarks to the Author:

This is an interesting and nicely executed study showing the action of the histone methyl-transferase EHMT2/G9a in heterochromatin formation at stressed replication forks. After realizing of the high density of repressive epigenetics marks in cancer cells, whose hallmark is replication stress, the authors explored whether there was a correlation between both events, to show that after HU treatment cells increased the levels of H3K9me3. ChIP-seq analysis of this mark revealed a genome-wide re-distribution of H3K9me3 upon replication stress (HU) that the authors attribute to heterochromatinization at sites of replication fork stalls. Using super-resolution STED microscopy authors found a significant overlap between H3K9me3 and replication sites in the presence of HU, that was not observed in their absence. Then, using an improved ChromStretch technique, the authors observed the accumulation of H3K9me3 at stalled replication forks in HU, a result that was extended for H3K9me1 and H3K9me2. To determine a cause-effect relationship of these epigenetic marks and histone methyl-transferases, authors show, by using G9a KO cells and a catalytic inhibitor of G9a, a drastic reduction in methylation of H3K9. Further analyses by iPOND and SILAC allowed them to identify a number of factors enriched or reduced at replication forks that were corroborated by PLA. Among the lost ones are known fork-protection factors whereas among the enriched factors are FANCD2, RAD51, MCM, SMARCAL1, PrimPOL or the KDM3A demethylase. In the last part of the study the authors show that under conditions lacking H3K9 methylation DNA replicates faster and leaves stretched of ssDNAs that are consistent with a restart by PrimPOL. In agreement, forks are degraded (not protected) and re-start is slow supporting that H3K9 methylation is essential for fork protection and also for fork re-start.

Altogether, they conclude that fork protection involves transient chromatin compaction at forks.

The study is solid and provides a rational and convincing model on the methylation of histone H3K9 as a mark of chromatin condensation. Authors propose that this is a transient process, as it would be reverted by the JMJD1A/KDM3A demethylase, that is associated with stressed replication forks, distinct to those previously shown for ATRX/DAXX, as in this study the condensation is checkpoint dependent in contrast to that of ATRX. It is a sound study that deserves publication and will be of great interest in the field of genome integrity and

replication stress. Few suggestions and questions are indicated below that may serve to strengthen the conclusions and clarify some points.

We greatly appreciate reviewer's positive comments.

Specific comments:

1. The experiments of colocalization of EdU and H3K9me3 have the caveat of not being a single fork (as they discuss) but also, they may identify ongoing forks and not just stalled forks. Upon nucleotide depletion, forks stall and this causes the firing of extra dormant or late origins (see for instance PMID: 12914702). Thus, one possibility not to be dismissed is that new origins in heterochromatin regions fire and this would explain the colocalization with H3K9me3. Discuss.

We agree with the reviewer that not all forks may stall at the same time. The HU time-course analysis shows a slowing down forks upon 20 mins of HU addition, at which time the H3K9me3 signal becomes visible and subsequently increases over time. These forks are, for the most part, annotated as 'stressed forks' and may include a combination of ongoing, slower or stalled forks upon replication stress. As per the reviewer's suggestion we have edited the text accordingly.

Furthermore, there may be a small percentage of dormant origins that lie in an existing heterochromatin region that may show EdU labelling, and therefore show a colocalizing H3K9me3 signal. However, unlike regions shown in **Fig. 2b** and **new Extended Data Fig. 3a-c**, such heterochromatin regions generally show longer stretches of H3K9me3 signal over the chromatin fiber and regardless of EdU label. Below we have provided some examples (*see attached Figure B*). Throughout our analysis we have avoided such fibers, in order to show the actual effects of replication stress in *de novo* establishment of heterochromatin at EdU labelled replicating sites in the euchromatic regions. These regions specifically show H3K9me3 upon HU treatment and generally overlap with the EdU signal and do not extend beyond it (**Fig. 2b** and **new Extended Data Fig. 3a-c**). We have also discussed the possibility of H3K9me3 signal appearing at origins of replication that exist in heterochromatin regions in the revised manuscript.

Figure B. Representative heterochromatin fibers

a: Left: Representative non-replicating heterochromatin fiber. Right: Intensity profile of EdU (red), H3K9me3 (green) and H3 (magenta) along the fiber.

b: Left: Representative replicating heterochromatin fiber. Right: Intensity profile of EdU (red), H3K9me3 (green) and H3 (magenta) along the fiber.

(Note: The 20 mins of EdU pulse labels 20-40kb of genomic region, according to 1kb/min. speed of fork estimated in MRC5 cells equals to 40kb region labelled by a bidirectional fork. The cis regions in the vicinity of EdU label shows extended H3K9me3 signal representing heterochromatin region unlike *de novo* heterochromatin that is mostly restricted to nascent DNA upon replication stress shown in Fig. 2c and Extended Data Fig. 3a-c.

2. *The ChromStretch technique reveals an overlap between H3K9me3 and EdU signals specifically when cells are treated with HU. However, the intensity of H3K9me3 is not further increased respect to the intensity of H3 (Figure 3g). This could mean that the increase of methylation detected is just a consequence of higher histone levels. Since these are chromatin fibers, it is important to clarify the meaning of higher histone intensity. Could it mean that there is more signal per DNA length or that fibers are not stretched enough in that region? Can authors rule out this last possibility? This is relevant, because if that was the case, the observations could be interpreted by a different structure of the chromatin fiber that impairs stretching rather than an accumulation of certain mark. In this regard, it does not look like that the genome-wide experiment is informative since the signals detected are not quantitative and the distribution profiles of total H3 look quite similar with or without replicative stress.*

We thank the reviewer for this insightful question. Heterochromatin formation is known to increase the density of nucleosomes which effects chromatin compaction that prevents access to underlying DNA. Such compaction has been shown to occur due to the suppressed turnover of deacetylated H3K9me3 modified nucleosomes in heterochromatin regions (Aygün et al, 2013, PMID: 23604080; Taneja et al, 2017, PMID: 28318821). Indeed, stretching variability is likely to occur between fibers, however, this will be the case regardless of HU treatment or susceptibility to EdU labelling. We have provided more examples of stretched fibers with longer regions to show the higher H3 and H3K9me3 correlation with the regions containing the EdU label than with the non-EdU labelled regions and with HU treatment (**Fig. 2b and new Extended data 3a-c**). These data strongly suggest that the correlation of higher H3/H3K9me3 is not the result of lesser stretching only at EdU labelled regions and only upon HU treatment when compared to fibers isolated from untreated cells.

3. *Page 3. Line 25. It is not clear why H3K9me is looked at specifically from the beginning. It seems that the reason is that previous studies that have reported an increment in H3K9me3 levels upon oncogene-induced replicative stress conditions (stated in line 44) and because they have previously reported higher levels of H3K9me1 upon replication stress (ref 39) rather than because of the increase in cancer cell lines (Extended Data Fig 1 A). Indeed, Extended Data Fig 1 A shows the analysis of ovarian cancers but experiments are then performed in TIG3 cells (lung fibroblasts). What happens in other cancers? Why are these cells used? It is also not clear or explained why H3K9 methylation levels are looked together with H3K14ac0. Clarify*

We appreciate the reviewer's curiosity. It is in fact true that the clue that H3K9me modifications play a critical role in adapting to replication stress came not only from our unbiased histone PTMs analysis in cells exposed to persistent mild replication stress but also from an independent collaborative project in which screening of various inhibitors of chromatin modification identified UNC0642, a G9a inhibitor, as one of the top candidate to sensitize cancer cells viability (unpublished findings, in personal communication with co-author R.Kanaar). Consistent with our studies performed in parallel, unbiased histone PTMs proteomics analysis also showed higher accumulation of H3K9me3 specifically in cells undergoing replication stress (**Fig.1c and Table S1**). We pursued these common threads pointing towards a critical role of increased accumulation of H3K9me3 upon replication stress by first performing a global chromatin profiling of various cancer cell lines for reported histone modifications which were frequently profiled as combinations (such as, H3K9me3K14ac0 or H3K9me3K14ac1 etc.) in publicly available data [Ghandi et al, Nature, 2022 (PMID: 31068700)]. Analysis of this data revealed the accumulation of heterochromatin signatures [i.e. deacetylated (ac0) and methylated K9 (me1/2/3) marks] shown in **Extended Data Fig. 1a** and led to the observation that most ovarian cancer cell lines show higher enrichment of these heterochromatin signatures.

After making these observations based on the multiple independent analysis, we wanted to understand how H3K9me marks are established upon replication stress using a genetic background that was simpler than those of the various cancer cell lines which have complex genetic predispositions and endogenous replication stress due to oncogene mutations. Therefore, we used human fibroblast cells to keep the system as close to wildtype as possible in order to study the specific effects of replication stress in modulating chromatin landscape at replication forks.

4. *Page 4. Line 5. The observed increase in H3K36me2 that is detected is not discussed or studied any further. Why? Is it also detected in cancer databases?*

We appreciate the reviewer's comment, and we have further discussed the finding of H3K36me2 in the discussion (*Pg12, paragraph 3*) but did not choose to pursue it further in order to limit the scope of this study to H3K9me marks.

Although the altered levels of H3K36me2 or the elevated expression of writers of H3K36me2 has been reported in various cancers (Yuan et al, 2020, PMID: 32188706; Kuo et al, 2011, PMID: 22099308), the high prevalence of this mark throughout the genome and the interplay between H3K36me2 and H3K27me mark is complex (Popovic et al, 2014, PMID: 25188243; Yuan et al, 2011, PMID: 21239496; Finogenova et al, 2020, PMID:

33211010). Therefore, for this study, we have focused on the more straightforward increased enrichment of the H3K9me3 mark upon replication stress.

5. Page 5. Line 45. Were H3K9me1 and me2 increased levels not detected at the masspec analysis upon HU? In that case, why?

The reviewer is correct, in that our proteomics analysis performed upon chronic replication stress did not show accumulation of H3K9me1 and me2 peptides upon replication stress as was seen for H3K9me3 (as shown in **Table S1** and now modified **Fig. 1c**). We believe this is due to the saturation of the H3K9me1/me2 substrate as it is converted to H3K9me3 upon the longer treatments with HU. The dynamics of the transition from a lower to higher state of H3K9me modification could be captured when a shorter treatment (1hour) of HU was used and all three states could be found enriched at replicating sites (**Fig. 2e and Extended Data Fig. 4d-f**). Supporting this, our analysis of HU time courses with H3K9me2 (**new Extended Data Fig. 4b**) shows that H3K9me2 levels are saturated by 30 mins, implying that the substrate is continuously used to establish H3K9me3 which, concomitantly, showed a gradual increase in its enrichment up to 1hour HU treatment (**Fig. 2c**).

6. Extended Data Fig. 3g-h should be in main figures as these data are crucial to state the dependency on the DNA replication checkpoint and to draw ATR in the model in Figure 3f and the final model.

As per the reviewer's suggestion we have moved the ATR regulation data from previously *Extended Data Fig. 3g-h* to **Fig. 3e**.

7. Page 6. Line 34. The model includes HDAC1 and deacetylation of lysine 16 on histone H4 but this is not based on any data from this manuscript. The author state here that 'replication fork restart alleviated these effects (Extended Data Fig. 4c-d)' but the effects only refer to H3K9me. It is not clear why the authors add HDAC1 acetylation to their model without experiments that support this role.

We apologize to the reviewer for lack of clarity. The data given in **Extended Data Fig. 5c-d** (previously *Extended Data Fig. 4c-d*) shows dynamic changes in histone deacetylase, HDAC1, at replication sites upon HU stress, i.e. increased levels of HDAC1 at EdU labelled sites detected by a Proximity Ligation Assay (PLA) upon replication stress (HU condition). The levels of HDAC1 are significantly reduced to the those of the untreated condition when the cells are released from HU stress (**Extended Data Fig. 5c**), and a similar pattern is observed for H3K9me enrichment (**Fig. 3a-c**). Furthermore, **Extended Data Fig. 5d** (previously *Extended Data Fig. 4d*) shows the level of H4K16ac, a substrate for HDAC1, at replication sites which is significantly reduced upon HU treatment in comparison to untreated (UT) or the HU release condition, suggesting that nucleosomes are deacetylated at replication sites upon replication stress (i.e. HU condition). This result is consistent with the observation of increased HDAC1 enrichment upon HU treatment. Furthermore, treatment of cells with UNC0642 that blocks the G9a enzymatic H3K9 methyltransferase activity, abolishes the changes in both HDAC1 or H4K16ac, suggesting that a platform of H3K9me is critical for the recruitment of HDAC1 in order to deacetylate nucleosomes that are critical to maintaining a closed chromatin conformation. Together these data have been depicted in **Fig. 7** to show the components of heterochromatin enriched at replication sites upon replication stress.

8. Authors propose that heterochromatin facilitates the loading of fork protection factors whereas it reduces that of PrimPOL and other proteins that may compromise genome integrity. This is a case in which PrimPOL is suggested to play a mutagenic role due to its potential to leave unreplicated ssDNA stretches behind the fork. Would PrimPOL overexpression increase ssDNA gaps under HU conditions and/or under G9a inhibition?

This is a great suggestion by the reviewer. We have performed the experiments by using the PRIMPOL over-expression system developed previously (Quinet et al, 2020, PMID: 31676232). Upon over-expression of PRIMPOL, we observed that ssDNA gaps accumulated behind the fork upon HU treatment suggesting that presence of increased level of PRIMPOL upon slowing down stressed forks can be toxic. The accumulation of ssDNA gaps further increased significantly when G9a activity was perturbed in the presence of enhanced PRIMPOL levels, suggesting that H3K9me3 accumulation at nascent DNA prevents repriming initiation by PRIMPOL. Thus, these findings together with earlier observations further strengthen our model that *de novo* heterochromatin formation at nascent DNA denies access to PRIMPOL in order to maintain genome integrity (Figs. 4a, 5c, 6a and Extended Data Fig. 7i). We have added this analysis in the revised manuscript as **new Extended Data Fig. 7i**.

Minor comments:

• *Page 6. Line 25. The authors state that 'combining knock down of Suv39h1 with inhibition of G9a did not significantly increase the accumulation defect of H3K9me3, suggesting that both G9a and Suv39h1 act in the same pathway (Fig. 3e).' However, the defect is already suppressed at all in both single knock down and inhibition conditions (and of course in the combination of both) and therefore it is difficult to draw any genetic conclusion regarding the pathways.*

We agree with the reviewer's suggestion and have changed the text in the manuscript accordingly.

• *Page 5. Line 30. Figure 2e is cited before 2d.*

We apologize for this error and have corrected it in the text.

• *Figure 4d-e axis legends are misspelled.*

We apologize for this error and have corrected the figure legends accordingly.

In summary, we have addressed the concerns raised by the reviewers and we hope that the manuscript is now suitable for publication in Nature Cell Biology.

Sincerely,
Nitika Taneja

Decision Letter, first revision:

Our ref: NCB-A49954A

13th April 2023

Dear Dr. Taneja,

Thank you for submitting your revised manuscript "Dynamics of de novo heterochromatin assembly and disassembly at replication forks ensures fork stability" (NCB-A49954A). It has now been seen by the original referees and their comments are below. Thank you for your efforts to address their points in revision to strengthen the main conclusions. The reviewers find that the paper has improved in revision, and therefore we'll be happy in principle to publish it in Nature Cell Biology, pending revisions to comply with our editorial and formatting guidelines.

The current version of your manuscript is in a PDF format; could you please email us a copy of the file in an editable format (Microsoft Word or LaTeX), as we can not proceed with PDFs at this stage? Thank you in advance for your help with this point.

Once the Word file is in, we will be performing detailed checks on your paper and will send you a checklist detailing our editorial and formatting requirements in about 1-2 weeks. Please do not upload the final materials and make any revisions until you receive this additional information from us.

Thank you again for your interest in Nature Cell Biology. Please do not hesitate to contact me if you have any questions.

Sincerely,

Melina

Melina Casadio, PhD
Senior Editor, Nature Cell Biology
ORCID ID: <https://orcid.org/0000-0003-2389-2243>

Reviewer #1 (Remarks to the Author):

the key results have not changed in the revised paper and represent a major advance in understanding the deposition of H3K9me after replication and replication stress. The results are original, rigorous and highly significant. The improvements made are significant, and this reviewer strongly appreciates the more balanced treatment of G9a as a H3K9me2 depositing enzyme (supp figure 4) and the fact that H3K9me2 is converted to H3K9me3 by another enzyme. As now presented the work is consistent with a large body of other work (in vivo) showing that G9a preferentially deposits me2 which is then converted to me3 by SETDB1 or SUV39H1/2. The data are well presented, conclusions are robust and I think the added data and experimentation

has enriched and clarified the story. I agree with the authors that adding a single barrier site for replication indeed requires a lot of development and is beyond the scope of this article. The references are now more complete (for a summary of the different roles of the mammalian HMTs see doi: 10.1038/s41580-022-00483-w. Epub 2022 May 1) and the clarity of the text is improved and excellent.

Reviewer #2 (Remarks to the Author):

In their revised manuscript, the authors have fully and thoroughly addressed my concerns and those of the other referees. I support publication in Nature Cell Biology.

Reviewer #3 (Remarks to the Author):

Authors have improved considerably the manuscript and I am satisfied with this new version

Decision Letter, final checks:

Our ref: NCB-A49954A

20th April 2023

Dear Dr. Taneja,

Thank you for your patience as we've prepared the guidelines for final submission of your Nature Cell Biology manuscript, "Dynamics of de novo heterochromatin assembly and disassembly at replication forks ensures fork stability" (NCB-A49954A). Please carefully follow the step-by-step instructions provided in the attached file, and add a response in each row of the table to indicate the changes that you have made. Please also check and comment on any additional marked-up edits we have proposed within the text. Ensuring that each point is addressed will help to ensure that your revised manuscript can be swiftly handed over to our production team.

In recognition of the time and expertise our reviewers provide to Nature Cell Biology's editorial

process, we would like to formally acknowledge their contribution to the external peer review of your manuscript entitled "Dynamics of de novo heterochromatin assembly and disassembly at replication forks ensures fork stability". For those reviewers who give their assent, we will be publishing their names alongside the published article.

Nature Cell Biology offers a Transparent Peer Review option for new original research manuscripts submitted after December 1st, 2019. As part of this initiative, we encourage our authors to support increased transparency into the peer review process by agreeing to have the reviewer comments, author rebuttal letters, and editorial decision letters published as a Supplementary item. When you submit your final files please clearly state in your cover letter whether or not you would like to participate in this initiative. Please note that failure to state your preference will result in delays in accepting your manuscript for publication.

Cover suggestions

As you prepare your final files we encourage you to consider whether you have any images or illustrations that may be appropriate for use on the cover of Nature Cell Biology.

Nature Cell Biology has now transitioned to a unified Rights Collection system which will allow our Author Services team to quickly and easily collect the rights and permissions required to publish your work. Approximately 10 days after your paper is formally accepted, you will receive an email in providing you with a link to complete the grant of rights. If your paper is eligible for Open Access, our Author Services team will also be in touch regarding any additional information that may be required to arrange payment for your article.

Please note that *Nature Cell Biology* is a Transformative Journal (TJ). Authors may publish their research with us through the traditional subscription access route or make their paper immediately open access through payment of an article-processing charge (APC). Authors will not be required to make a final decision about access to their article until it has been accepted. Find out more about Transformative Journals

Authors may need to take specific actions to achieve compliance with funder and institutional open access mandates. If your research is supported by a funder that requires immediate open access (e.g. according to Plan S principles) then you should select the gold OA route,

and we will direct you to the compliant route where possible. For authors selecting the subscription publication route, the journal's standard licensing terms will need to be accepted, including self-archiving policies. Those licensing terms will supersede any other terms that the author or any third party may assert apply to any version of the manuscript.

Please use the following link for uploading these materials:
[Redacted]

Best regards,

Kendra Donahue
Staff
Nature Cell Biology

On behalf of

Melina Casadio, PhD
Senior Editor, Nature Cell Biology
ORCID ID: <https://orcid.org/0000-0003-2389-2243>

Reviewer #1:

Remarks to the Author:

the key results have not changed in the revised paper and represent a major advance in understanding the deposition of H3K9me after replication and replication stress. The results are original, rigorous and highly significant. The improvements made are significant, and this reviewer strongly appreciates the more balanced treatment of G9a as a H3K9me2 depositing enzyme (supp figure 4) and the fact that H3K9me2 is converted to H3K9me3 by another enzyme. As now presented the work is consistent with a large body of other work (in vivo) showing that G9a preferentially deposits me2 which is then converted to me3 by SETDB1 or SUV39H1/2. The data are well presented, conclusions are robust and I think the added data and experimentation has enriched and clarified the story. I agree with the authors that adding a single barrier site for replication indeed requires a lot of development and is beyond the scope of this article. The references are now more complete (for a summary of the different roles of the mammalian HMTs see doi:

10.1038/s41580-022-00483-w. Epub 2022 May 1) and the clarity of the text is improved and excellent.

Reviewer #2:

Remarks to the Author:

In their revised manuscript, the authors have fully and thoroughly addressed my concerns and those of the other referees. I support publication in Nature Cell Biology.

Reviewer #3:

Remarks to the Author:

Authors have improved considerably the manuscript and I am satisfied with this new version

Final Decision Letter:

Dear Dr Taneja,

I am pleased to inform you that your manuscript, "Dynamic de novo heterochromatin assembly and disassembly at replication forks ensures fork stability", has now been accepted for publication in Nature Cell Biology. Congratulations on this very nice work!

Please note that *Nature Cell Biology* is a Transformative Journal (TJ). Authors may publish their research with us through the traditional subscription access route or make their paper immediately open access through payment of an article-processing charge (APC). Authors will not be required to make a final decision about access to their article until it has been accepted. Find out more about Transformative Journals

If you have not already done so, we strongly recommend that you upload the step-by-step protocols used in this manuscript to the Protocol Exchange (www.nature.com/protocolexchange), an open online resource established by Nature Protocols that allows researchers to share their detailed experimental know-how. All uploaded protocols are made freely available, assigned DOIs for ease of citation and are fully searchable through nature.com. Protocols and Nature Portfolio journal papers in which they are used can be linked to one another, and this link is clearly and prominently visible in the online versions of both papers. Authors who performed the specific experiments can act as primary authors for the Protocol as they will be best placed to share the methodology details, but the Corresponding Author of the present research paper should be included as one of the authors. By uploading your Protocols to Protocol Exchange, you are enabling researchers to more readily reproduce or adapt the methodology you use, as well as increasing the visibility of your protocols and papers. You can also establish a dedicated page to collect your lab Protocols. Further information can be found at

www.nature.com/protocolexchange/about

With kind regards,

Melina

Melina Casadio, PhD
Senior Editor, Nature Cell Biology
ORCID ID: <https://orcid.org/0000-0003-2389-2243>
